# Wonderful Team: Zero-Shot Physical Task Planning with Visual LLMs

**Zidan Wang** *                                              *zidanwang2025@u.northwestern.edu*
*Department of Statistics, Northwestern University*

**Rui Shen** *                                                      *kqn9mt@virginia.edu*
*Department of Computer Science, University of Virginia*

**Bradly Stadie**                                              *bstadie@northwestern.edu*
*Department of Statistics, Northwestern University*

**Reviewed on OpenReview:** *https://openreview.net/forum?id=udVkqIDYSM*

## Abstract

We introduce Wonderful Team, a multi-agent Vision Large Language Model (VLLM) framework for executing high-level robotic planning in a zero-shot regime. In our context, zero-shot high-level planning means that for a novel environment, we provide a VLLM with an image of the robot's surroundings and a task description, and the VLLM outputs the sequence of actions necessary for the robot to complete the task. Unlike previous methods for high-level visual planning for robotic manipulation, our method uses VLLMs for the entire planning process, enabling a more tightly integrated loop between perception, control, and planning. As a result, Wonderful Team's performance on real-world semantic and physical planning tasks often exceeds methods that rely on separate vision systems. For example, we see an average 40% success rate improvement on VimaBench over prior methods such as NLaP, an average 30% improvement over Trajectory Generators on tasks from the Trajectory Generator paper, including drawing and wiping a plate, and an average 70% improvement over Trajectory Generators on a new set of semantic reasoning tasks including environment rearrangement with implicit linguistic constraints. We hope these results highlight the rapid improvements of VLLMs in the past year, and motivate the community to consider VLLMs as an option for some high-level robotic planning problems in the future.

## 1 Introduction

High-level robotic planning is a difficult task to pin down. It requires an extensive knowledge of the physical world. If planning is to be done via language instructions, it also requires a semantic understanding of the language itself, and how the objects in the environment relate to sometimes vague linguistic instructions. Consider, for example, the task "Pick up the object to the right of the grapes and put it in the box." To solve this task, we need to identify 'the object to the right of the grapes,' which is a non-trivial visual relationship inference problem. We also need to consider the implicit constraints of the environment. For example, if the box has a closed lid, the task necessitates reasoning about lid removal before the object can be placed inside.

Recently, Large Language Models (LLMs) have emerged as an enticing pathway for achieving high-level robotic planning. LLMs already have a built-in world model, including priors over object names and relative relationships. They also have the ability to understand and parse semantic meaning from language instructions. Several promising pipelines exist for converting language commands to high-level robotics plans Kwon et al. (2024). Typically, if vision-based planning is required, these pipelines make use of an off-the-shelf vision encoder such as LangSAM or OWL-ViT, which finds objects in an environment and

---

*Equal contribution.

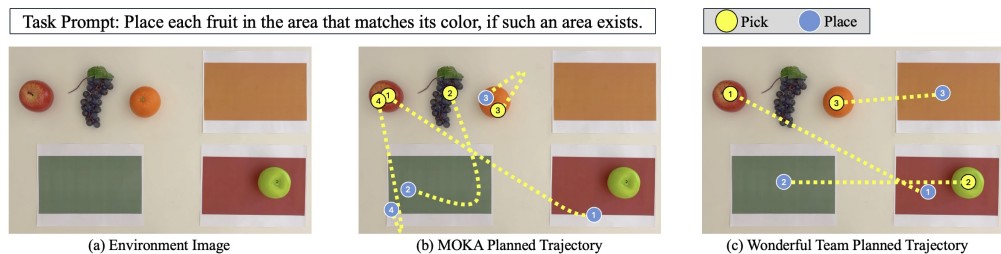

Figure 1: Comparison of plans for a color-matching fruit placement task. MOKA's trajectory (b) shows limitations arising from both unverified plans (e.g., step 2 treating the grape as green) and misalignment between visual grounding and VLLM planning (e.g., step 3 misidentifying the orange fruit as the 'orange area' and step 4 failing to pick the green apple due to unspecified color). These errors are often irrecoverable, even with VLLM correction (see Figure 15). See Appendix G.4 for details on the comparison with MOKA.

delivers their coordinates in pixel space. While this is a great idea in theory, we find that, in practice, such reliance on an external vision system often results in pipelines that are brittle and unable to recover from mistakes originating in the visual system.

In this paper, we consider the problem of using Visual Large Language Models (VLLMs) to generate action-level robotics plans directly from environmental images. Compared to prior works, we find that having VLLMs as the backbone of both planning and grounding in the pipeline offers several advantages. First, VLLMs combine the semantic reasoning power of language models with robust visual understanding, bridging the traditional gap between high-level planning and visual grounding as shown in Figure 1. More importantly, VLLMs enable self-correction at both the planning and perception levels. For planning, if the initial high-level plan overlooks environmental constraints, VLLMs can identify these errors and suggest revisions. At the perception level, while specialized detection models fail irreparably when objects cannot be found, our VLLM-based perception agents can refine their output through multimodal self-correction. By using updated visual feedback and annotations that include richer multimodal information, like the spatial relations between bounding boxes and nearby objects, the system improves its predictions and fixes errors more effectively. Our framework enhances error recovery by combining self-feedback with external guidance from higher-level agents when self-correction alone fails. This approach significantly improves task success rates, with real-world spatial and semantic planning tasks showing an approximate 80% increase compared to a baseline GPT-4o model without self-correction. Many tasks improve from 0% to 100% success rates with the introduction of self-correction. For detailed ablation studies on the impact of each component in the Wonderful Team framework, see Appendix E.

Our contributions can be summarized as follows:

- **Zero-Shot Coordinate-Level Control in Complex Robotics Tasks:** Our system operates without prior training, fine-tuning, or detailed task prompts and performs robustly across diverse domains while requiring minimal domain knowledge.

- **Development of a Multi-Agent VLLM Framework to Overcome Limitations in Robotic Planning:** We have developed a novel hierarchical multi-agent structure where specialized agents collaboratively handle various aspects of demanding robotics tasks, from high-level robotic planning to low-level position-based controlling. By integrating perception and action, enabling inter-agent communication, and employing a divide-and-conquer approach with reflection capabilities, we address the shortcomings of previous models, including challenges in long-horizon planning, implicit goal reasoning, context-aware object identification, irrecoverable perception failure, precise localization, and managing multiple instances of the same object.

- **Empirical Validation through Extensive Experiments and Ablation Studies:** We validated our framework through extensive tabletop experiments, including long-horizon pick-and-place, sweeping, and fine-grained trajectory planning and generation, demonstrating its effectiveness beyond standard

pick-and-place tasks. It outperformed other zero-shot approaches and fine-tuned vision-language action models, attributed to its self-correction mechanism and multi-agent system design, which enable robust performance in tasks requiring complex planning and precise object grounding. We additionally explored tasks outside our scope, including navigation and grounding in cluttered environments, showing our system's potential for generalization. Detailed results and discussions are provided in Appendix B.

Demonstration videos of the robotic policies in action, along with the code and prompts, can be accessed on our  project website .

## 2 Motivating Examples

Developing robotic systems that can understand and execute planning over complex manipulation tasks remains a significant challenge, particularly when dealing with uncommon objects and abstract visual attributes. Existing frameworks often employ a Large Language Model (LLM) as a text planner combined with a separate vision model (e.g., CLIP, OWL-ViT, LangSAM, GroundingDINO) to perceive the environment. While this modular approach seems logical, it faces critical limitations when handling less common objects, interpreting abstract visual relationships, or reasoning about semantic properties - challenges where our integrated approach shows significant improvements over prior methods.

### 2.1 Can an LLM Planner with a Separate Vision Model Find Objects?

**Not Always.** Specialized detection models have limitations in context-aware perception. In both Trajectory Generator and MOKA, GPT extracts the objects to segment based on the task prompt and passes them to a separate detection model (LangSAM and GroundDINO, respectively) for object identification and segmentation. As illustrated in Figure 2, LangSAM and GroundingDINO both fail to correctly identify or segment target objects based on the provided prompt. While these examples highlight several challenges inherent in using separate vision models for complex tasks, they do not capture the full scope of limitations, which are discussed in detail below:

1. Difficulty with Less Common or Hard-to-Bound Objects: they struggle with identifying uncommon objects (e.g., robot grippers, box lids), as seen in Figure 2 (a), and abstract regions that cannot be clearly segmented. They also fail when objects are less prominent or boundaries are ill-defined.

2. Misinterpretation of Spatial and Positional Instructions: they often misinterpret vague spatial instructions like "pick up the rightmost object" due to its lack of precise spatial reasoning. In multi-instance scenarios, positional references like "the middle can" are challenging because the model frequently miscounts objects, leading to incorrect identification.

3. Lack of Contextual Awareness and Differentiation: they lack contextual understanding required to differentiate between similar targets based on instructions. For example, in Figure 1, when tasked with "place fruits in the corresponding colored region," the model identifies the orange itself as both the "orange" and the "orange-colored region." While this interpretation aligns with the literal prompt, it ignores the contextual requirement and results in execution failure.

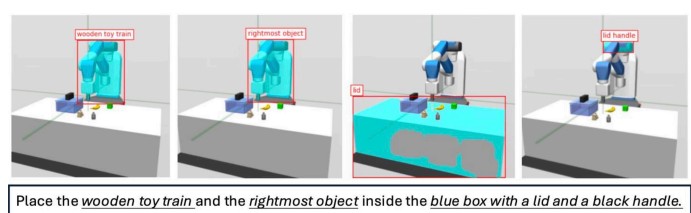

Place the *wooden toy train* and the *rightmost object* inside the *blue box with a lid and a black handle.*

(a) Segmentation Results from Trajectory Generator (LangSAM)

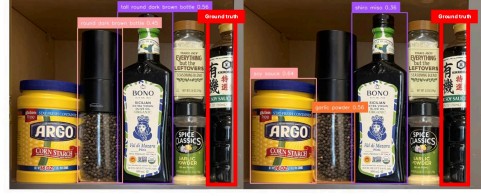

(b) Segmentation Results from MOKA (GroundingDINO)

Figure 2: Examples of detection failures by specialized segmentation models. Segmentation errors are common across prior methods, regardless of how the model is prompted. See Figure 15 for a detailed discussion on the extent of this issue and Appendix G for extensive comparisons with TG and MOKA.

**Can These Issues Be Fixed Easily?** Even though Visual LLMs can verify results from specialized detection models, correcting detection errors remains a significant challenge. Separate detection models retain full control over object identification, limiting Visual LLMs to refining prompts with alternative object descriptions. As shown in Figure 2 (b) (and more extensively in Figure 15) larger changes—such as shifting from an object name (e.g., "soy sauce") to a description of its shape or color (e.g., "tall, dark brown bottle")—can sometimes improve results. However, the finite set of descriptors for any given object can lead to a dead-end. In some cases, this iterative process may become an inefficient loop, failing to isolate the desired object, especially when the optimal descriptor does not succeed in the given context.

However, recent advancements in VLLMs present a potential solution, as they are designed to handle both visual reasoning and context understanding. This brings us to the question:

## 2.2 Could simply replacing LangSAM with a VLLM solve these issues?

The answer to this question is **'no.'** While replacing LangSAM with a VLLM may improve context comprehension, the resulting system fails to match the precision that LangSAM already provides. Thus, the substitution is largely ineffective. There are a few reasons for this.

1. Imprecise Spatial Understanding: Recent VLLMs can generate more accurate approximate locations, but they still lack the precision required for effective robotic manipulation. In our ablation experiments, 90% of the coordinates were close to the target (Table 13), yet only 33% (GPT-4o) were accurate enough to be directly actionable (Table 12).

2. Difficulty with Complex Instructions: Tasks that require understanding spatial relationships or handling multiple objects can overwhelm the reasoning capabilities.

However, while experimenting with using VLLMs to generate high-level plans, we made an interesting observation: **VLLMs often know they're wrong,** and they can often diagnose their own errors.

For example, when asked to locate a cluster of grapes, the model may initially provide an imprecise answer, but can correct it when prompted to reassess (see Figure 3). Table 14 shows GPT-4o's 97% success in classifying bounding boxes, highlighting its self-assessment abilities. This suggests VLLMs can iteratively refine outputs, even from initially imprecise coordinates.

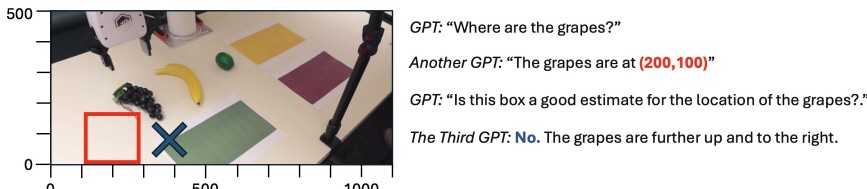

Figure 3: An example of multiple VLLMs working together to recognize and correct an error in object positioning upon review.

Even more interestingly, we also noticed that **VLLMs often know how to iteratively self-correct their perception errors.** Over several iterations, they can improve their estimation of an object's position, moving closer to the correct target (see Figure 4). This process of an LLM iteratively refining its outputs based on feedback is known as *reflection*.

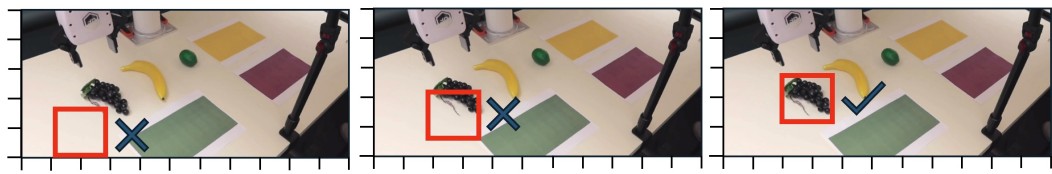

Figure 4: A VLLM improving its estimation of the grapes' position over several iterations.

Thus, we arrive at our main thesis. While naively using a VLLM for high-level robotic planning is often insufficient, leveraging their self-correction capabilities in a structured way can substantially improve their effectiveness.

## 3 Wonderful Team

Building on insights from the previous section, we propose a novel pipeline for high-level physical task planning in robotics that leverages specialized agents, each responsible for a distinct part of the reasoning process within a structured framework. By combining the strengths of Vision-Language Models (VLLMs) and breaking down complex tasks into manageable components, each agent can focus on a specific role, which results in more precise and reliable high-level planning. As illustrated in Figure 6, our multi-agent framework defines the distinct roles of each agent, the flow of information from high-level tasks to low-level actions, and their collaborative efforts in executing tasks effectively.

We discuss the role of each team member and the flow of information between them below. Simplified prompts and more details on the implementation and workflow for each team member can be found in Appendix C. Full prompts are available in the project's codebase.

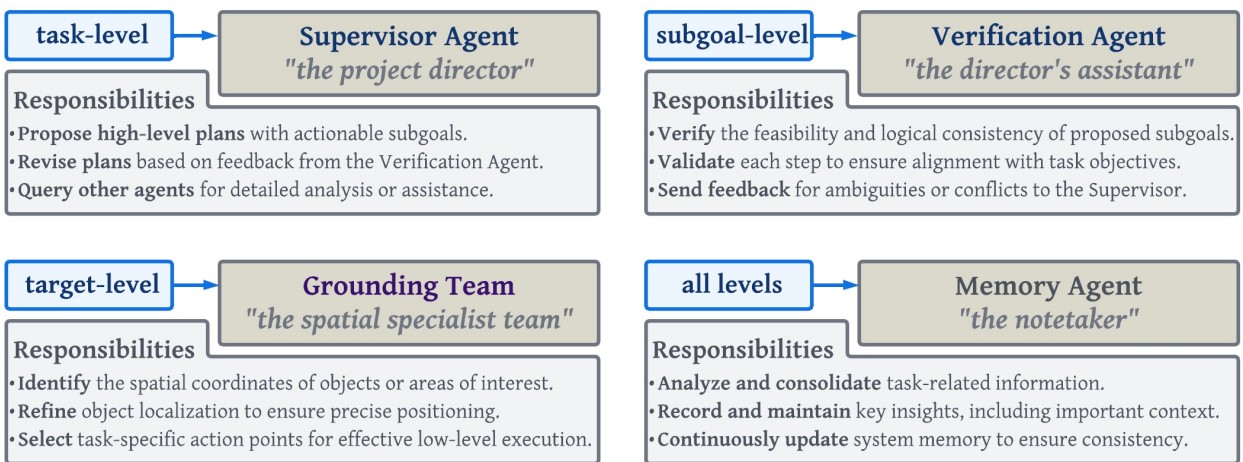

Figure 5: Overview of the major components of the Wonderful Team. Each part of the pipeline receives a different level of input, with a unique scope and specialization within the project. The agents collaboratively handle tasks ranging from high-level planning and logical verification to precise spatial reasoning and memory management, ensuring robust and efficient execution.

Each agent in our system is designed to address specific challenges in high-level planning for physical manipulation tasks. For example, in Figure 6(b), when given the instruction 'put the banana into the box,' the Supervisor agent's initial plan often overlooks obstacles like the box's lid. This is where the Verification agent plays a critical role. Its reflection process involves reviewing the subgoal plan, checking for potential issues, such as physical constraints or incomplete steps, and cross-referencing this plan with the current state of the environment. If an issue, like the lid blocking access to the box, is detected, the Verification agent raises this concern to the Supervisor. This early feedback allows the system to refine the plan before executing any action.

The Grounding team then takes over to refine the coordinates for each target, ensuring precise and collision-free movements. The Mover and Checker agents collaborate through an iterative process of refining positional groundings. Figure 4 provides an example of the Grounding team in action. The separation of tasks into a multi-agent system proves advantageous, as it allows each agent to focus on its distinct responsibilities with varying levels of access to critical information. For a detailed discussion on the benefits of this multi-agent approach, refer to Appendix G.3.3.

**Are all components of the Wonderful Team necessary?** Ablation studies show that all components of the Wonderful Team are essential. Removing memory agents leads to failures, such as mistaking irrelevant objects for targets, while omitting grounding agents results in inaccurate coordinates. A supervisor-only setup works for simple tasks but fails with complex ones, lacking precision and self-correction mechanisms. Appendix E provides detailed analysis, and Table 11 shows success rate impacts. Appendix C illustrates the team's information flow.

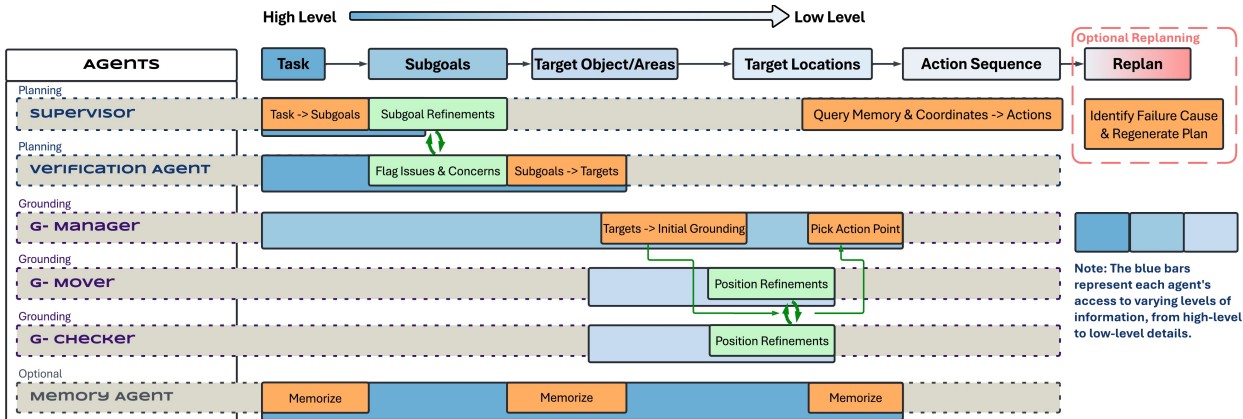

(a) This figure illustrates the agent roles and information flow within our pipeline, moving from high-level tasks to low-level actions. The blue bars indicate each agent's level of information access. For instance, the Grounding Manager has a broad overview, encompassing both the task and subgoals, while the Mover and Checker agents focus only on specific details within their target areas, without managing the entire task context.

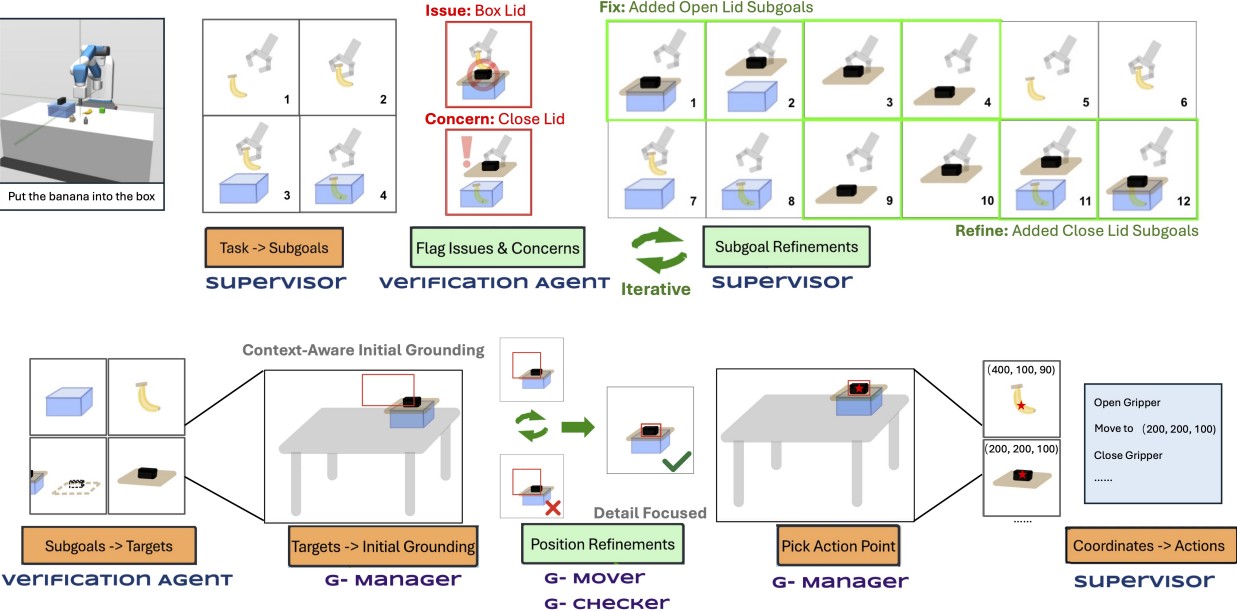

(b) A symbolic example illustrating the framework in Figure 6(a).

Figure 6: Illustration of our multi-agent framework and a symbolic example showcasing agent roles, information flow, and collaborative task execution.

# 4 Related Work

**High-Level Planning with Predefined Task Modules:** Many methods focus on high-level planning using LLMs or VLLMs, decomposing tasks into subtasks but relying on predefined task modules or APIs

for action execution, which are not directly executable without prior knowledge or training (Hu et al., 2023; Huang et al., 2022b; Liang et al., 2023).

**Low-Level Coordinate Generation with Separate Vision Models:** Other approaches generate low-level coordinates using separate vision models for perception, often relying on predefined or fine-tuned vision APIs. While leveraging off-the-shelf models like Convolutional Neural Networks (CNNs) (Ichter et al., 2022; Mees et al., 2023), CLIP (Bucker et al., 2023; Huang et al., 2022c), Vision Transformer (ViT) variants (Huang et al., 2023b; Stone et al., 2023; Jiang et al., 2023), or LangSAM (Kwon et al., 2024) has shown promise in zero-shot capabilities, these methods still face limitations. The reliance on separate perception systems can fail to fully capture the environmental context required for precise planning and action generation.

Recent advancements in robotics and artificial intelligence have integrated Large Language Models (LLMs) and Vision-Language Models (VLMs) into robotic systems. Our work builds upon and differs from several key areas in this evolving landscape.

**Foundation Models in Robotics:** Foundation models, trained on vast internet-scale datasets, have demonstrated strong zero-shot capabilities across various tasks. LLMs like GPT-3 (Brown et al., 2020), LLaMA (Touvron et al., 2023), and ChatGPT have excelled in generating human-like text, understanding natural language instructions, and performing extensive reasoning and planning. VLMs extend these capabilities by incorporating visual understanding. In robotics, these models offer the potential to endow robots with real-world priors and advanced reasoning abilities without extensive task-specific training.

**Language Models Empowering Robotics:** Prior work has leveraged natural language to enhance robotic learning and adaptation. Early approaches equipped agents with learned language embeddings, requiring large amounts of training data (Bing et al., 2023; Jiang et al., 2023). Others focused on connecting language instructions with low-level action primitives to solve long-horizon tasks (Hu et al., 2023; Huang et al., 2022b; Liang et al., 2023). While effective in specific contexts, these methods often struggle to generalize to new tasks without retraining. Foundation VLA models like RT-1 (Brohan et al., 2022), RT-2 (Brohan et al., 2023), and OpenVLA (Kim et al., 2024) have advanced versatile robotic systems, but they still require significant training to achieve robust performance across diverse tasks. Additionally, from a system design perspective, directly mapping vision to action embeddings in these approaches is less intuitive compared to breaking down the problem into interpretable subtasks or subgoals, making it harder to intuitively reason about and improve the system's behavior. For a detailed comparison with RT-1 and OpenVLA, see Appendix G.5.

**Zero-Shot and Few-Shot Approaches:** Recent studies have explored zero-shot and few-shot solutions for robotic planning and manipulation tasks (Huang et al., 2022a; Liang et al., 2023; Huang et al., 2022b;c; Zeng et al., 2023; Singh et al., 2023; Vemprala et al., 2023; Gu et al., 2023). These approaches aim to handle unseen scenarios without prior training, primarily focusing on high-level planning. However, they often rely on predefined programs or external modules for control, limiting their adaptability in dynamic or complex environments.

**Vision-Language Models for Localization:** *PIVOT* (Nasiriany et al., 2024) enables VLMs to localize actionable points without fine-tuning on task-specific data. Their approach centers on localization through visual question answering, with minimal focus on planning—similar to the role of our Grounding Team. Unlike our method, which integrates both localization and planning within a multi-agent framework, PIVOT primarily addresses localization without managing complex, long-horizon tasks. In PIVOT, a single agent iteratively selects action points, whereas our approach employs multiple agents with distinct roles for refining and verifying actions. A detailed comparison is provided in Appendix G.2.

**Language Models as Zero-Shot Trajectory Generators and MOKA:** Recent works by Kwon et al. (2024) and Fang et al. (2024) demonstrate the use of pre-trained large language models (LLMs) for generating affordance-based, point-level subgoal trajectories. Both approaches rely on dedicated object detection systems that use a combination of GroundingDINO (Liu et al., 2023b) and SAM (Kirillov et al., 2023) to extract object information, which informs LLM-based planning. These methods employ LLMs to generate executable Python scripts that define the robot's trajectory. In contrast, our approach fundamentally differs by leveraging the same VLLM for perception to seamlessly integrate planning and grounding without depending on external modules. We provide detailed comparisons in Appendices G.3 and G.4.

**Natural Language as Policies:** Concurrent with our work, *Natural Language as Policies (NLaP)* (Mikami et al., 2024) developed a few-shot, end-to-end model for coordinate-level action prediction. Their approach involves providing a one-shot example, either from the same task or a closely related one, rather than adopting a zero-shot paradigm. Unlike our method, which integrates both grounding and planning within a multi-agent framework, NLaP focuses less on grounding and directly uses system information from the environment, bypassing the need to extract coordinates from images using VLMs. NLaP serves as one of the baselines in our experiments, and a detailed comparison is presented in Appendix G.1.

**Our Contribution in Context:** Our work differs from prior approaches by proposing a zero-shot, single-model, multi-agent system that integrates high-level planning and low-level action execution within a unified VLLM framework. By eliminating the need for external vision encoders and predefined action modules, our method achieves greater adaptability and precision in dynamic environments.

## 5 Experimental Results

In this section, we evaluate the performance of Wonderful Team across a diverse set of tasks that challenge various aspects of robotic reasoning and manipulation. We address key elements of robotics, including multimodal reasoning, contextual decision-making, and complex spatial planning. Our experiments are categorized into three main groups, each designed to tackle specific challenges and contribute to the broader evaluation of the system's capabilities.

**1) Multimodal Reasoning** (17 Tasks in Simulated VIMABench)

**2) Implicit Goal Inference** (3 Custom Real-world Tasks)

**3) Spatial Planning** (4 Real-world Tasks Adapted from Trajectory Generator)

### 5.1 Multimodal Reasoning - Simulated VIMABench

To assess our approach's ability to understand multimodal prompts, reason about abstract concepts, and follow constraints, we tested it on all 17 tasks from VIMABench (Jiang et al., 2023). Unlike traditional robotics benchmarks, VIMABench offers a broad range of objects and task types (see Figure 7), requiring advanced scene understanding, multimodal comprehension, and precise planning for manipulation.

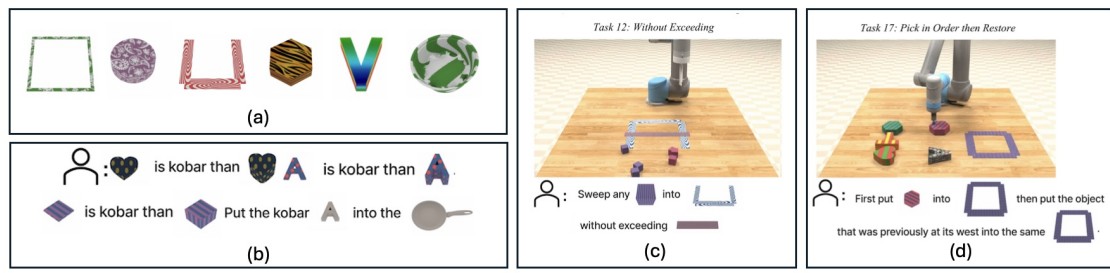

Figure 7: Key Challenges in VIMABench (Jiang et al., 2023): (a) Manipulating uncommon objects and textures, (b) Interpreting multimodal prompts with abstract nouns and adjectives, (c) Executing constraint satisfaction tasks, and (d) Handling spatial relations and sequential dependencies.
We evaluated all 17 tasks in VIMABench, categorized into four main task suites as defined by Jiang et al. (2023), each targeting distinct robotic capabilities:

**1) Simple Object Manipulation**: pick-and-place and rotate tasks using multimodal prompts that combine images and text.

**2) Novel Concept Grounding**: Tasks with abstract terms like "kobar" (see Figure 7(b)), testing the agent's ability to understand and act on novel concepts.

**3) Visual Constraint Satisfaction**: Manipulating objects while adhering to specific constraints not easily segmentable, such as avoiding certain areas (see Figure 7(c)).

**4) Visual Reasoning**: Higher-level reasoning tasks that involve understanding object properties and maintaining state, such as "put the object that was previously at its west ..." (see Figure 7(d)).

## 5.2 Implicit Goal Inference - Real Robots

To evaluate our framework's reasoning abilities and visual context understanding in real-world settings, we designed a set of **Implicit Goal Inference Tasks**, each with four variations, to assess the system's capacity for long-horizon reasoning and context-aware interpretation of high-level instructions (see Figure 8).

We evaluated our method on three real-world tasks:

**1) Fruit Placement**: The robot is asked to place each fruit in a color-matched area across various setups using the same general prompt. This task challenges the system to infer the desired placement and identify and correct any initially misplaced fruits (see Figure 8(a)).

**2) Superhero Companions**: The robot is tasked with placing fruits and snacks based on color similarity, requiring it to identify objects and make suitable matches, even with non-exact color matches, multi-colored objects, and cases where no clear match is available. (see Figure 8(b)).

**3) Fruit Price Ranking**: The robot is tasked with ranking fruits by price. This challenges the system to interpret visual discount information, apply comparative reasoning, and execute precise ranking to correctly order the fruits (see Figure 8(c)).

All tasks require the system to interpret high-level prompts, perform contextual reasoning, and execute multi-step actions to achieve the implicit goal state based on the provided instructions.

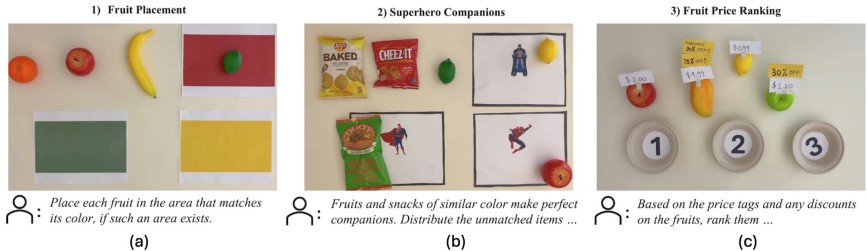

Figure 8: Examples of Ambiguous Instruction & Contextual Reasoning Tasks: (a) Fruit Placement, (b) Superhero Companions, and (c) Fruit Price Ranking.

## 5.3 Spatial Planning - Real Robots

To further challenge our system, we introduced tasks that require precise planning and subgoal management. These tasks test the agent's ability to produce accurate action sequences and handle dependencies carefully (see Figure 9).

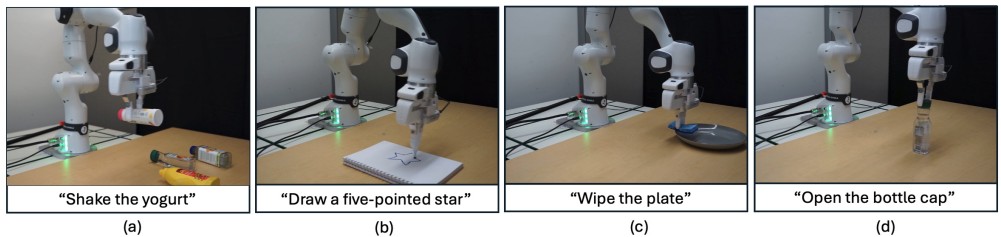

Figure 9: Examples of Complex Planning Tasks.

We evaluated our method on four real-world tasks:

**1) Shaking the Bottle**: The agent grasps a bottle, shakes it in the air, and places it back on the table. (see Figure 9(a)).

**2) Drawing a Five-Pointed Star**: The agent holds a marker and draws a five-pointed star on a notebook. This task demands very precise path planning for both lowering the marker to the paper and accurately tracing the star's points (see Figure 9(b)).

**3) Wiping the Plate with Sponge**: The agent cleans a plate using a sponge. This task involves coordinating the sponge's movement to cover the entire surface of the plate (see Figure 9(c)).

**4) Opening a Bottle Cap**: The agent grasps a bottle and unscrews its cap (see Figure 9(d)).

All four tasks require the robot to generate accurate intermediate subgoals and carefully plan and execute actions within spatial contexts.

## 5.4 Results and Discussion

| Method | Experience | Planning | Prompt Format | Perception |
|---|---|---|---|---|
| Ours | Zero-Shot | VLLM | Multimodal *(text + image)* | Multi-Agent VLLM *(GPT)* |
| Trajectory Generator | Zero-Shot | LLM | Text-Only | VLM *(LangSAM)* |
| MOKA | Zero-Shot | VLLM | Multimodal *(text + image)* | VLM *(GroundingDino+SAM)* |
| NLaP Variants | One-Shot | LLM | Text-Only | Ground Truth State |
| Our Planning + Vanilla GPT | Zero-Shot | VLLM | Multimodal *(text + image)* | Single VLLM *(GPT)* |
| Our Planning + OWL-ViT | VLLM | VLLM | Multimodal *(text + image)* | VLM *(OWL-ViT)* |

(a) Comparison with Baseline Methods. Grey boxes highlight areas of reduced complexity attributable to the framework's inherent design. This reduction should be factored into the interpretation of results. It is important to note that LangSAM is a variant that integrates GroundingDino and SAM, so both the Trajectory Generator and MOKA leverage the same base models for perception tasks.

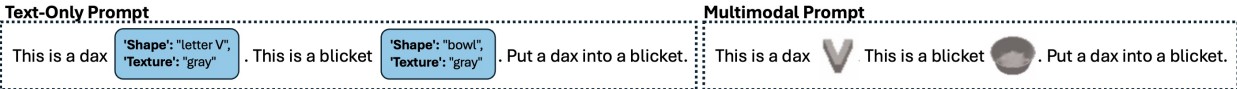

(b) Examples of prompts: text vs. multimodal. Multimodal prompts require visual understanding, making them more challenging than text prompts that rely on ground-truth data.

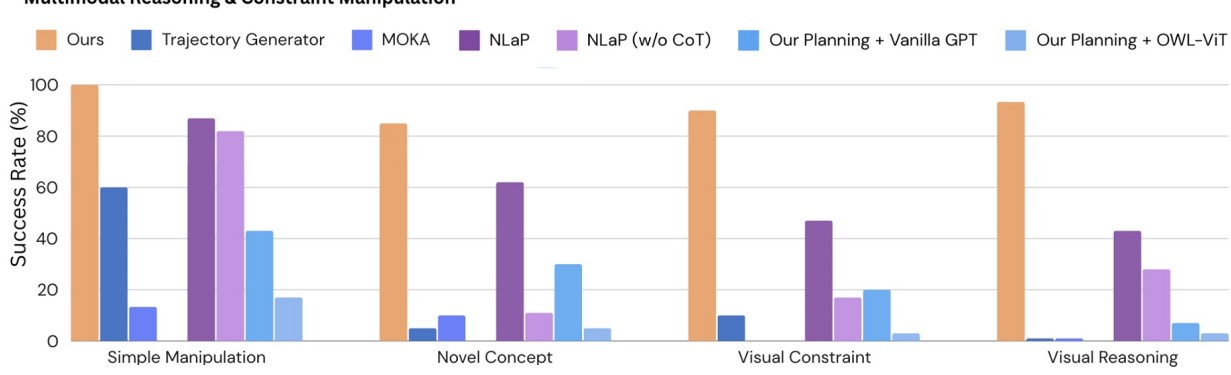

(c) Performance on VIMABench tasks. Wonderful Team achieves strong results across all task domains. Performance declines when the Grounding Team is removed or replaced.

Figure 10: Overall comparison and results on VIMABench tasks.

In VIMABench (Jiang et al., 2023), we compared Wonderful Team against the following methods: (1) Trajectory Generator(Kwon et al., 2024), which uses an LLM for planning and LangSAM for perception; (2) Natural Language as Policies (NLaP) (Mikami et al., 2024), which employs one-shot prompting and directly accesses ground-truth coordinates, bypassing perception; and (3) Ablations Replacing the Grounding Team, where we replace the multi-agent Grounding Team with a single VLLM for inferring object coordinates directly and a separate vision-language model, OWL-ViT.

Table 10(a) outlines each method's characteristics, including zero-shot versus one-shot settings, prompt types, and the modules used for planning and perception. Methods without vision rely on text prompts rather than the more complex multimodal prompts (Figure 10(b)). Notably, NLaP employs one-shot examples in its prompting and directly uses the ground truth state coordinates from the environment, entirely bypassing the perception challenge and, therefore, any comparisons must be made carefully. Due to this lack of perception capability, we can only compare with NLaP in the simulated tasks.

As shown in Figure 10(c), Wonderful Team outperforms baselines across all VIMABench tasks. The Grounding Team and multi-agent structure are crucial; removing or replacing them significantly reduces performance. Methods like Trajectory Generator and our ablation with a separate VLM struggle to detect uncommon objects and lack nuanced reasoning for detection and manipulation. Even with perfect localization (as in NLaP), complex long-horizon planning remains challenging without the multi-agent structure, leading to misinterpretations and errors (Appendix G.1). Ablation studies (Appendix E) **confirm the importance of each component in Wonderful Team**.

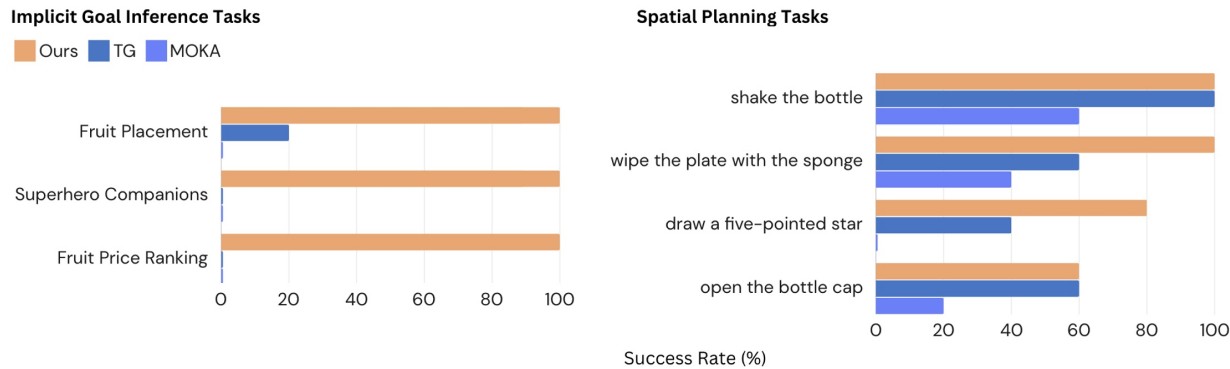

Figure 11: Success rates of *Wonderful Team* (Ours), *Trajectory Generator* (TG), and *MOKA* on real-world robotic tasks, categorized into Implicit Goal Inference Tasks (left) and Spatial Planning Tasks (right). WT achieves perfect success in Implicit Goal Inference Tasks by effectively resolving ambiguous instructions, while TG and MOKA struggle, particularly with conceptual inference. In Spatial Planning Tasks, WT outperforms others, leveraging precise function creation and the verification agent to refine execution. However, all methods face challenges with fine-grained depth estimation (e.g., *opening the bottle cap*).

**Implicit Goal Inference Tasks** In real robot tasks with more general instructions (e.g., placing fruits based on color), as shown in Figure 11, Wonderful Team achieved a 100% success rate, while Trajectory Generator significantly struggled due to its separation of reasoning and vision. Trajectory Generator relies on an LLM to extract information from the text prompt, which requires explicit instructions. When multiple objects from the same category (e.g., various fruits) were present without specific identifiers, it failed to distinguish between them. Using only "fruit" as the identifier for LangSAM, the system could extract the coordinates of all fruits but could not proceed without knowing each fruit's identity and color. Since the LLM lacks grounding knowledge and only has access to these coordinates, it fails to perform meaningful reasoning, resulting in ineffective planning and ultimately causing the low success rate.

**Spatial Planning Tasks** In real robot spatial planning tasks (e.g., drawing a star), as illustrated in Figure 11, Wonderful Team performed comparably or slightly better, benefiting from the Verification Agent ensuring trajectories were within correct spatial boundaries. The Verification Agent checked the planned paths against workspace constraints (e.g., notebook to draw the star on). Both methods exhibited similar failure modes, often due to depth camera sensor inaccuracies affecting tasks requiring height precision (e.g., particularly problematic for opening a bottle cap). These inaccuracies led to errors in estimating the z-axis position, highlighting areas for future improvement in sensor integration and error correction.

# 6 Further Discussions

## 6.1 Comparison with Training-Based Methods

In recent years, the machine learning community has often seen new LLMs exceed the performance of previous-generation fine-tuned models in zero-shot settings, despite the latter's advantage of task-specific tuning. To explore this trend in the context of visual LLMs and robotics, we compare Wonderful Team with several methods that were at least partially fine-tuned on high-level planning robotics tasks.

In particular, we compare against: 1) VIMA Jiang et al. (2023) and 2) Instruct2Act Huang et al. (2023a). In Table 1, we consistently see that the advantage of fine-tuning is outweighed by having a more powerful VLLM.

|  | Ours | VIMA-200M (L3) | Instruct2Act |
|---|---|---|---|
| **Visual Reasoning** | Zero-Shot | Domain Fine-Tuned Mask R-CNN | Pre- and Post-Processing |
| **Task Execution** | Zero-Shot | BC Offline Learning | Pre-defined API + One-Shot |
| **Success Rate (%)** | 91.25 | 88.71 | 79.67 |

Table 1: Comparison with non-zero-shot Methods on VIMABench Tasks. Success rates are averaged across the same tasks considered in figure 10(c)

In addition, we also compare with fine-tuned models RT-1 (Brohan et al., 2022) and OpenVLA (Kim et al., 2024) on a selected subset of the tasks. For a detailed discussion and results of these comparisons, see Appendix G.5.

## 6.2 Limitations: Where does Wonderful Team struggle?

**Limited 3D Reasoning and Partial Observability:** While the integration of depth cameras allows Wonderful Team to capture 3D data, its reasoning and planning are still largely confined to 2D space. This limitation hinders tasks that require precise manipulation along the height axis or a full understanding of 3D spatial relationships. Additionally, it struggles with partial observability, often leading to incorrect interpretations of spatial relationships.

**Real-Time Adaptation and Error Recovery:** Although the Replanning Agent is designed to address failures post-execution, the framework could be improved with real-time dynamic error detection to catch issues immediately. However, reprocessing parts of or the entire task can be computationally expensive and sometimes impractical, requiring careful system design. This limitation is particularly important in navigation tasks or rapidly changing dynamic environments, where constant replanning can be costly and reduce applicability. Improving the system's robustness to environmental variations and enhancing real-time error recovery remain key areas for future work.

**High-Level Task Planning:** We want to stress that we have only considered high-level physical task planning in this paper. Robotics is an exceptionally broad area, and we cannot make claims about the effectiveness of our methods on problems such as visual navigation or low-level kinematic control. We have also considered 3D manipulation environments exclusively, and do not make any claims about the applicability of these methods to other domains.

## Acknowledgments

We would like to thank Professor Tesca Fitzgerald from the Inquisitive Robotics Lab at Yale University and Professor Yen-Ling Kuo from the UVA Learning and Interactive Robotics Lab for their invaluable support and access to equipment that greatly contributed to our real-world robotics experiments.

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

## Appendix Table of Contents

# A    Things are Moving Extremely Fast

While it is readily apparent to everyone that LLM progress has been rapid since 2021, it is perhaps less apparent how rapidly these capabilities are influencing robotics. The initial version of this project, which was started in 2022, was largely dead in the water, because VLLMs at the time struggled greatly to understand their environment. In the past year, VLLMs have improved rapidly, which has allowed them to make substantial progress on robotics environments. To better understand this progress, we took Wonderful Team and changed the language model to earlier VLLMs. The results roughly track the average performance our system has been able to obtain over time.

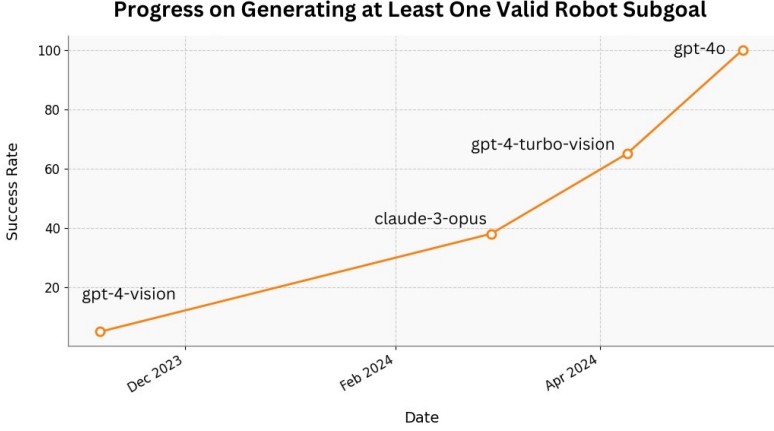

(a) Improvement of VLLMs on robotics tasks over time.

(b) Ability of VLLMs to generate at least one valid subgoal.

Figure 12: Progress of VLLMs in robotics, presenting the success rates evaluated on VIMABench tasks, the same benchmarks used in Figure 10(c), highlighting the impact of each modification.

As we can see, the capabilities of these underlying vision-language models are improving at a blistering pace. Suppose we instead consider a slightly easier problem: the ability of Wonderful Team with VLLMs to generate at least one valid subgoal, which shows the system is working to some extent but perhaps lacks more refined planning ability. In Figure 12(b), we see that here too the improvements have been rapid.

In the Appendix F, we examine the impact of this rapid progress on the grounding team in particular, and show that older VLLMs often struggled to draw bounding boxes with any regularity, suggesting they lacked the fidelity needed for fine-grained robotic control.

# B  Scope and Applications

While many of our tasks focus on table-top pick-and-place scenarios, the design and underlying principles of the Wonderful Team framework extend beyond this domain. This section clarifies the scope of our experiments, showcases the framework's versatility within table-top manipulation, and explores its potential for broader robotic applications.

## B.1  Scope of Experiments

**Tasks Beyond Pick-and-Place.**  In addition to pick-and-place tasks, we evaluated the framework on tasks where success depends on factors beyond accurate object segmentation. These tasks underscore Wonderful Team's versatility:

- **Push Tasks (e.g., Sweep Without Exceeding):** In this VIMABench task, the robot must push objects into a specified target area while avoiding boundary violations. Beyond accurate object localization, this task introduces two distinct additional challenges: (1) reasoning about the correct starting point outside the objects, requiring logical consideration of the opposite side to enable an effective pushing trajectory, and (2) identifying the target, which is an abstract area rather than a concrete object. Wonderful Team achieves a 90% success rate, showcasing its ability to handle spatial reasoning and abstract target identification alongside precise object detection.

- **Spatial Planning Tasks:**
  - **Drawing a Five-Pointed Star:** This task requires planning and executing a complex, fine-grained trajectory with multiple subgoals, going beyond the simpler two-point trajectories typical of pick-and-place operations. The framework achieves an 80% success rate, successfully coordinating intricate movements. Figure 13 shows the planned trajectory sent to the Supervisor Agent for review prior to execution.
  - **Opening a Bottle Cap:** This task involves fine-grained manipulation, requiring the robot to understand the 3D scene (e.g., bottle height), precisely position the arm without disturbing the bottle, and apply controlled rotational forces. Wonderful Team achieved a 60% success rate, demonstrating its ability to handle tasks requiring both spatial reasoning and constrained movements. Failure modes were primarily caused by inaccurate depth estimation from the depth camera, leading to the bottle being knocked over—a scenario that is particularly challenging to recover from.

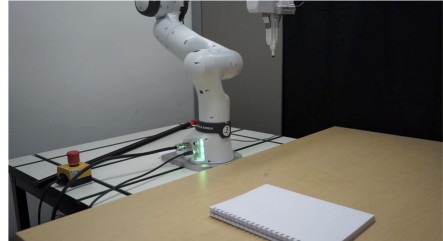 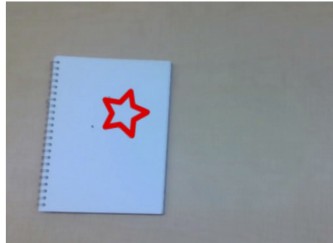 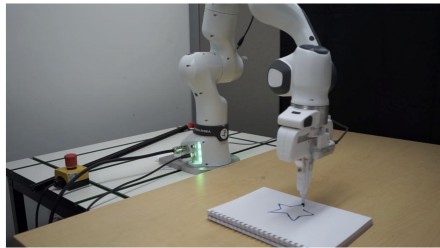

| Prompt: Draw a star | Plan Generated by WT | Execution |

Figure 13: Execution of the drawing task: (Left) Prompted task setup; (Center) Plan generated by Wonderful Team; (Right) Successful execution of the task.

**Summary of Our Scope.**  Our experiments primarily focus on structured table-top environments with third-person view images. Within this controlled setup, the Wonderful Team framework demonstrates effectiveness across a variety of tasks, including pick-and-place, push, trajectory planning, and rotational manipulation. These results highlight the framework's versatility and its capacity to extend to a broader range of robotic applications.

## B.2    Potential Applications Beyond Table-Top Manipulation

Although our research primarily focused on structured table-top scenarios, the modular design of the Wonderful Team framework, built on generalizable models like GPT-4o, demonstrates significant adaptability. With its self-correction mechanism, failure recovery capabilities, and a grounding team to handle spatial reasoning, the framework can be extended to a wide range of tasks with minimal modifications to the prompts and agent components.

**Generalization to Diverse Environments.** Without altering the core framework, Wonderful Team can seamlessly adapt to tasks beyond table-top manipulation. Its ability to perform complex reasoning while extracting accurate positional information from diverse images allows it to handle scenarios involving unstructured and noisy environments.

For example, in the task illustrated in Figure 14, the robot is prompted with: *"Plan a dish using ingredients in the cabinet."* This task challenges the robot to reason through a cluttered cabinet with diverse ingredients, some partially occluded, such as the soy sauce. Despite these challenges, Wonderful Team successfully generates a plan (e.g., "Prepare miso soup using miso paste, chicken stock, garlic powder, and soy sauce"), identifies the necessary ingredients, and accurately localizes their positions. This demonstrates the framework's robustness in complex reasoning and precise localization, even in cluttered and occluded settings.

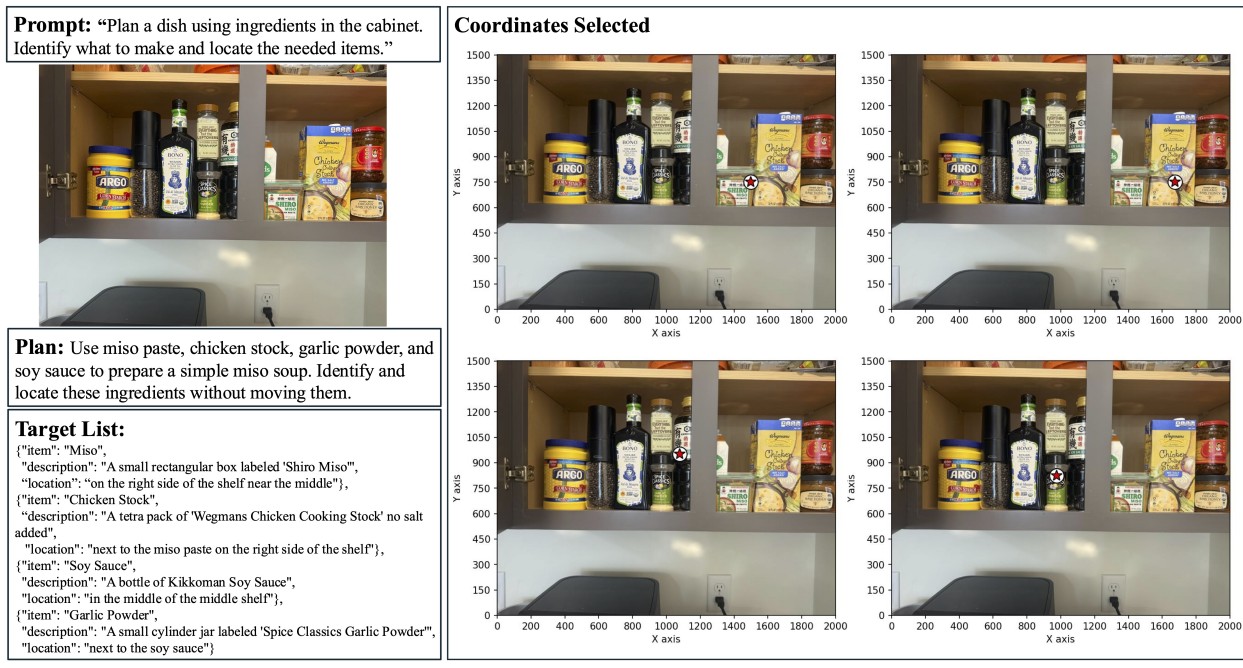

Figure 14: WT results on reasoning and coordinate extraction in an unstructured cabinet task. WT generates a plan based on the given prompt, identifies the required ingredients, and accurately localizes their positions.

A key challenge in extending beyond table-top environments lies in translating 2D coordinates into 3D world actions. While methods like camera pose estimation or depth cameras can be used, many real-world settings lack precise camera pose data, and depth estimation can introduce inaccuracies. These limitations make it difficult to integrate coordinate-based outputs with action APIs. As our focus is on structured table-top scenarios, we did not extensively address these complexities.

Compared to frameworks like MOKA (Fang et al., 2024), which utilize a separate vision segmentation model for object identification, Wonderful Team exhibits significantly improved accuracy and robustness in handling abstract prompts in noisy environments with diverse object appearances and layouts. GroundingDINO

struggles with object identification in noisy environments containing diverse objects with text-heavy labels and varied appearances. As illustrated in Figure 15, it misidentifies corn starch as soy sauce, black pepper as garlic powder, and olive oil as shiro miso, with only the chicken stock correctly identified. Such critical mislabeling severely hampers downstream reasoning and planning tasks, highlighting its limitations in handling complex, cluttered environments.

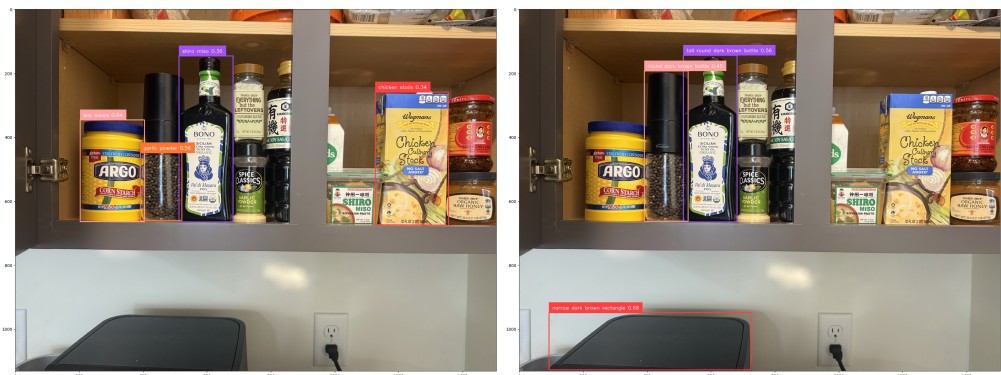

Figure 15: (a) Segmentation Results from GroundingDINO (used in MOKA) using the same target list as in Figure 14. 3 out of 4 items are misidentified. (b) A variation of prompts using simple descriptions for the soy sauce. **Notably, even using an exhaustive list of keywords, it still fails, which makes it almost impossible to recover even with VLLM corrections.**

With Wonderful Team's more robust reasoning and localization capabilities compared to models specifically trained for predefined object detection and grounding tasks, it also demonstrates significant potential for data labeling tasks requiring both reasoning and spatial localization. The high-quality annotations generated by Wonderful Team can potentially serve as valuable training data for future supervised models.

### B.3 Navigation

The cabinet example (Figure 14) demonstrates WT's ability to handle first-person perspectives in cluttered settings, akin to inputs used in navigation tasks. Building on this, we explore its capabilities with a constructed top-down map for navigation tasks, assuming such a map is available, with the following adaptations shown in Figure 16:

- **Obstacle Awareness:** Emphasized detecting and incorporating obstacles.

- **Path Planning Module:** Replaced the original action conversion step with a path planning algorithm to generate collision-free trajectories based on the start location, goal location, and detected obstacles.

- **Prompt:** Slightly adjusted prompts to focus on trajectory planning rather than manipulations.

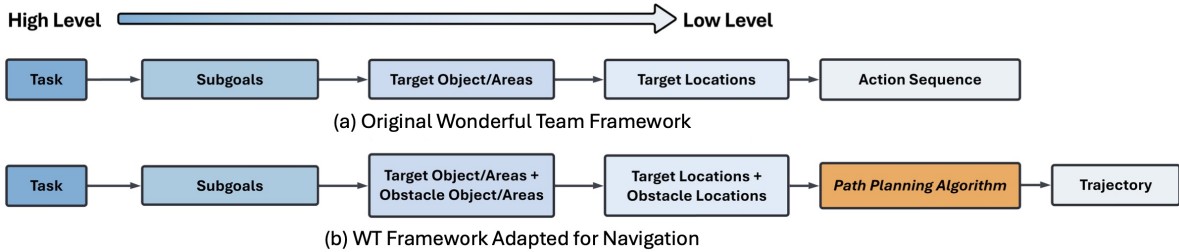

Figure 16: Adapting the Wonderful Team framework for navigation tasks. (a) Original framework for tabletop tasks and (b) adapted framework for navigation, incorporating obstacle and a path planning algorithm.

In our toy navigation setup shown in Figure 17, the system is given the prompt

*"The home robot initial location is marked by the orange icon. Bring the magazine on the coffee table to the kitchen counter."*

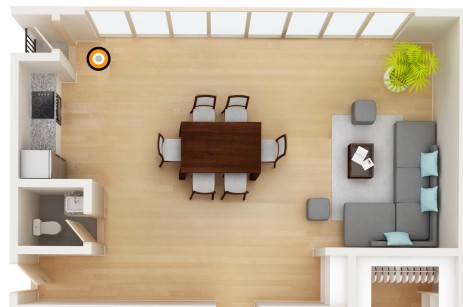

Figure 17: Top-down view of a home robot environment for a navigation task. The orange icon on the top left corner represents the initial robot location.

Figure 18: Visualization of Wonderful Team's results for a navigation task, showcasing identified target locations, obstacle positions, and the planned collision-free trajectory.

Wonderful Team's execution results are shown in Figure 18. The framework first generates a high-level plan consisting of sequential subgoals, such as *"navigate to (x, y)"* or *"place (magazine at (x', y'))"*. These subgoals assume predefined action primitives and are further refined through two critical components:

**Target Location Identification:** Target locations are determined by converting bounding box information into feasible robot target positions. This step is critical, as an incorrect position—such as one inside or to the left side of the kitchen counter's bounding box—would result in an infeasible trajectory. Accurate reasoning about the spatial relationship between the robot's starting position and the target object is essential in the reasoning process.

**Obstacle Location Identification:** Obstacles are identified as bounding boxes that define "DO NOT ENTER" zones. The entire area within each obstacle's bounding box is avoided during path planning to ensure a collision-free trajectory.

Figure 18 illustrates the framework's ability to adapt to a structured home environment by generating obstacle-aware paths. *While this toy navigation experiment demonstrates the feasibility of adapting Wonderful Team for navigation tasks, it represents a simplified setting and does not encompass the complexities of full navigation problems, which are beyond the scope of this work.*

For more complex navigation environments, the Memory Agent could be extended with a hierarchical memory structure, allowing it to store and recall room-based and object-based spatial information. This extension would enable more sophisticated planning in dynamic and unstructured environments.

# C  Prompt and Implementation Details

In this section, we detail the implementation of the Wonderful Team framework, focusing on the prompts used by the agents and their workflows. These prompts demonstrate how each agent fulfill specific roles, enabling the execution of complex tasks in unstructured environments. For reproducibility and further experimentation, the full prompts are available in our  codebase . The Wonderful Team framework introduces a novel multi-agent VLLM-based approach for zero-shot high-level robotic planning, where specialized agents collaborate to decompose complex tasks into manageable components while ensuring cohesive objectives. This section outlines the overall architecture, agent roles, planning pipeline, and coordination mechanisms to provide a comprehensive understanding of the framework.

## C.1  Multi-Agent Architecture

We adopt a multi-agent approach to address long-horizon, multi-step robotics tasks because traditional single-agent frameworks often suffer from scope overload, where a single agent must manage diverse task stages with different focus, leading to reduced effectiveness. By dividing the problem into smaller, specialized sub-tasks, each agent focuses on a specific role, such as planning, grounding, or intermediate validation, ensuring more precise and efficient handling of each step. Each agent also stores their previous input and output to their own internal memory to maintain context-awareness and scope consistency when working on tasks. The agents collaborate by sharing information and verifying intermediate outputs, which minimizes errors and enhances robustness. With this multi-agent approach, we highlight several characteristics:

- **Specialized functionality:** Dividing responsibilities among agents tailored to specific roles ensures efficient task execution, as no single agent is burdened with managing every aspect of the process.

- **Internal memory:** Agents equipped with memory capabilities can make consistent, context-aware decisions, leading to higher-quality outcomes.

- **Different scope of information:** Assigning different scopes to each agent allows them to concentrate on their specific objectives without being distracted by irrelevant details, such as a high-level agent focusing on strategic planning without managing low-level execution and vice versa.

## C.2  Wonderful Team Pipeline

Our framework operates through three primary stages: high-level planning, coordinate-level target location grounding, and low-level action generation. A memory agent collects outputs of other agents across all stages and maintains a system memory of important task-relevant information.

### C.2.1  High-Level Planning

The high-level planning phase involves iterative refinement between the **Supervisor** and **Verification agents**, as outlined in Algorithm 1.

**Input:** Task Prompt (*text, image, or both*), Environment Observation (*image*)
**Output:** Initial Plan, Target Subgoal Object/Area

The Supervisor focuses on high-level objectives and proposes plans with subgoals. The Verification agent evaluates the plan for feasibility and logic, checking for issues such as collision risks, physical constraints, and missing prerequisites. In each iteration, the Verification agent raises specific concerns about subgoals and tracks its feedback and completed checks using internal memory to avoid redundancy. The Supervisor revises the plan based on the Verification agent's feedback. This process repeats until the Verification agent approves the plan, ensuring it meets all constraints.

Table 2: A simplified prompt for **Supervisor** agent proposing high-level plans.

```
You are the supervisor creating and modifying a robotics plan.
Your focus should be on overall task-level objectives, providing guidelines and
direction for other agents.  Low-level grounding and control details are not
within your scope of concern.
Here is the task:  {task prompt}
Here is the environment:  {env}
(If feedback is applicable) Here are the questions raised by the verification
agent on potential issues based on your previous plan.
Please output a high-level plan consisting of a list of subgoals.
```

Table 3: A simplified prompt for **Verification** agent verifying high-level plans.

```
You are a verification agent responsible for checking the feasibility of a
robotics task plan and giving feedback for revision if applicable.
Your sole objective is to ensure the plan itself is logical and achievable.
Here is the current plan:  {plan}
Here is the environment:  {env}
Check for these potential issues:
1.  Collision avoidance
2.  Physical constraints
3.  Missing prerequisite steps
Please output APPROVED or feedback for plan revision.
```

---

**Algorithm 1** High-Level Planning

---

**Given:** Task prompt $T$, Environment observation $E$
**Output:** Verified plan $P$, Target objects/areas list $L$
1: $P \leftarrow$ Supervisor.CreatePlan$(T, E)$
2: **while** not approved **do**
3:      feedback $\leftarrow$ Verification.Verify$(P \mid T, E)$
4:      **if** feedback is approved **then**
5:          **break**
6:      **else**
7:          $P \leftarrow$ Supervisor.Revise$(P \mid$ feedback$, E)$
8:      **end if**
9: **end while**
10: $L \leftarrow$ Supervisor.ExtractTargets$(P)$
11: **return** $P, L$

---

### C.2.2 Coordinate-Level Target Location Grounding

The Grounding Team has three agents: the Manager, the Mover, and the Checker, to identify action points for targets in an environment based on a verified plan. For each target, the Manager observe the entire environment image and initializes a bounding box. To streamline their tasks, the Mover and Checker agents are provided with a zoomed-in image centered on the bounding box rather than the entire environment image, minimizing distractions from irrelevant details. The Checker then evaluates the bounding box image for feedback and approval. If the bounding box is not approved, the Mover revises the bounding box based on feedback, and the process repeats until approval is obtained. Once approved, the Manager determines an action point within the bounding box, and it is added to the final set of action points. Details are shown in Algorithm 2.

Table 4: A simplified prompt for **Grounding Team Manager** agent estimating initial bounding boxes.

```
You are the manager of the grounding team.
With full visibility of the scope, tasks, and subgoals, your task is to guide
two workers who refine your approximate guesses of target locations into precise
bounding boxes using zoomed-in views.  When the two workers have finalized the
box, you choose an action point based on the task and plan, considering context
like pushing in a direction or picking up an object.
Here is the environment:  {env}
Here is the verified plan:  {plan}
Here is the current object/area we are working on:  {target}
Please give an approximate starting point and bounding box.
```

Table 5: A simplified prompt for **Grounding Team Mover** agent adjusting bounding boxes.

```
You are an agent responsible for adjusting bounding box location and dimensions,
ensuring bounding boxes are accurately positioned and sized for precise object
manipulation tasks.
Your adjustments must be meaningful and significant before the final version is
concluded.
Here is the object you are working on:  {object}
Here is the current bounding box:  {bounding box specification}
Here is the object:  {zoomed-in image around current bounding box}
Please propose a revision.
```

Table 6: A simplified prompt for **Grounding Team Checker** agent evaluating bounding boxes.

```
You are responsible for verifying the accuracy and alignment of bounding boxes
for objects of interest.
Collaborating with another agent who proposes revisions, you evaluate whether
the suggested changes improve the current bounding box.  Additionally, you decide
when no more adjustments are needed for a bounding box for final output
Here is the object you are working on:  {object}
Here is the revision:  {bounding box before and after revision}
Please determine if the revision for the bounding box is acceptable; if it is
acceptable, you have to then decide if it no longer needs further revision.
```

Table 7: A simplified prompt for **Grounding Team Manager** agent selecting action points.

```
Your task is to analyze the image and task requirements to identify an
appropriate point of action based on the task's characteristics.  Ensure the
chosen point aligns with the plan objectives and key object properties.
Here is the object:  {zoomed-in environment}
Here is the revised plan:  {plan}
Here is the current object/area we are working on:  {target}
Please give the action point.
```

---

**Algorithm 2** Coordinate-Level Target Location Grounding

---

**Given:** Target objects/areas $L$, Environment $E$, Verified Plan $P$
**Output:** Action points $A$
 1: **for** target $t$ in $L$ **do**
 2:     $bbox_t \leftarrow$ Manager.InitializeBox$(t, E, P)$
 3:     **while** not approved **do**
 4:         feedback $\leftarrow$ Checker.Verify$(bbox_t)$
 5:         **if** feedback is approved **then**
 6:             **break**
 7:         **end if**
 8:         $bbox_t \leftarrow$ Mover.Revise$(bbox_t)$
 9:     **end while**
10:     $a_t \leftarrow$ Manager.DetermineActionPoint$(bbox_t, P)$
11:     $A \leftarrow A \cup \{a_t\}$
12: **end for**
13: **return** $A$

---

### C.2.3 Low-Level Action Generation

In the final stage, the Supervisor agent consults the system memory maintained by the memory agent and converts the verified plan $P$ and action points $A$ into low-level commands. These commands form the executable action sequence, ensuring alignment with the spatial and temporal constraints in the plan and grounding information. The Supervisor is tasked with this step because each agent operates within its own scope and maintains an internal memory of its specific inputs and outputs. Other agents are focused on their specialized roles and responsibilities unrelated to the overall task objectives. In contrast, the Supervisor has been consistently managing the high-level task throughout the process, ensuring continuity and focus. With the finalized grounding information readily available, the Supervisor can efficiently generate the action sequence based on the plan it previously produced, ensuring no critical constraints or details from the system memory are lost.

Table 8: A simplified prompt for **Supervisor** agent generating actions.

```
You have access to a system memory containing all essential task-relevant
information summarized from the multi-agent workflow, along with a top-view image
of the environment.  Your task is to translate this information into a sequence
of actions that align with the plan subgoals.
Here is the memory:  {memory}
Here is the environment:  {env}
Please convert to the final plan.
```

# D    Experimental Details

## D.1    Evaluation Protocol

All experiments were conducted with consistency and rigor to accurately assess our framework's performance.

- **Multimodal Reasoning & Constraint Manipulation**: Each task was executed in 10 runs, allowing only a single attempt per run. An open-loop, single-attempt evaluation protocol was employed to ensure fair comparisons with existing methods and to effectively evaluate the capabilities of the multi-agent framework.

- **Ambiguous Instruction & Contextual Reasoning**: Each task was performed in 2 runs for each of the 4 variations with varying difficulty. For instance, increasing the number of price tags for fruit ranking. An open-loop, single-attempt evaluation protocol was used to consistently measure the system's ability to interpret and execute ambiguous instructions.

- **Spatial Planning & Execution**: Each task was carried out in 5 runs under a closed-loop evaluation protocol, permitting up to three replanning attempts. This method assesses the system's ability to manage complex planning, handle unforeseen challenges, and execute multi-step procedures with precision and coordination.

## D.2    Multimodal Reasoning - Simulated VIMABench

VIMABench features 17 tabletop manipulation tasks, including pick-and-place and push, with various combinations of objects, textures, and initial configurations. It includes 29 objects with 17 RGB colors and 65 image textures, many of which are uncommon in other robotics tasks, making them ideal for testing our approach. We selected VIMABench because it presents a significant variety of objects and textures compared to traditional environments with easily detectable items. This requires advanced scene understanding and careful planning for successful manipulation. VIMABench also includes multimodal prompts with images and textual instructions, creating a complex and realistic testing environment that necessitates reasoning and long-horizon planning.

### D.2.1    Task Details

**Simple Object Manipulation**: Tasks such as "put ⟨object⟩ into ⟨container⟩," where each prompt image corresponds to a single object. These tasks test the basic pick-and-place capabilities of the system.

**Novel Concept Grounding**: Tasks with abstract terms like "fax" and "blicket" paired with images, testing the agent's ability to internalize and act upon newly introduced concepts quickly.

**Visual Constraint Satisfaction**: Tasks that require the robot to perform actions like pushing objects while adhering to specific constraints, such as not exceeding certain boundaries or avoiding designated areas. These tasks test the system's safety and precision in manipulation.

**Visual Reasoning**: Tasks involving higher-level reasoning skills, such as "move all objects with the same textures into ⟨location⟩," and visual memory tasks like "put ⟨object⟩ in ⟨location⟩ and then restore them to their original position." These tasks assess the framework's ability to reason about object properties and maintain state over multiple actions.

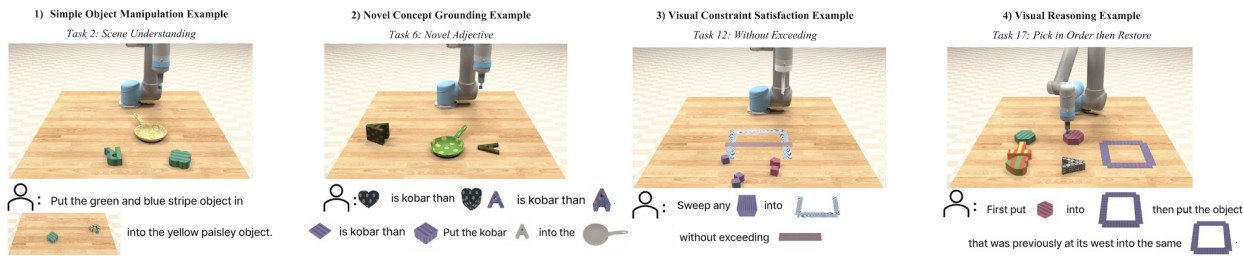

Figure 19: Examples of tasks in VIMAbench Tasks(Jiang et al., 2023).

### D.2.2 Full Experimental Results

In the main paper, we presented results from a selective number of tasks within four categories out of the 17 VIMABench tasks. This was due to the nature of some tasks not being optimal for visual testing. For instance, the twist task requires the robot to determine the precise degree of rotation from before and after images, a challenge without prior training on such tasks.

In Table 9, we present the full experimental results across all 17 tasks of VIMABench. VIMABench defines six main categories of tasks, which are separated in the table by alternating grey and white blocks. From top to bottom, these categories are: Simple Object Manipulation, Visual Goal Reaching, Novel Concept Grounding, One-shot Video Imitation, Visual Constraint Satisfaction, and Visual Reasoning.

Table 9: Success Rates Across All VIMABench Tasks

| Task Num | VIMA 200M | Instruct2Act | NLaP (w/o CoT) | NLaP | TG | MOKA | Ours |
|---|---|---|---|---|---|---|---|
| 1: Visual Manipulation | 99 | 91 | 93 | 100 | 60 | 20 | 100 |
| 2: Scene Understanding | 100 | 81 | 60 | 67 | 40 | 20 | 100 |
| 3: Rotate | 100 | 98 | 93 | 93 | 80 | 0 | 100 |
| *4: Rearrange | 97 | 79 | 52 | 73 | - | 0 | 80 |
| *5: Rearrange then Restore | 54.5 | 72 | 25 | 73 | - | 0 | 70 |
| 6: Novel Adjective | 100 | 82 | 13 | 43 | 10 | 10 | 70 |
| 7: Novel Noun | 99 | 88 | 8 | 80 | 0 | 10 | 100 |
| *8: Novel Adjective and Noun | - | - | - | - | - | 20 | 60 |
| *9: Twist | 17.5 | - | - | - | - | 0 | 50 |
| *10: Follow Motion | - | 35 | 0 | 12 | - | 0 | 10 |
| *11: Follow Order | 90.5 | 72 | 0 | 0 | - | 0 | 0 |
| 12: Without Exceeding | 93 | 68 | 17 | 47 | 10 | 0 | 90 |
| *13: Without Touching | - | 0 | 0 | 3 | - | 0 | 40 |
| *14: Same Texture | - | 80 | 3 | 71 | - | 0 | 100 |
| 15: Same Shape | 97.5 | 78 | 10 | 80 | 0 | 0 | 100 |
| 16: Manipulate Old Neighbor | 46 | 64 | 8 | 20 | 0 | 0 | 90 |
| 17: Pick in Order then Restore | 43.5 | 85 | 10 | 30 | 0 | 0 | 90 |

*Note: Tasks marked with a star (\*) were excluded from Figure 10(c), but their results are included in this table for completeness.*

**Reasons for Task Exclusion in Figure 10(c)**: Tasks marked with a star (*) were excluded from the main paper's results for the following reasons:

**1. Nature of Tasks:** Certain categories, such as Visual Goal Reaching (Tasks 4 and 5) and One-shot Video Imitation (Tasks 10 and 11), were excluded because they are not ideal for evaluating Vision-Language Learning Model (VLLM) capabilities without additional task-specific prompting.

For example, Figure 20 illustrates Task 11 in the One-shot Video Imitation category, where several consecutive frames serve as "goal scenes." Without task-specific prompting or training, it becomes challenging to infer the required actions between frames, as there is no single definitive solution.



Figure 20: Comparison between images without and with ticks for positional reference.

Consider the transition from Frame 1 to Frame 2 in the example above: - One possible approach involves moving the yellow "O" onto the red "O." - Alternatively, another approach might first remove the red "O" and then place the yellow "O" in the same position.

For methods capable of processing multimodal prompts, these tasks require additional tools or workflows, such as detailed explanations of the relationships between consecutive frames (e.g., treating them as a continuous video with specific temporal assumptions). This adds complexity to zero-shot evaluation, making task-specific prompting necessary to infer inter-frame relationships. While task-specific prompting could improve performance, it falls outside the scope of our research. Consequently, evaluations presented in Table 9 were conducted using standardized prompts.

For methods that do not support multimodal inputs, textual prompts for these tasks are often insufficient. They lack the spatial and temporal context necessary to infer relationships, making it difficult to interpret frame-based tasks correctly.

Tasks requiring such inferences include Tasks 4, 5, 9, 10, 11, and 17.

**2. Missing Baseline Results:** Tasks 8, 9, 13, and 14 were excluded due to the absence of baseline results for comparison.

A comprehensive list of tasks, along with video illustrations, is available at  this link .

### D.3   Implicit Goal Inference - Real Robots

### D.3.1   Task Details

As discussed in Section 5, we evaluated our method on three real-world tasks. This section provides more examples of the diverse scenes used for each task.

**Fruit Placement**: The robot is given a random set of fruits and areas of different colors. The prompt is:

> "Place each fruit in the area that matches its color, if such an area exists."

Some scenarios included fruits with no matching color or mismatched colors.

**Superhero Companions**: The robot is provided with fruits and snacks of different colors and three bins designated for different superheroes. The prompt is:

> "Fruits and snacks of similar color make perfect companions. Distribute the unmatched items from the top left corner to the superheroes to help each of them have companion pairs."

**Fruit Price Ranking**: Various fruits with price tags are presented to the robot. The prompt is:

> "Based on the price tags and any discounts on the fruits, rank them from the most expensive to the cheapest and place them in the corresponding bowl."

To further challenge its visual and reasoning skills, we added promotional discounts on top of the original price tags.

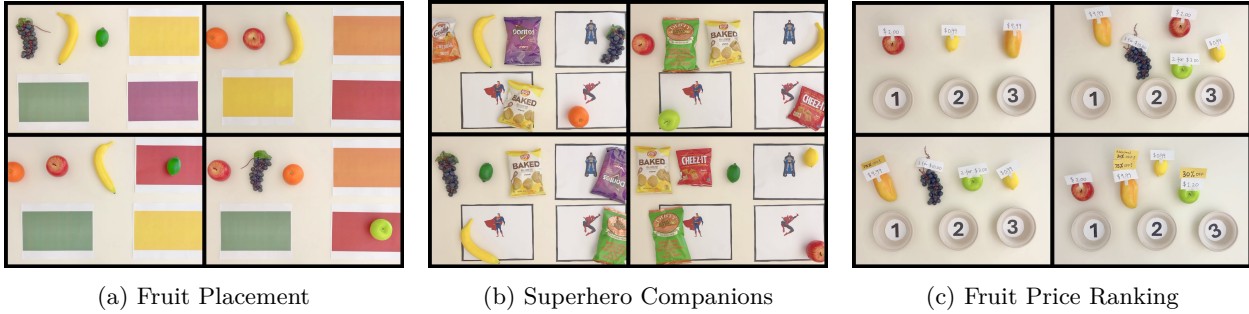

(a) Fruit Placement      (b) Superhero Companions      (c) Fruit Price Ranking

Figure 21: Examples of task environments: (a) Fruit Placement, (b) Superhero Companions, (c) Fruit Price Ranking.

### D.3.2 Robot Setup

For our real-world experiments, we used the UFactory xArm 7, a versatile robotic arm with 7 degrees of freedom, a maximum payload of 3.5 kg, and a reach of 700 mm. It was controlled via the xArm Controller using Python and ROS, allowing seamless integration with our multi-agent system. The robot was equipped with a 2-finger gripper for manipulating various objects. The experiments were conducted on a standard laboratory workbench with predefined task areas, and the robot was calibrated before each experiment to ensure accurate positioning and movement. Our framework mapped the relative displacement of the target position to the robot arm and the pixel coordinates used by the framework, enabling precise picking and placing actions.

For the visual input, we set up a camera directly above the predefined task area, as the robot itself does not come equipped with one. This setup provided a clear and consistent view of the workspace, allowing the VLLM to interpret the environment accurately and plan actions effectively.

### D.3.3 Results

Our real robot experiments demonstrated that our framework successfully completed all three tasks 100% of the time. Note that we **did not modify any of the prompt or pipeline** moving from simulated VIMABench environment to the tasks on the real robot. It was surprising to us how robust the reasoning and planning capabilities of Wonderful Team are. This section provides qualitative results from these experiments, illustrated in Figures 22, 23, and 24. These figures highlight specific aspects of the tasks, illustrating the effectiveness of our framework. It is important to note that these results only reflect the work of the planning team. The role of the grounding team, locating objects and determining their positions, is crucial for the successful execution of these plans.

In the fruit placement task (Figure 22), we present the final execution plan to illustrate the structure of a complete plan. Due to the straightforward nature of the task, this figure does not include the reasoning process. For the superhero companions and fruit price ranking tasks (Figures 23 and 24), we emphasize the reasoning process and omit the block for the complete final plan for the sake of conciseness. The final plans

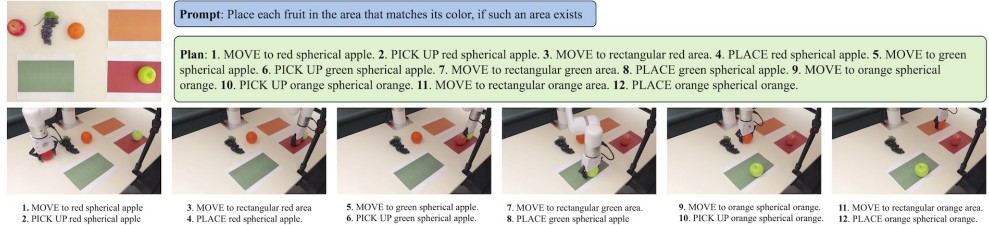

Figure 22: Example Execution on Fruit Placement Task

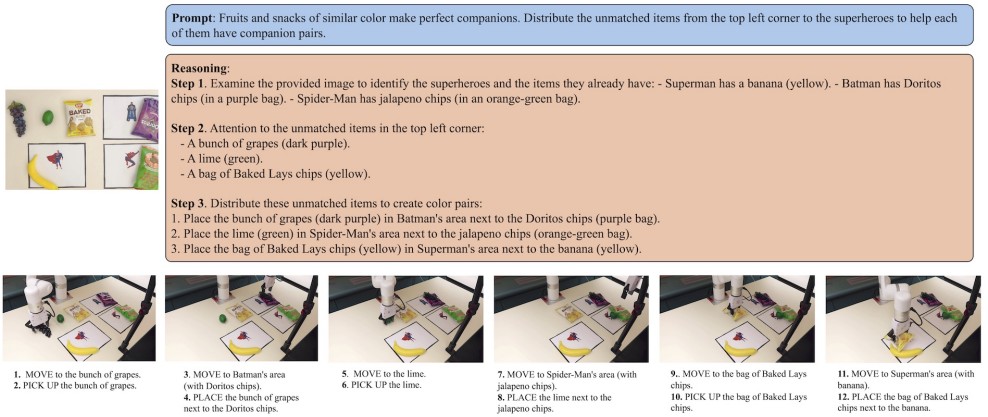

Figure 23: Example Execution on Superhero Companions Task

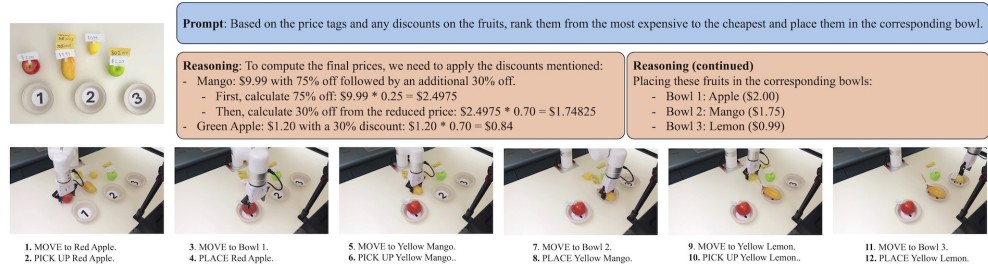

Figure 24: Example Execution on Fruit Price Ranking Task

for these tasks are similar in structure to the fruit placement task, essentially combining the substeps in the execution sequence at the bottom of the figures.

Videos of the experiments and actual execution can be viewed  here .

## D.4 Spatial Planning - Real Robots

### D.4.1 Task Details

This section provides further insight into the spatial planning tasks performed by the Wonderful Team in real-world environments. Each task required precise planning, knowledge of spatial boundaries, and the ability to handle multiple subgoals to complete successfully. Here, we present visual results for each task and discuss the inherent difficulties.

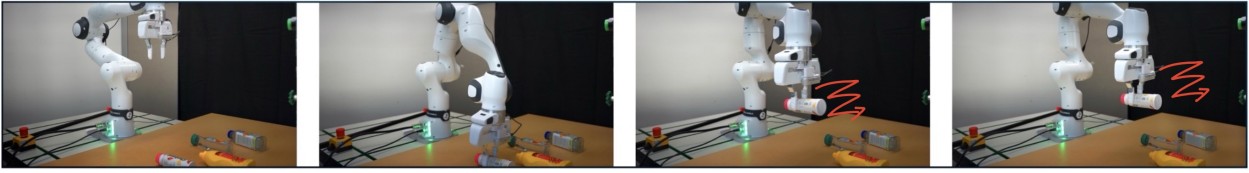

(a) Shaking the Bottle: The task requires the agent to accurately grasp the bottle, perform a shaking motion, and place it back. This involves understanding the correct trajectory for shaking in the 3D space.

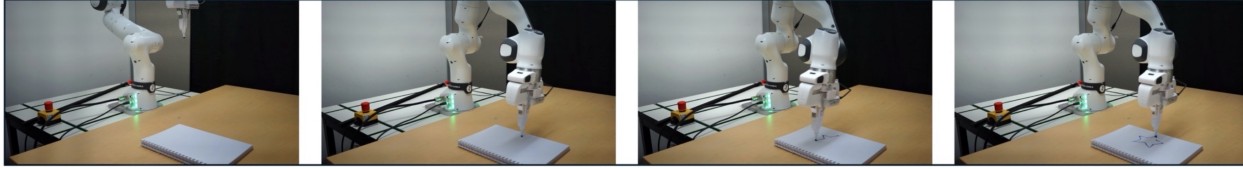

(b) Drawing a Star: The complexity arises from the need to generate the star's points accurately within the frame of the notebook and trace them.

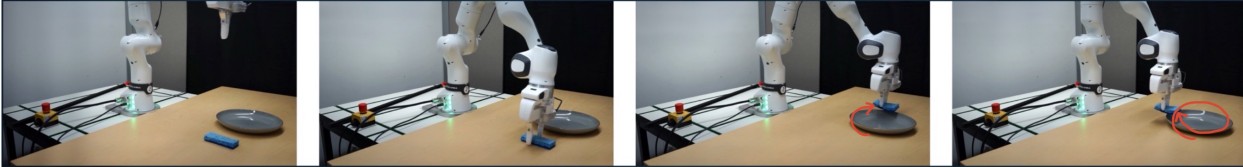

(c) Wiping the Plate: This task involves covering the majority of the surface area of the plate uniformly. It requires planning the path for the sponge to ensure most of the plate is cleaned.

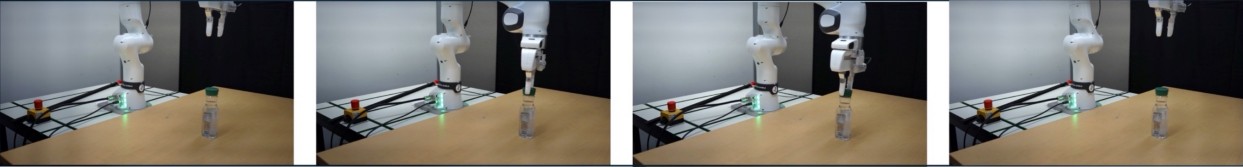

(d) Opening a Bottle Cap: A delicate task that demands precise rotation and grasping control.

Figure 25: Visualization of the spatial planning tasks: (a) Shaking the Bottle, (b) Drawing a Star, (c) Wiping the Plate, (d) Opening a Bottle Cap. Each task requires detailed planning and context-aware decision-making.

These tasks were particularly challenging due to the requirement for the Supervisor agent to have a deep understanding of both spatial and sequential dependencies. For example, the 'Drawing a Star' task required the Supervisor to generate the star's points by writing and calling additional Python functions, ensuring precise path planning for drawing. Similarly, other tasks demanded careful subgoal management and context-aware decision-making to achieve successful outcomes.

### D.4.2 Robot Setup

For our real-world experiments, we used the Franka Emika Panda robot, a 7-degree-of-freedom robotic arm controlled using ROS. We used an Intel RealSense D435 camera positioned above the workspace to extract visual and depth information.

For top-view D-RGB images, the camera was mounted directly above the predefined task area, as the robot itself does not come equipped with an onboard camera. This setup provided a clear and consistent view of the workspace, allowing the VLLM to accurately interpret spatial relationships and plan actions. The depth information was especially valuable for tasks that required accurate height estimation and object manipulation.

# E  Ablation Studies: Are All Parts of Wonderful Team Necessary?

In this section, we present an ablation study to isolate and evaluate the contributions of our proposed hierarchical prompting mechanism relative to the capabilities of GPT-4o itself. The objective is to determine the extent to which the hierarchical prompting enhances system performance beyond what GPT-4o alone can achieve.

We systematically remove or modify various components of our system, such as the Verification Agent and the Box Checking Agent, to observe their individual impacts on performance. This process helps to identify the specific contributions of each component within the hierarchical framework.

The study addresses the following key questions:

- How significant is the hierarchical prompting mechanism in improving system performance compared to GPT-4o alone?

- What are the individual contributions of the agents to the system's accuracy and efficiency?

- How does the removal or modification of these components affect performance metrics?

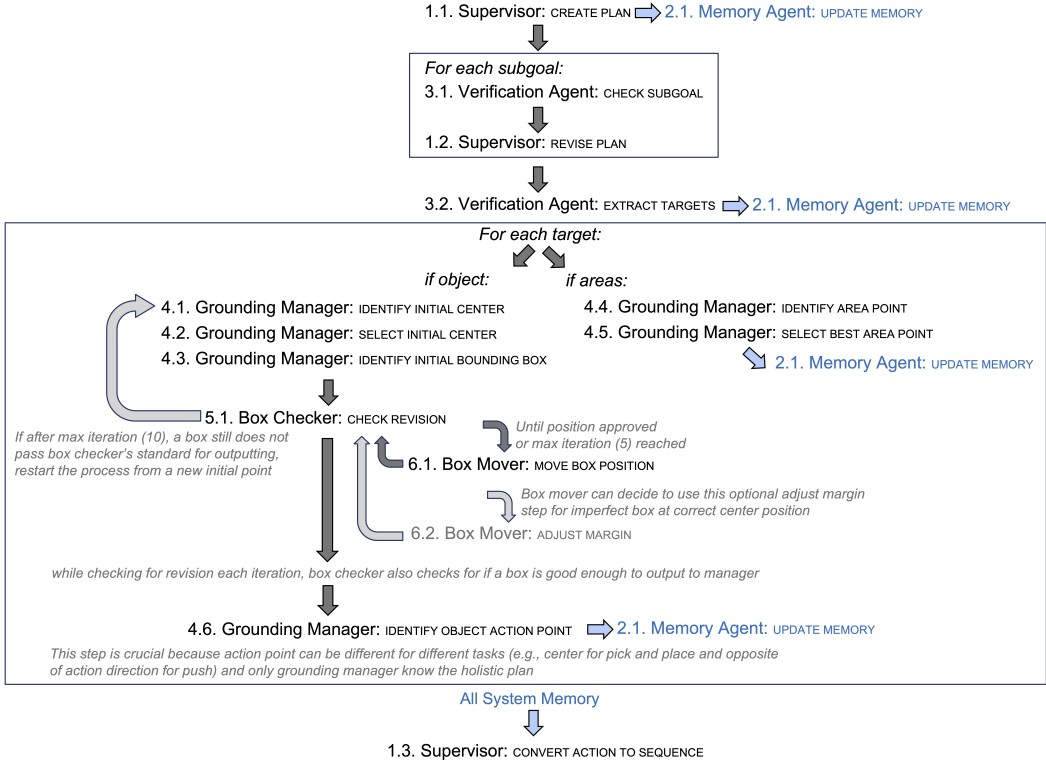

Figure 26: Workflow: Complete

Figure 26 shows the workflow of the complete framework of Wonderful Team. We also provide the full prompt and example input and output corresponding to this workflow chart in Appendix E for more concrete details.

We systematically removed or modified various components of our system, such as the Verification Agent and the Box Checking Agent, to observe their individual impacts on performance. This approach helps identify the specific contributions of each component within the hierarchical framework.

The study addresses the following key questions:

- How significant is the hierarchical prompting mechanism in improving system performance compared to GPT-4o alone?

- What are the individual contributions of the agents to the system's accuracy and efficiency?

- How does the removal or modification of these components affect performance metrics?

Figure 26 shows the workflow of the complete framework of Wonderful Team. Detailed prompts, input examples, and output corresponding to this workflow can be found in Appendix E.

To isolate the effects, we tested the following configurations:

- **1: Removing the Verification Agent:** Without the Verification Agent, the system directly used the supervisor's initial set of subgoals as the final output. This led to errors, as there was no reflection to refine subgoals based on real-time feedback.

- **2: Removing the Box Checking Agent:** The Box Checking Agent evaluates proposed revisions by the Box Mover for improvements and final output quality. When removed, the Box Mover had to perform self-checks, resulting in less accurate outcomes due to the lack of a secondary verification layer.

- **3: Removing Both the Verification and Box Moving Agents:** The system relied solely on the initial bounding box identified by the Grounding Manager, skipping the iterative refinement process and leading to suboptimal action points.

- **4: Removing the Box Checking Agent and Box Moving Agent:** The initial grounding position was used directly without any further verification or adjustments, significantly affecting the robot's ability to select precise action points.

- **5: Removing the Verification Agent, Box Checking Agent, and Box Moving Agent:** The supervisor operated independently, approximating coordinates directly from the image without hierarchical feedback or bounding box identification, resulting in reduced accuracy and adaptability in task execution.

- **6: Removing the Grounding Team:** The supervisor generated plans and extracted targets without identifying bounding boxes, leading to a decline in precision for coordinate-level actions.

- **7: Removing the Verification Agent and Grounding Team:** The supervisor handled all steps, from planning to coordinate generation. Without the Grounding Team, the system relied on rough estimations for actionable points, reducing overall accuracy.

- **8: Removing the Memory Agent:** The Memory Agent selectively stores important information to reduce hallucinations and aid in complex, long-horizon tasks. Its removal had a lesser impact on simpler tasks but proved crucial for maintaining key information in more complex scenarios involving multiple subgoals.

In summary, our settings considered can be summarized in Table 10.

Table 10: Settings Summary

| Setting Number | Supervisor | Verification | (G) Manager | (G) Checker | (G) Mover | Memory |
|---|---|---|---|---|---|---|
| 1 | ✓ | ✗ | ✓ | ✓ | ✓ | ✓ |
| 2 | ✓ | ✓ | ✓ | ✗ | ✓ | ✓ |
| 3 | ✓ | ✗ | ✓ | ✗ | ✓ | ✓ |
| 4 | ✓ | ✓ | ✓ | ✗ | ✗ | ✓ |
| 5 | ✓ | ✗ | ✓ | ✗ | ✗ | ✓ |
| 6 | ✓ | ✓ | ✗ | ✗ | ✗ | ✓ |
| 7 | ✓ | ✗ | ✗ | ✗ | ✗ | ✓ |
| 8 | ✓ | ✓ | ✓ | ✓ | ✓ | ✗ |

Table 11 shows the results of the main tasks from the four primary task suites used in our comparison in Figure 10(c).

Table 11: Success Rates Across Different Settings

| Task Num | Complete | 1 | 2 | 3 | 4 | 5 | 6 | 7 | 8 |
|---|---|---|---|---|---|---|---|---|---|
| 1: Visual Manipulation | 100 | 100 | 80 | 80 | 60 | 50 | 50 | 70 | 100 |
| 2: Scene Understanding | 100 | 70 | 60 | 60 | 60 | 70 | 60 | 20 | 100 |
| 3: Rotate | 100 | 60 | 80 | 60 | 70 | 30 | 40 | 80 | 100 |
| 6: Novel Adjective | 70 | 30 | 20 | 0 | 30 | 0 | 10 | 0 | 50 |
| 7: Novel Noun | 100 | 60 | 80 | 60 | 40 | 20 | 20 | 20 | 70 |
| 12: Without Exceeding | 90 | 10 | 20 | 10 | 0 | 0 | 10 | 10 | 40 |
| 15: Same Shape | 100 | 10 | 10 | 10 | 0 | 0 | 0 | 20 | 60 |
| 16: Manipulate Old Neighbor | 90 | 30 | 40 | 20 | 10 | 0 | 10 | 0 | 50 |
| 17: Pick in Order then Restore | 90 | 0 | 0 | 0 | 0 | 0 | 0 | 0 | 40 |

Generally speaking, tasks with higher task numbers are typically more complex, involving longer horizons and requiring more sophisticated reasoning. The verification and memory agents are particularly beneficial in complex environments with multiple subgoals. Removing them from the framework often results in failure modes such as treating irrelevant distractor objects as task objects or misidentifying arbitrary empty spaces as target locations.

Omitting grounding members tends to lead to less accurate coordinates, which can impact performance. Even for simple tasks without long-horizon planning, the lack of precise grounding can hinder task execution and result in suboptimal outcomes.

Interestingly, the simplest version, where only a supervisor is used, achieved decent success rates on simpler tasks. This could be due to the framework's reduced complexity with fewer components. Simpler tasks usually involve only two or three task objects and locations, making them manageable by the supervisor. There is also a higher probability of guessing an actionable location for larger objects. However, failure modes in this setting include the lack of precise location identification and partially incorrect or infeasible plan. When tasks become more complicated, the absence of corrective processes often leads to failure, especially when hallucination is common.

### E.1 Understanding What Each Part of Wonderful Team Does

Below, we give a summary of this section, summarizing the responsibilities of each team member and how the overall system suffers if we remove them. This shows the relative strength of the multi-agent approach, and how when working together the team members can compliment each other's strengths.

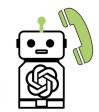

## supervisor

| | |
|---|---|
| RESPONSIBILITY | Receive the initial task, develop a plan for carrying out the task including subgoals. Verify the plan is followed and send the final actions to the robot. |
| PROMPT | You have received a multimodal robotic task description in the form of a combination of text and images, followed by a top-view and a front-view image of the environment. Your task is to interpret this combination of text and images and output a plan with key subgoals.….[more details about environment and specific goals] |
| INPUT | A textual description of the task and an image of the environment. |
| OUTPUT | A subplan of steps that should be followed to achieve a goal. After the subplan is executed, this agent returns the final actions the agent should take. |
| WHAT HAPPENS WITHOUT IT? | If we replace the multi-agent framework with a flat single agent structure, success on all tasks in VimaBench fall dramatically. For simple tasks like Visual manipulation, this fall is from 100% to 70%. For complex tasks like "Pick in Order and Restore" success goes from 90% to 0%. Similar results are seen on the real robot.

The key advantage of the multi-agent framework is that it can self-correct in sub-loops, protecting against hallucination or bad initial estimation. Single agent methods such as NLaP and PIVOT often struggle with precise object manipulation and visual reasoning. |

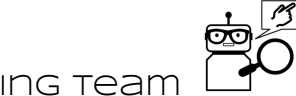

# grounding team

---

| RESPONSIBILITY | Identify the location of objects in the environment. Tell the robot the correct action points (points where it should center its gripper when interacting with objects) |
|---|---|
| PrOMPT | You are an agent that plays a crucial role in a multi-agent robotic system, responsible for accurately identify coordinates of target locations and objects in a robotic environment….[more details about environment and specific goals] |
| INPUT | A high-level plan, a top-view images with x and y axis ticks, and a specific object of interest to identify |
| OUTPUT | Thought process. Final (x, y, z) location of object center points. |
| WHAT HAPPENS WITHOUT IT? | The agent can not corre ctly identify the location of objects in the scene, leading to imprecise actions.

Consequently, on simple visual manipulation, success falls from 100% to 50%.

The grounding team is important because it can iteratively improve upon its estimate of the location of key objects in the environment. Normal VLLM estimates of key points are noisy. But the model is capable of self-correcting initial estimates by looping with the grounding team. This is not possible with a single agent structure. |

---

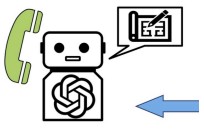

## Memory Agent

| | |
|---|---|
| **Responsibility** | Managing a memory dictionary, which has locations of key objects in the environment, and past plan for object manipulations provided by the supervisor. |
| **Prompt** | You will receive a system memory dictionary, an agent's name, a response from that agent, and a context of this response generated by the agent itself. Your task is to determine if this information is relevant to successful task execution. If so, summarize and update system memory of this information. |
| **Input** | Memory dictionary, output from other agents, context of generated outputs. |
| **Output** | Thought process, Updated memory dictionary with locations of key objects from the prompt. |
| **What Happens Without It?** | Tasks such as "pick in order then restore," rely on memories of previous actions. Without memorizing the order of previous actions, success rates on these tasks fall from 90% to 40%.

In general, the performance on most tasks suffer because the agent struggles to remember where it is in task execution. The supervisor becomes burdened trying to remember this information and suffers from hallucinations. |

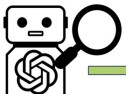

## VERIFICATION AGENT

| | |
|---|---|
| RESPONSIBILITY | Analyze the high-level plan provided by the supervisor, paying attention to potential environmental hazards. Especially consider feasability. Ask informative or clarifying questions. |
| PROMPT | You are an agent that plays a crucial role in a multi-agent robotic system, responsible for verifying a given high-level plans with each subgoal for the successful execution of robotic tasks in a specific environment. [more details about environment] |
| INPUT | High level plan from the supervisor. Image of the environment. |
| OUTPUT | Either a clarification question or concern related to the feasibility of the generated plan, or approval to execute the plan. |
| WHAT HAPPENS WITHOUT IT? | In "Without Exceeding," if there is no Verification Agent then the supervisor often fails to consider where it must stop the sweeping action. The supervisors instructions are also overly ambiguous about how many objects need to be moved, even though this is explicitly in the task command! |
| | If we give the LLM the ability to self-verify with the Verification agent, then success on Without Exceeding increases from 10% to 90% because the agent double checks its ambiguities and corrects them. Similar effects are observed in Scene Understanding and Rotate, where success rises from 70% to 100% and 60% to 100% respectively upon the inclusion of the Verification Agent. |

# F   Ablation Studies: VLLMs' Spatial Reasoning Limitations and Potentials

## F.1   Evaluating VLLM's Spatial Understanding

We aim to answer the question: **How capable are VLLMs at finding accurate actionable position coordinates?**

We set up a toy tabletop environment with various colored and shaped objects placed on a grey table mat, with a single target object (a circle) used to calculate deviation. An example of the environment is shown in Figure 27.

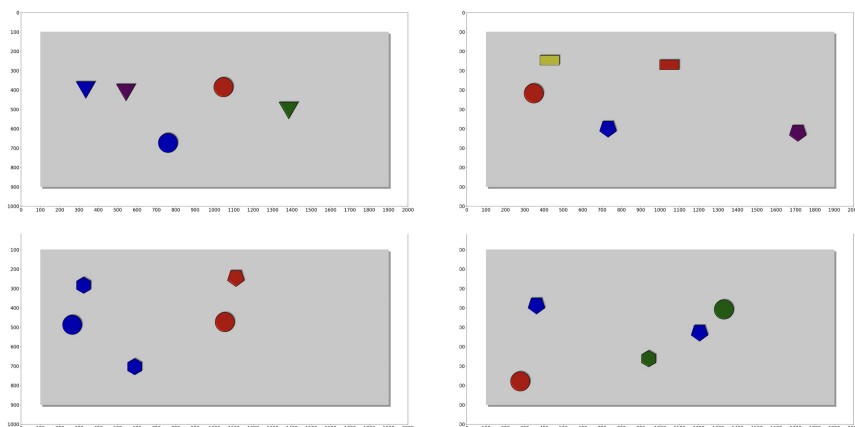

Figure 27: Toy Environment Illustration

We prompt different VLLMs to provide actionable coordinates for the target object, using the overlaid pixel coordinates as a reference. Our goal is to determine whether the coordinates generated by VLLMs are directly usable for action generation and execution.

### F.1.1   Experimental Setup

We tested three state-of-the-art VLLMs:

- **gpt-4o**

- **gpt-4-turbo-vision**

- **claude-3-opus**

Each model was asked to provide the coordinates of the target object based on the given image with pixel coordinates.

### F.1.2   Results

**Are the coordinates directly usable?** Using this simple environment, we want to answer this question we asked earlier concretely. Although actual robotics environments can look much more complicated visually, we can get an idea of the performance of these models. Any point with deviations from the circle center smaller than the circle radius is considered actionable (lies on the circle for picking).

Table 12: Success Rates of Directly Usable Coordinates

| Model | Success Rate (%) |
|---|---|
| gpt-4o | 33 |
| gpt-4-turbo-vision | 5 |
| claude-3-opus | 4 |

We can see from Table 12 that earlier models have a very low success rate. Even with the very strong GPT-4o model, directly using the generated coordinates, even with a perfect plan, can only achieve a 33% success rate, which is far from optimal, not to mention the simple nature of this task.

### F.1.3   Deviation Analysis

**Are the coordinates at least somewhat close to the target objects?**

Although the generated coordinates might not be directly usable for action generation, we wondered if the coordinates are at least informative and close to the target objects for further refinements. In the toy environment, we illustrate the circle of 3 times the radius of the original target circle (the radius of the target circle is always 50 here). This seems to be a good definition of being close in the environment. However, we tried different thresholds to see a fuller picture, as shown in Table 13.

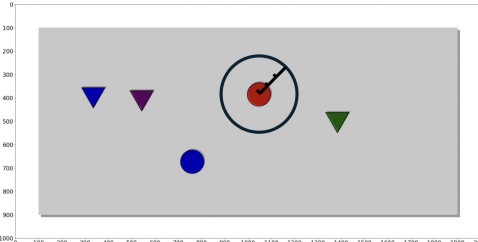

Figure 28: Illustration of the definition of "close to" (3× radius) target objects.

Table 13: Deviation Analysis of Generated Coordinates

| Model | ≤ 3× radius (%) | ≤ 4× radius (%) |
|---|---|---|
| gpt-4o | 89 | 97 |
| gpt-4-turbo-vision | 46 | 68 |
| claude-3-opus | 19 | 58 |

From the table, we can see that although not directly actionable, the proposed coordinates of GPT-4o are of pretty good quality and can be refined with improvements. They are mostly around the target objects, indicating great potential for further refinement and effective use in real-world tasks.

### F.2   Evaluating VLLMs' Error Recognition and Correction

Given that VLLMs have the power to estimate positions, **can we build a framework that can self-improve?** A major component needed here is an agent to check or modify the proposed coordinates. In many robotics tasks, the goal of position finding starts with identifying a bounding box around objects. Suppose we have some proposed bounding box for the object of interest. To further improve upon the initial

version, VLLMs need to know if a bounding box is good enough, or if it is completely wrong and should restart from generating a new one instead of modifying the current one. The question we ask is: **Are the VLLMs capable of visually examining and evaluating proposed coordinates?**

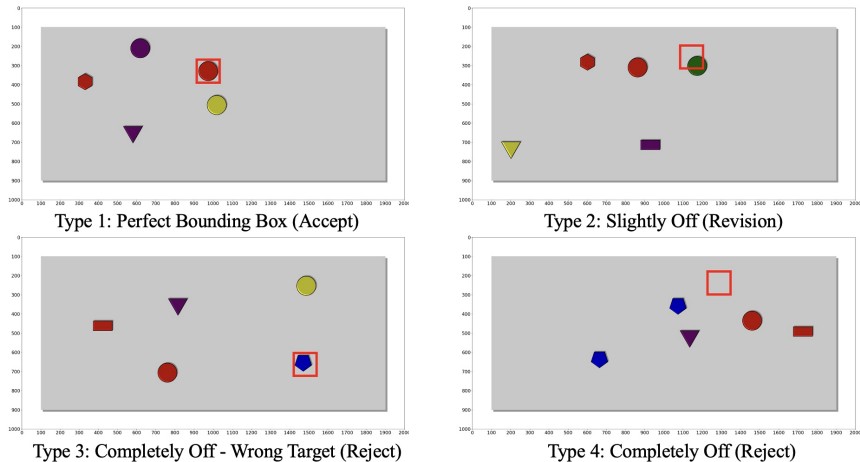

Figure 29: Bounding Box Types: 1) Perfect Bounding Box, 2) Slightly Off, 3) Completely Off - Wrong Target, 4) Completely Off

### F.2.1 Experimental Setup

To test this ability, we randomly generated 4 types of bounding boxes around the circle of interest. Examples are shown in 29. The types are:

**1. Perfect Bounding Box:** The bounding box is correctly placed around the target.

**2. Slightly Off:** The bounding box is close but not perfectly aligned with the target.

**3. Completely Off - Wrong Target:** The bounding box is around a different object.

**4. Completely Off - Around:** The bounding box is sampled around the target (within $4\times$ radius) but is far enough and significantly misplaced, not touching or including the target at all.

Specifically, we give the model a randomly generated bounding box and use the following prompt

"In the given plot, You are tasked with checking if a bounding box should be accepted, accepted with revision, or rejected.
Follow these guidelines to determine whether to accept, advise, or reject the new bounding box:
Criteria:
- **Accept**: If the bounding box covers the target object well without much extra space, pretty much a perfect bounding box
- **Revision Needed**: If the bounding box covers at least a small part of the desired object, but more precision is needed
- **Reject**: If the bounding box is completely irrelevant and does not even touch the desired object
The target object is: [color] circular object.
Your output should be in the following text format. Do not include anything else in your output. This means no reasoning process, no json-like format, no explanation, no other types of texts.
**Output Format:**
Accept Or
Revision Needed
Or
Reject"

### F.2.2 Results

Table 14: Success Rates of Classifying Bounding Boxes

| Model | Success Rate (%) |
|---|---|
| gpt-4o | 97 |
| gpt-4-turbo-vision | 72 |
| claude-3-opus | 33 |

From Table 14 and 15, we can see that GPT-4o demonstrated a very strong ability to examine and decide whether a bounding box is good enough just by visual inspection. This capability opens up new possibilities for self-refinements using current VLLMs. Even in cases where initial coordinate generation is not perfect, incorporating a checker as an additional layer of safety along the pipeline can iteratively improve coordinate accuracy until a satisfactory result is achieved.

| | gpt-4o | | | gpt-4-turbo | | | claude-3-opus | | |
|---|---|---|---|---|---|---|---|---|---|
| Ground Truth | Accept | Revision | Reject | Accept | Revision | Reject | Accept | Revision | Reject |
| Perfect | 25 | 0 | 0 | 18 | 6 | 1 | 22 | 2 | 1 |
| Slightly Off | 1 | 24 | 0 | 0 | 24 | 1 | 19 | 4 | 2 |
| Completely Off - Around | 0 | 2 | 23 | 0 | 11 | 14 | 23 | 0 | 2 |
| Completely Off - Wrong Object | 0 | 0 | 25 | 0 | 9 | 16 | 19 | 1 | 5 |
| Total | **100** | | | **100** | | | **100** | | |

Table 15: Evaluation of Grounding Box Decisions by gpt-4o, gpt-4-turbo, and claude-3-Opus Against Ground Truth Across 100 Examples (4 Ground Truth Classes, 25 Examples Each).

In previous tests with claude-3-opus, the checker often hallucinated during tasks, making it unreliable. For instance, when a bad bounding box is accepted, it not only leads to unsuccessful execution but also confuses the agent itself or other agents in a multi-agent system. This level of complete hallucination is very detrimental. However, in cases where a slightly off bounding box is accepted or a completely off box is sent for revision, it can still be corrected by later parts of the workflow. As shown in Table 15, this level of complete hallucination is predominantly seen in claude-3-opus outputs. In contrast, the strong performance of GPT-4o suggests that a more reliable approach is now feasible.

## G   Comparison with Other Methods

### G.1   Replicating Natural Language as Policies Using GPT-4o

In Section 5, we presented experimental results of the Natural Language as Policies (NLaP) system as reported in the original paper (Mikami et al., 2024). Their implementation utilized GPT-3.5, whereas our method leverages the more advanced GPT-4o. To ensure a fair comparison, this section presents the results of replicating the NLaP system using GPT-4o.

However, since NLaP does not provide their codebase or the full prompt, including images and object information for the one-shot examples used, we attempted to recreate their framework by writing one-shot examples for each task with human-labeled coordinates and object names according to the framework shown in Figure 1 of their paper. For the one-shot prompt, we closely followed and mimicked their provided prompt examples in Table V.

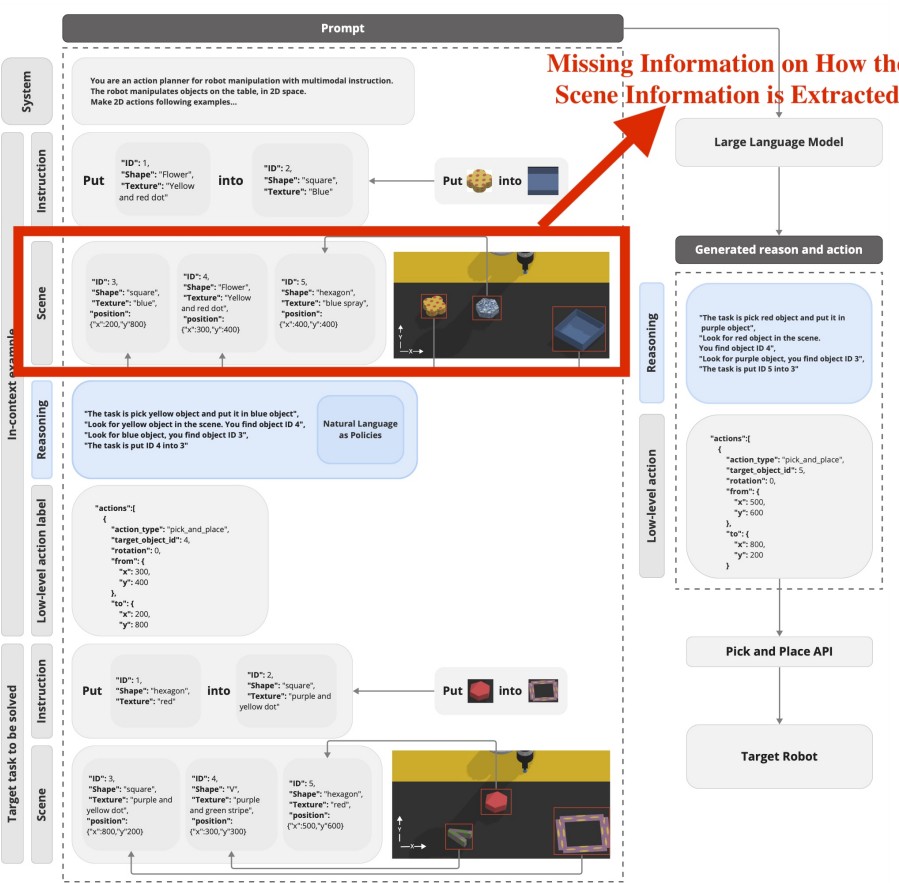

Figure 30: Workflow of Natural Language as Policies by Mikami et al. (2024)

While implementing their framework, we realized that NLaP **does not use the framework to extract coordinate information.** Instead, the extracted coordinates are provided and given to the LLM. The authors did not mention how the coordinates were extracted; the only job of the LLM is to incorporate the coordinates into a detailed final plan. This approach is not a fair comparison to our framework because using the VLLM to extract accurate, actionable coordinates is the more challenging part of this task.

Since the authors did not mention how the coordinates were extracted, and from our previous exploration, using off-the-shelf trained object extraction models such as OWL-ViT did not perform well on VIMABench (Figure 10(c) shows this fact), we assume that NLaP used information as accurate as human-extracted data.

We tried two versions of implementation for this: 1) using GPT-4o to extract this information in the same format, and 2) using ground truth information. For the second approach, we used the ground truth object names from the environment and the ground truth coordinates by mapping the environment state to the pixel coordinate scale. Note that although this approach does not offer a fair comparison to our method, we implemented it to understand how well the planning component performs and to replicate their original results. However, it is important to keep this major difference in mind when interpreting the results.

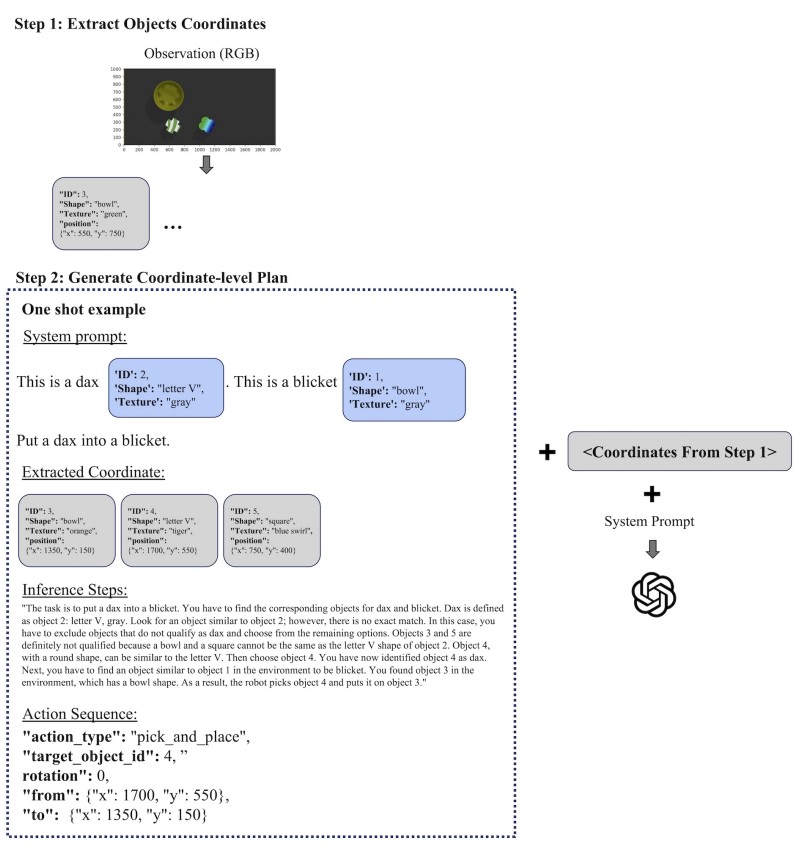

Figure 31: Example - Original Framework of NLaP

Another significant difference between their framework and ours is that the planning component of NLaP does not use any visual information, as shown in Figure 31. In the extraction part, information on objects and their coordinates is derived from visual data, either by human labeling, VLLM, or another model. During the planning phase, the LLM only has access to the textual information. This explains why there wouldn't be a significant difference between using GPT-4o and GPT-3.5-Turbo, as GPT-3.5-Turbo is already very proficient at planning, and the planning part of the framework would not benefit substantially from switching to GPT-4o.

In our implementation of NLaP using GPT-4o for both coordinate extraction and action sequence generation, however, we added the corresponding visual information of both the extracted information and the one-shot example to facilitate the understanding of VLLM of the environment. The idea of our implementation of this added vision version is shown in Figure 32.

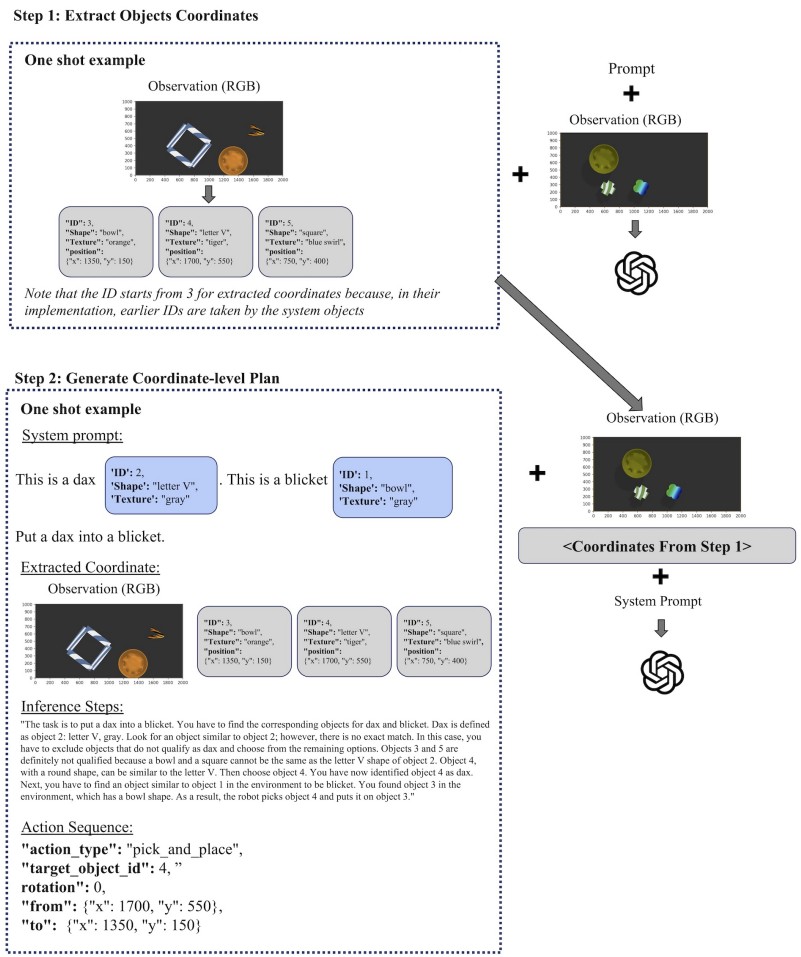

Figure 32: Example - Framework of NLaP with Visual Information Added

Another difference in our experimental evaluation between our method and Natural Language as Policies is that NLaP directly takes the system information of objects for multi-modal prompts. For instance, see an example in Figure 33. In some VIMABench tasks, the prompts can be made multi-modal, and parts of the prompts, usually objects, are not described by words but by images. We used this version of the prompt without any text information for these parts in our evaluation to test the robustness on multi-modal tasks. However, in NLaP, they used the system text information on the shape and texture instead of visual data.

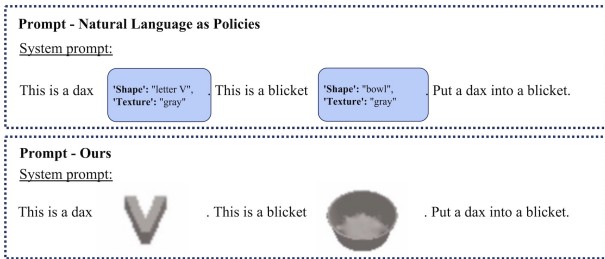

Figure 33: Illustration of the Difference in Multi-modal Prompts: This figure shows the variation in how prompts are constructed between our method and the NLaP system. Our method uses visual information (images) for object description, while NLaP uses system-generated shape and texture information.

One last difference between our methods is that in their prompt, a one-shot example is given. Examples can be viewed in Table V of their paper. The example simply illustrates a typical thought process of a successful execution. They used different examples for different tasks, and during our experiments, we found that sometimes the tasks can be overly similar to the actual task in terms of reasoning, object shape, even object number. For instance, in simpler scenes with two objects, the final desired output is always putting object 3 into object 4 or vice versa. Examples like this may sometimes provide unintended hints that could over-simplify the task.

Table 16: Success Rates Across Different Settings

| Task Num | GPT-4o + GPT-4o | GPT-4o + ground truth | GPT-3.5 + ground truth | NLaP Reported | Ours |
|---|---|---|---|---|---|
| 1: Visual Manipulation | 20 | 100 | 100 | 100 | 100 |
| 3: Rotate | 30 | 100 | 90 | 93 | 100 |
| 6: Novel Adjective | 10 | 80 | 60 | 43 | 70 |
| 7: Novel Noun | 40 | 100 | 80 | 80 | 100 |
| 15: Same Shape | 0 | 10 | 70 | 80 | 100 |
| 16: Manipulate Old Neighbor | 0 | 60 | 20 | 20 | 90 |

In Table 16, we present the results of our ablation studies. We used a '+' sign to denote the combination of settings for planning and coordinate extraction, respectively. For example, 'GPT-4o + GPT-4o' represents the setting where we used GPT-4o to extract scene information (as shown by the red box in Figure 30), while 'GPT-4o + ground truth' means that we directly fed the language model with the actual coordinates and system object names.

From the results, we can see that the comparable version of NLaP, where both planning and grounding are done by the VLLM, barely succeeds on VIMABench tasks, even on simple, one-step tasks. It performs significantly worse compared to our method. The failure modes are often caused by both shortcomings in planning and inaccuracies in the position-finding step. In their original implementation, where coordinate-level information is directly gathered from the environment system instead of by a zero-shot VLLM model, switching from GPT-3.5-Turbo to GPT-4o achieves slightly better results. This improvement is likely due to GPT-4o's enhanced reasoning capabilities, which are beneficial for more complex tasks, such as identifying multiple old neighbors that require reasoning about relationships.

However, since their implementation primarily relies on textual information extracted from the previous steps rather than vision information during the reasoning phase, the gain from switching to GPT-4o, which excels in vision understanding, is limited. As a result, GPT-4o under the NLaP framework still struggles with tasks involving identifying objects of similar shape. A common failure mode is its insistence that no object has a similar shape.

These results further show that **the multi-agent structure is crucial for our system's overall performance.** Even with perfect system output for localization used by Natural Language as Policies, long-horizon planning with complex reasoning remains challenging without the self-corrective multi-agent structure.

### G.2 Comparison with PIVOT

PIVOT (Iterative Visual Prompting Elicits Actionable Knowledge for VLMs) focuses on localization through visual question answering, with minimal emphasis on planning—similar to the role of our grounding team within our hierarchical framework. PIVOT (Nasiriany et al., 2024) introduces an innovative approach to enabling VLMs to localize actionable points or actions by progressively shrinking the action distribution and resampling. The process begins by sampling a set of actions from the action space, which are then mapped onto a 2D image. A VLM is used to select the most promising actions from this set. Based on these selections, a new action distribution is created, and the process is repeated over a fixed number of iterations to refine the actions further.

In their robotic environment implementation, PIVOT handles two versions of localization: one involves finding a multi-dimensional relative Cartesian $(x, y, z)$ coordinate in the action space, and the other involves finding a pixel coordinate in the pixel action space—similar to our approach in VIMABench, where control is based on pixel coordinates rather than relative Cartesian coordinates. For action mapping, PIVOT maps actions to a final endpoint, effectively aligning with the pixel coordinate localization method.

In our comparison, we use VIMABench, where control is based on coordinate-level actions. Therefore, PIVOT's coordinate mapping implementation and the prompts they used on the RAVENS simulator are applied throughout our analysis. There are several similarities and differences between our work and PIVOT that are worth highlighting.

**Similarities:**

- Both frameworks extract coordinate-level information.

- Both operate in a zero-shot manner without any fine-tuning.

- Both annotate 2D images and provide these annotations to the VLLM to guide its decision-making.

**Differences:**

- Our framework focuses on both planning and localization, with localization being one component within a hierarchical structure designed to handle long-horizon tasks with complex planning. In contrast, PIVOT **only focuses on localization**, where their prompts typically describe an object or subgoal rather than addressing a broader task.

- PIVOT uses a **single agent** responsible for iteratively selecting a point from a sample of points or action-mapped points. In contrast, our grounding team consists of **multiple agents**, each playing a distinct role in a self-corrective process.

- PIVOT's method can be viewed as a process of shrinking or guiding the sampling distribution closer to the target object, with each iteration's samples based on the previous one (Fig 34). While our method is also iterative, we begin with a point chosen by the grounding manager and refine it iteratively from there (Fig 35), rather than starting with the entire distribution of possible locations.

- PIVOT identifies a **single action point** for the target object, maintaining this as the goal throughout their iterative process. In contrast, our method offers two distinct workflows that the grounding manager can choose from before localization. When selecting an area point, such as a position between a box and a frame, we also employ point selection. However, for object selection, our method first identifies a center point, then determines a **bounding box** of appropriate size, and iteratively refines this bounding box until it is accurate. The grounding manager then selects an actionable point within the bounded area. We found that this bounding box process greatly enhances robustness and precision, especially for smaller objects or manipulation tasks that require more precise control. We further ablate and discuss this in Appendix G.2.

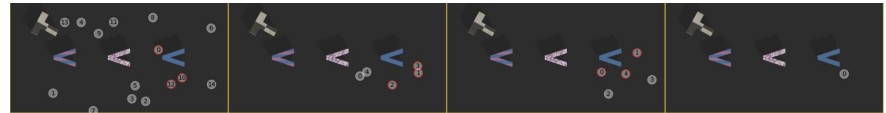

Figure 34: PIVOT Workflow, Blue Letter V

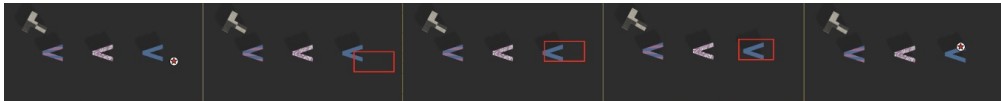

Figure 35: Wonderful Team Workflow, Blue Letter V

Next, we present some quantitative evaluation on object identification results in selected VIMABench environments followed by further discussions on the failure modes.

In Table 17, we compare the experimental results of our method with those from PIVOT. While PIVOT originally utilizes GPT-4V in its framework, we implemented their approach using the more advanced GPT-4o to ensure a fair comparison. Our replication of their framework was carried out to the best of our knowledge to highlight the differences and performance improvements. Additionally, we include results obtained from their official HuggingFace demo to demonstrate the performance of their original implementation. For example output of different grounding approaches, please see 37.

Table 17: Location Grounding Success Rates

| Task | PIVOT (GPT-4V) (HF) (%) | PIVOT (GPT-4o) (%) | GPT-4o Direct Output (w/ labeled axes) (%) | Ours (grounding team) (%) |
|---|---|---|---|---|
| 1. Visual Manipulation | 10 | 30 | 40 | 90 |
| 6. Novel Adj | 0 | 0 | 20 | 80 |
| 17. Pick in Order then Restore | 0 | 0 | 10 | 90 |

**Implementation Details**

*Uniform Sampling:* PIVOT begins by sampling a set of actions from the action space (in VIMABench or RAVENS, as reported in their paper, this involves sampling 2D coordinates), which are then mapped onto a 2D image. A VLM is used to select the most promising actions. Based on these selections, a new action distribution is fitted, and the process is repeated over a fixed number of iterations to refine the actions. Due to the absence of specific details regarding the distribution used in their original implementation, we opted for a uniform sampling strategy. The sampling radius was determined as twice the maximum distance from the average action point to any other point in the set. To ensure alignment with the original method, we also utilized their Hugging Face demo (GPT-4V) to replicate their reported performance.

*Parallel Runs:* The original study also employs a parallel call strategy. To combine results from different runs, they explored two approaches: (1) fitting a new action distribution from the output actions and returning it, and (2) selecting a single best action using a VLM query. In our implementation, we used the second approach with "3 Iterations 3 Parallel" combinations to enhance robustness in our comparison. Additionally, while the original implementation uses the same sampling radius for both width and height, we addressed this by defining separate radii for the shorter and longer edges of the input image.

*Grounding Team Only:* Since PIVOT's framework is primarily comparable to our grounding team, which focuses on processing object descriptions rather than broader tasks, we isolated the grounding component for a direct comparison with their method.

*Success Evaluation:* For evaluation, we conducted 10 runs on different objects from a set of varied initial frames. A task was considered successful if the center point label of each target object had at least half of its area within the object's boundary or if the center point fell within a specific range around the target area center, ensuring successful picking.

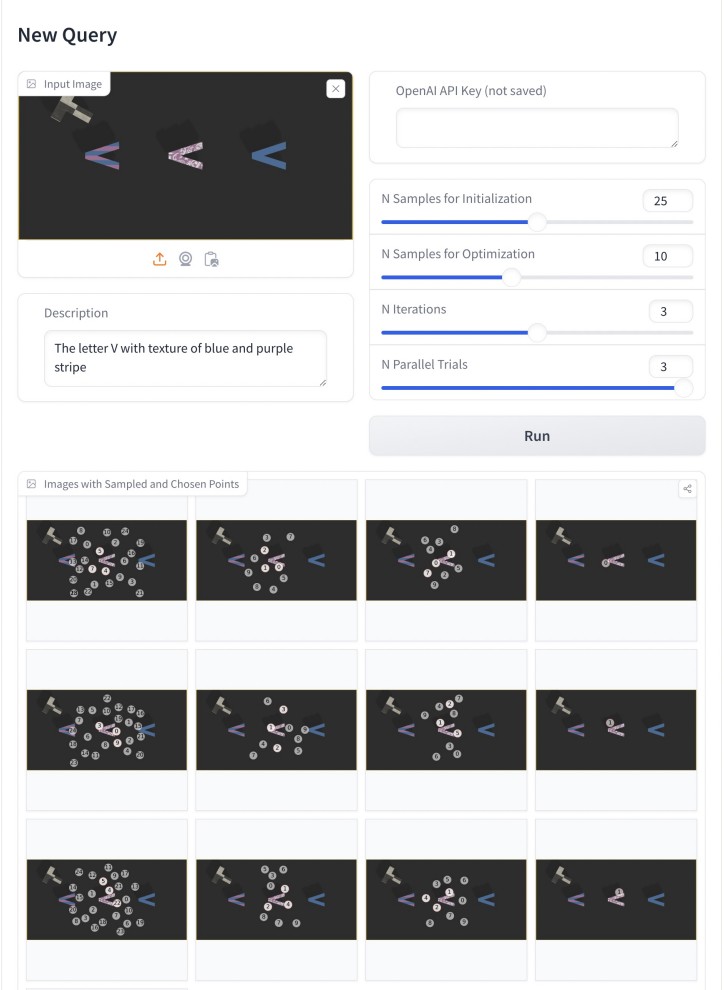

Figure 36: Screenshot of HuggingFace PIVOT Demo

**Failure Mode Discussions**

It's notable that PIVOT's output on tabletop tasks does not over-perform the direct output from GPT-4o. However, this is with the help of the labeled coordinate system, which significantly enhances precision in quantification, as discussed in our motivation section. We further discuss the possible explanations of PIVOT failures:

*Incomplete Sampling Coverage:* In 36, when attempting to select the left object, the initial sampling failed to provide sufficient coverage, with the majority of points being sampled from the center of the image and scattering on the purple paisley letter "V" instead of the target object with blue and purple stripes. As a result, subsequent iterations were confined to a suboptimal region, ultimately leading to poor final results.

*Difficulty in Recovery:* During our implementation, we identified a critical limitation in the sampling strategy: if the sampling radius is too small, it becomes difficult to recover from an inadequate initial selection. Conversely, if the sampling radius is too large, the framework struggles to converge, as the sampled actions may scatter too broadly, reducing the effectiveness of the refinement process.

*Lack of Iterative Continuity:* Another factor that may explain PIVOT's low performance in precise location finding is the lack of continuity between iterations. Although the new set of actions is sampled from a distribution fitted using previously selected promising actions, there is a notable discontinuity in the process. For instance, if a good point is identified during one iteration, it is not guaranteed to be preserved in

subsequent iterations. The framework's fixed number of resampling processes means it cannot exit the process once a good point is found, potentially resulting in the loss of successful actions. This resampling process can lead to promising actions being either diluted or completely discarded in the next round due to inherent randomness, causing inefficiencies and inconsistencies as the framework may fail to build on previous successes.

*Messy Annotations:* Additionally, the framework's annotations can become cluttered, leading to a loss of crucial information from the original image. Unlike our approach, which maintains a clear connection to the original image to preserve full context, PIVOT's method can lose track of the overall scene, making it difficult to refine action points effectively. This loss of context can be particularly detrimental in scenarios where precision and consistency are critical.

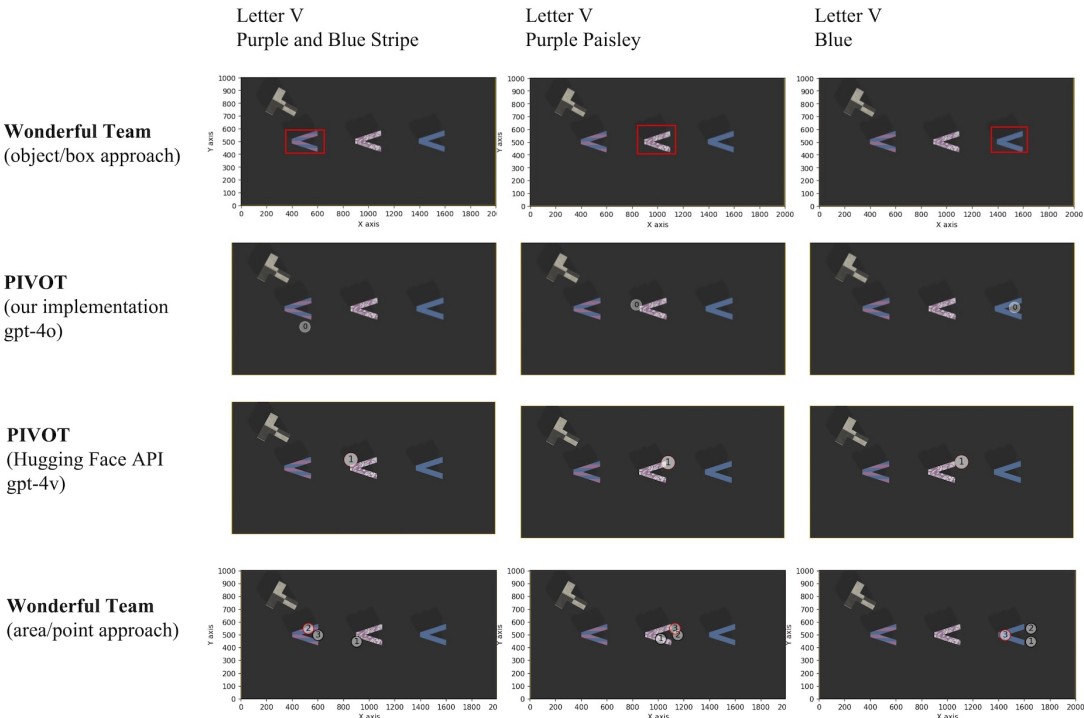

Figure 37: Example Outputs - Wonderful Team vs PIVOT

*Point Selection vs. Bounding Box:* Since the PIVOT method is inherently more similar to our area/point approach discussed earlier—where points are selected throughout the process without the aid of bounding boxes—we further compare PIVOT's outputs with both our bounding box approach and our point approach. Figure 37 provides insight into how these methods perform relative to each other. While both PIVOT and our area/point approach can get reasonably close to the desired objects, they often lack the precision required for tasks involving small objects or when execution demands more accuracy than just proximity to the object.

In Figure 38, we present example executions using the results from these methods. The task involves stacking the purple and blue striped letter "V" on top of the blue letter "V," followed by stacking the purple paisley letter "V" on top. For this execution, we used the PIVOT results from our implementation using GPT-4o, as the HuggingFace outputs were less reliable, with all points concentrated on the same object. The execution screenshots reveal that points not accurately placed on the object lead to failures in picking it up. On the bottom row of Figure 38, even though both points for the first pick-and-place action are technically correct, the misalignment causes the stacking task to partially fail, as the letters "V" are not properly aligned, resulting in an unsuccessful stack.

These results highlight the importance of considering whether a bounding box is needed in the iterative process. With the current level of visual reasoning skills in models, we found that incorporating a bounding box significantly enhances precision, reduces hallucinations, and adds robustness to the execution.

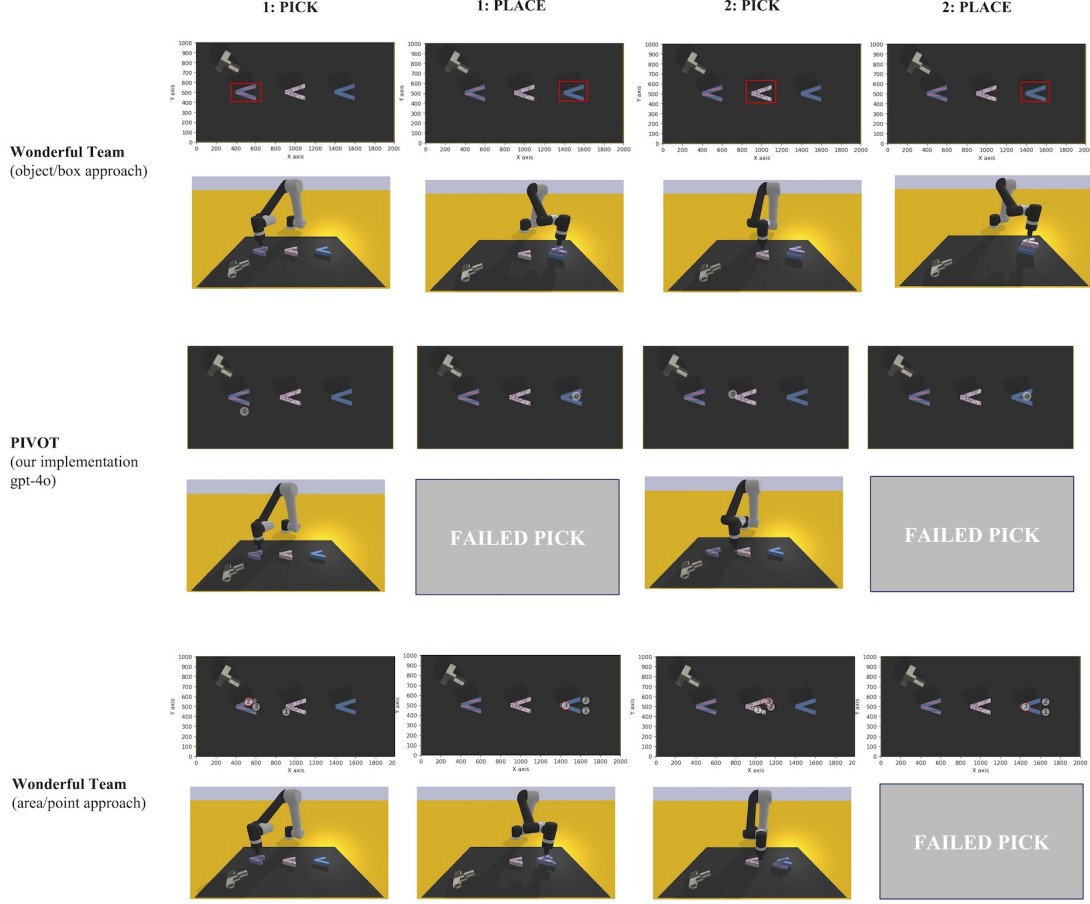

Figure 38: Example Executions - Wonderful Team vs PIVOT

These limitations underscore the shortcomings of the PIVOT framework and highlight the necessity of a more guided and context-aware approach, as implemented in our method.

### G.3 Comparison with Language Models as Zero-Shot Trajectory Generators

#### G.3.1 Key Differences

In Language Models as Zero-Shot Trajectory Generators (Kwon et al., 2024), the task is given to a LLM (GPT-4) in text form. After this, the LLM identifies task-related objects and call an object detection API to retrieve the information about these objects (xyz, height, orientation etc). Using this retrieved information, the LLM starts to plan. In particular, it achieves planning by writing python scripts to generate a trajectory to be executed.

When compared to Wonderful Team, there are a few key differences.

First, the authors employed GPT-4, which does not have vision capability. This means when LLM is making decisions on what objects to detect and generating plans, it does not have any context of the environment except for the one-line command from the user. To improve on the lack of context when making plans, the authors could swap GPT-4 with GPT-4o and provide an image of the environment. This way, the VLLM could identify any task-related objects that are NOT in the command for object detection.

However, even in this case, there are still some issues with the detection process. We experimented with swapping our grounding team with detection models, such as OWL-ViT or langSAM, in the early stage of our research. These methods fail to detect almost all objects that cannot be directly described within a few words. As a concrete example of the problems we encountered with this approach, imagine a user issuing the command: "Pick up the thing to the left of the bottle." Upon reading this command, the detection module will try to find "the thing" and fail, because obviously such an abstract concept can not be encoded into a detection module.

Language Models as Zero-Shot Trajectory Generators uses a single-agent system, where one agent is responsible for generating plans based on user commands. While this method can work under certain conditions, it has inherent limitations, particularly in handling complex, ambiguous instructions and managing long-horizon tasks, especially those that require detailed contextual understanding. In contrast, our system employs a multi-agent architecture, where different agents specialize in specific tasks such as localization, planning, and validation.

#### G.3.2 Single Agent vs Multi-Agent

When comparing the single-agent approach, as exemplified by models like Language Models as Zero-Shot Trajectory Generators, to our multi-agent system, it's important to recognize the distinct challenges each method addresses. Single-agent systems typically solve a more straightforward problem that focuses solely on planning. These systems rely on a separate detection module to identify objects, followed by planning over these detections. While this approach can work in controlled settings, it often leads to instability and misinterpretation of language instructions, particularly when the model encounters more complex or ambiguous commands.

In contrast, our multi-agent system integrates both planning and localization directly within the framework, using Vision-Language Models (VLLMs) to extract object location information. This direct extraction requires a multi-agent setup, where each agent is responsible for a specific aspect of the task, incorporating additional confirmation steps and sub-loops to ensure accuracy. This multi-agent architecture not only addresses the grounding problem but also significantly enhances the system's capability to solve complex, long-horizon tasks, as demonstrated in our evaluations. For instance, in the "manipulate old neighbor" task from VIMABench, even when given ground truth coordinates, a single-agent system using GPT-4o within the NLaP framework often failed to generate successful plans (see Table 16).

#### G.3.3 Benefits of Using a Multi-Agent System

The multi-agent system we propose offers several key advantages over single-agent systems:

**1. Suitability for Robotics Tasks.** A multi-agent system is particularly well-suited for robotics tasks because these tasks typically involve distinct and varied challenges that require different approaches. Unlike

language-only tasks, which may be more uniform, robotics tasks often demand specialized strategies for different components, such as object detection, manipulation, and planning. By employing a multi-agent system, each aspect of the task can be handled by an agent specialized in that area, improving both the efficiency and accuracy of the system. Moreover, the ability of agents to communicate and validate each other's work leads to more reliable decision-making and reduces the likelihood of errors, especially in complex, dynamic environments.

**2. Simplified System Complexity.** At first glance, a multi-agent system might seem more complex than a single-agent approach. However, by dividing the task into smaller, more manageable components, each agent can focus on a specific, well-defined role, which actually simplifies the overall system. This division of labor is especially beneficial in robotics, where different aspects of a task require different strategies. By tailoring each agent's prompts and tasks to their specific role, we avoid the pitfalls of trying to handle everything within a single, monolithic prompt. For instance, when a single agent is responsible for object detection, manipulation, and planning, it often struggles with precise location identification and may produce partially incorrect or infeasible plans.

**3. Effective Communication and Validation.** Communication between agents is another significant advantage of our multi-agent approach. Instead of an agent re-evaluating its own output — potentially leading to unnecessary adjustments or confusion — different agents can validate the outputs independently. This reduces the risk of hallucinations, which can occur when an agent is overly influenced by its previous decisions. For example, when a verification agent (or box checker) evaluates the outputs from the supervisor (or box mover), it treats these outputs as a new query, asking questions like "Is A better than B?" or "Is this action feasible?" This approach contrasts with single-agent systems, where the agent might simply consider whether to fix an existing plan, a situation that often leads to further errors.

**4. Enhanced Self-Correction.** One of the primary strengths of a multi-agent system is its ability to self-correct through agent interaction. In a single-agent system, the same agent must generate a plan and then evaluate it, which can lead to confusion and unnecessary revisions due to hallucinations or biases from previous outputs. In contrast, our multi-agent system allows agents to communicate and validate each other's outputs, significantly reducing the likelihood of such errors. For example, if a VLLM proposes an incorrect object location, this often results in a failed trajectory in 78% of cases. However, when a team of agents iteratively improves the target locations, the success rate increases to 93% (see page 35, Table 11).

**5. Improved Memory Management.** In a multi-agent system, no single agent is burdened with managing the entire context or retaining all information, which can lead to hallucinations or errors. For example, in the "pick in order then restore" task, the success rate was only 40% without a memory module, but it increased to 90% when a dedicated memory agent was included. This demonstrates how distributing responsibilities among agents enhances both performance and reliability by reducing the cognitive load on any single agent.

### G.3.4 Experimental Comparison in Fetch

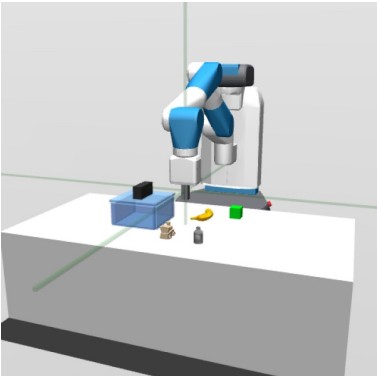

Figure 39: Default View of Fetch Environment with a Box with a Lid

We further compared our methods in a Gymnasium environment involving a box covered by a lid.

**Environment:** The robot used is a 7-DoF Fetch Mobile Manipulator equipped with a two-fingered parallel gripper. The setup includes a closed box with a lid and four other objects placed on the table. See Figure 39 for an example setup.

**Task:** The task is to place one or two of the objects into the box.

**Example Prompt:** "Place the wooden toy train and the rightmost object inside the small blue box with a lid and a black handle." (The exact prompt depends on the target objects.)

**Why This Task is Challenging:**

- It requires accurate 3D estimation. Although this can be partially addressed by using a 2D image with a depth array, there can be challenges when converting 3D information to 2D. Even small deviations in this process can lead to significant errors in execution.

- Items are positioned at different height levels, so collision avoidance must be carefully considered. This is particularly important because the box is quite deep, requiring a thoughtful approach to placing objects inside.

- Correctly identifying the components of the environment, including the box lid, is difficult. The black handle on the lid is very small and requires precise detection for successful execution. Additionally, the handle's common shape and color may cause it to be misidentified or overlooked.

- The plan needs to include the step of removing the lid, which is often omitted. Moreover, the plan should identify an empty area on the table to place the lid without displacing other objects.

**Planning Results:**

In the example task, where the goal is to place the wooden toy train and the rightmost object inside the box, the plan generated by Wonderful Team using the prompt, after validation with the verification agent, is shown in Figure 40(b). For comparison, the plan generated with the exact same task prompt by our system is shown in Figure 40(a). We will further discuss the results in the last section.

**Step-by-Step Trajectory Plan**

**Step 1: Moving to the Wooden Toy Train**

- Move to a position directly above the wooden toy train.
- Lower the gripper to a suitable height.
- Close the gripper to pick up the train.
- Raise the gripper to avoid collisions.

**Step 2: Moving to the Rightmost Object**

- Move above the rightmost object.
- Lower the gripper to a suitable height.
- Close the gripper to pick up the object.
- Raise the gripper to avoid collisions.

**Step 3: Moving to the Small Blue Box**

- Move above the small blue box.
- Lower the gripper to just above the inside of the box.
- Open the gripper to release the first object.
- Raise the gripper slightly and repeat for the second object.
- Open the gripper to release the second object.

(a) Plan Generated by Trajectory Generator

```
output["verified_high_level_plan"]

['Pick up the box lid',
 'Place the lid at an empty spot to the side',
 'Pick up the wooden toy train',
 'Place the toy train inside the box',
 'Pick up the green cube on the right',
 'Place the green cube into the box',
 'Pick the box lid from the side',
 'Place the lid back onto the box']
```

(b) Plan Generated by Wonderful Team

Figure 40: Comparison of Plans Generated by Trajectory Generator and Wonderful Team

**Detection Results:**

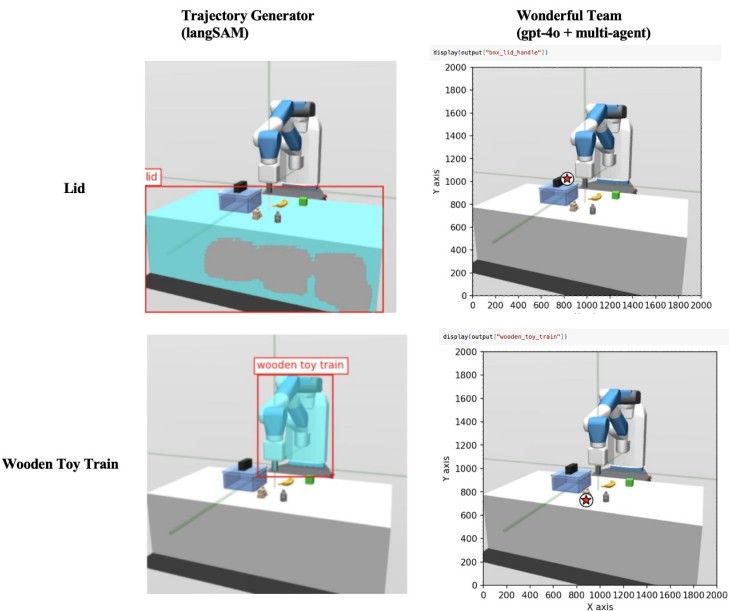

Figure 41: Examples of Object Detection. Check Google Colab notebooks for more example results for Wonderful Team and Trajectory Generator .

**Success Rate Results:**

Table 18: Success Rates on Fetch Box

| Method | Success Rate (%) |
|---|---|
| Wonderful Team (single attempt) | 50 |
| Wonderful Team (re-planning allowed) | 80 |
| Trajectory Generator (single attempt) | 0 |
| Trajectory Generator (re-planning allowed) | 5 |

**Summary of Findings:**

- **Trajectory Generator (Planner):** The planner often fails to understand the implied requirements in the task instruction and is only capable of considering the explicit commands. See Figure 40(a) for an example. Without the command to remove the lid, the planner starts by picking up a target object instead of opening the box to prepare for later steps. In addition to this, the planner also assumes that the gripper can hold two objects at a time before placing them down in the specified container, which is a result of not having access to the environment in context.

- **Trajectory Generator (LangSAM):** This model struggles to correctly identify many objects. See Figure 41 for instance, when asked to find the wooden toy train, it points to the Fetch robot; when asked to locate the lid, it points to the entire table. Similarly, when asked to identify the rightmost object, it again points to the Fetch robot, and when asked to locate the tomato soup can, it points to the mustard bottle.

- **Wonderful Team's Performance:** Wonderful Team achieves a 50% success rate on this task. The main failure mode arises from the difficulty in integrating the depth camera for accurate position estimation, which sometimes results in missed targets.

- **Impact of Replanning Module:** When we introduced a replanning module, Wonderful Team's success rate improved to 80%.

### G.4 Comparison with MOKA

### G.4.1 Key Differences

MOKA (Fang et al., 2024) employs a framework where tasks are given to a VLLM to extract object names, which are then passed to separate vision models (GroundingDINO + SAM) for precise location extraction. These locations are then mapped back to the task plan. While MOKA shares some design principles with Wonderful Team and Trajectory Generator, several critical differences highlight its unique strengths and limitations. Below, we outline the comparisons with both methods:

**1. Vision-Language Integration:**

- Similar to Wonderful Team, MOKA integrates VLLMs (e.g., GPT-4o) for multimodal prompts, whereas Trajectory Generator relies solely on language capabilities without vision integration.

- Like Trajectory Generator, MOKA relies on separate vision models (GroundingDINO + SAM, comparable to LangSAM). As discussed in Appendix G.3, these models struggle with detecting objects that are context-specific, complex, or ambiguously described.

**2. System Architecture:**

- MOKA employs a single-agent system, similar to Trajectory Generator, and lacks the multi-agent architecture of Wonderful Team.

- While MOKA incorporates some chain-of-thought (CoT) reasoning and hierarchical structuring, its simpler prompts and system design lack mechanisms for:

  - Error correction or task failure recovery.
  - The ability to write additional functions or dynamically adjust plans.
  - Divide-and-conquer handling for long-horizon tasks.

**3. Action Point Selection:** Action point selection refers to determining the precise coordinates for interaction after identifying an object or region. For example, pushing an apple from left to right requires selecting a starting point on the left of the apple, not its center. The three methods differ significantly in their approaches:

- **Wonderful Team:** Zooms into bounding boxes with margin space, creating a focused image with annotated pixel coordinates for VLLM input. This ensures accurate action point selection.

- **Trajectory Generator:** Relies on LangSAM's numerical outputs, with the LLM adding or subtracting offsets to approximate coordinates. However, without environmental perception, these offsets are prone to spatial inaccuracies or infeasible placements.

- **MOKA:** Uses annotated images from GroundingDINO + SAM, similar to PIVOT (Appendix G.2). The VLLM selects a point from these annotations, but struggles with scattered or unreadable points for small or ambiguous objects, as illustrated in Figure 42.

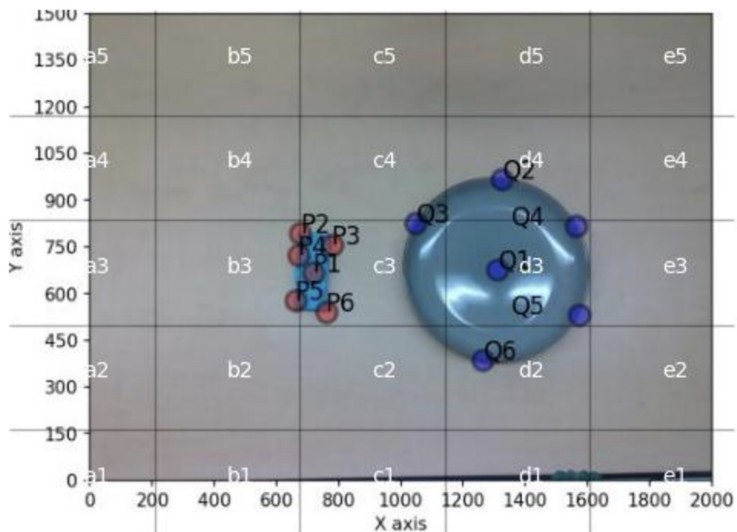

Figure 42: Example of Action Point Selection.

### G.4.2 Error Analysis

The Visual Manipulation Task from VIMABench (Task 1 in Table 9), a straightforward pick-and-place operation requiring minimal reasoning, was selected to isolate and analyze the code execution error rate and object identification performance of each framework. This task simplifies the evaluation by reducing the influence of other dimensions, such as long-horizon planning, complex reasoning, and scene understanding. The results are summarized in Table 19.

Table 19: Success Rates and Error Analysis on Visual Manipulation Task (VIMABench)

| Method | Success Rate (%) | Failure-Complete Plan (%) | Failure-Code Crashes (%) |
|---|---|---|---|
| Wonderful Team | 100 | 0 | 0 |
| Trajectory Generator | 60 | 40 | 0 |
| MOKA | 20 | 40 | 40 |

**Key Observations:** Similar to Trajectory Generator's pipeline—which fails on complex objects and longer-horizon planning and reasoning (details discussed in Appendix G.3)—MOKA also suffers from object detection errors due to the separation of planning and grounding. However, while Trajectory Generator consistently generates complete plans without execution errors, half of MOKA's failure cases result in incomplete plans. These failures are caused by code crashes, which prevent the generation of any plan.

**1. Out-of-Bounds Index Errors:** Out-of-bounds index errors arise from annotated images being difficult for the VLLM to interpret. These issues stem from two primary causes:

- **Overlapping Annotations:** When a single object is mislabeled as multiple objects by GroundingDINO, overlapping points in the annotated image create unreadable data. For example, in Figure 43(a), "container" and "muffin" were both identified by GroundingDINO as the same object, resulting in overlapping points shown in Figure 43(b).

Prompt:  *Put the* 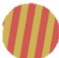 *into the* 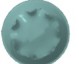 *.*

gpt-4o's output
(target objects):    *[”container”,
”muffin”]*

Grounding DINO + SAM:

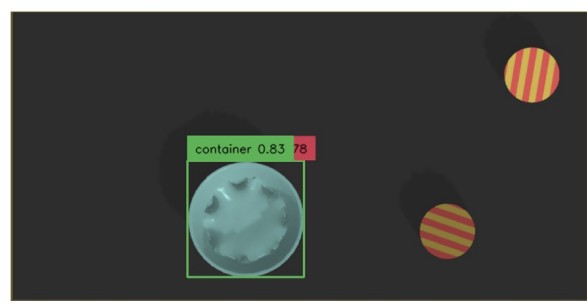 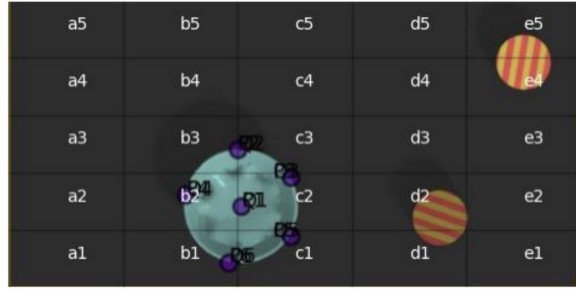

(a) Overlapping Segmentation from GroundingDINO.

(b) Resulting Scattered Annotations for Overlapping Segmentation.

Figure 43: Comparison of segmentation results: (a) Overlapping Segmentation from GroundingDINO and (b) Resulting Scattered Annotations for Overlapping Segmentation.

- **Small Objects Relative to the Environment:** Objects that are too small in the environment also lead to scattered annotations, as shown in Figure 44, making it difficult for the VLLM to select valid action points.

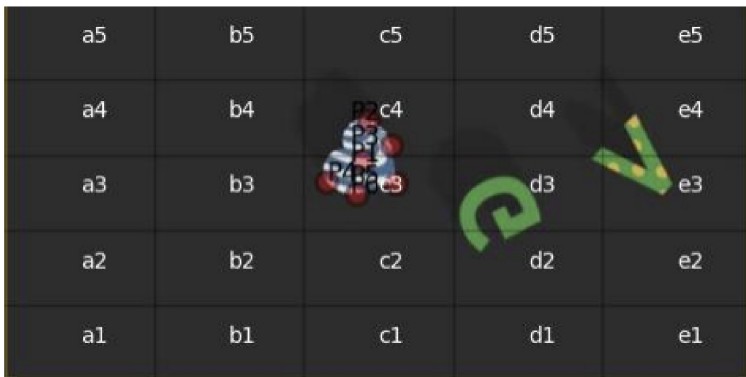

Figure 44: Scattered Annotations for Small Objects.

**2. Object Detection Failures:** Failures in object detection occur when the vision models fail to identify objects listed by the VLLM, causing the pipeline to exit prematurely. These failures arise from two primary causes:

- **Abstract or Vague Descriptions:** Poor descriptions generated by the VLLM (e.g., "striped piece" instead of "blue and yellow striped letter M") often lead to failures in GroundingDINO's object detection, as illustrated in Figure 45.

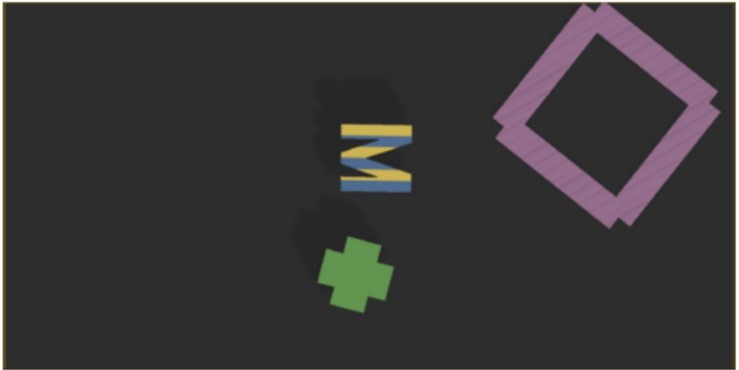

Figure 45: Missed Object Detection Due to Abstract Descriptions.

While this type of failure frequently leads to the pipeline exiting early, it does not always result in an outright Object Detection Failure error. In some cases, the system identifies unintended objects that contribute to the other half of MOKA's failure cases. For example:

– In Figure 46, the command specifies placing the lime in the green area. Without additional contextual details or shape specifications, GroundingDINO misidentifies the lime itself as the green area.

– In Figure 47, the task plan involves placing fruits into bowls labeled 1, 2, and 3. However, GroundingDINO identifies only a single "bowl," failing to distinguish between the intended objects.

These examples highlight scenarios where neither the planning nor the grounding component is entirely at fault. Instead, the failure stems from the fundamental, intrinsic differences between Vision-Language Models (VLLMs) and traditional vision models. VLLMs excel at high-level reasoning and understanding multimodal inputs but rely heavily on contextual cues that vision models cannot inherently provide. On the other hand, vision models, while precise in object detection, lack the broader contextual understanding necessary for complex reasoning. This disparity creates an inherent incompatibility between the two components, leading to downstream errors even when each operates correctly within its own domain.

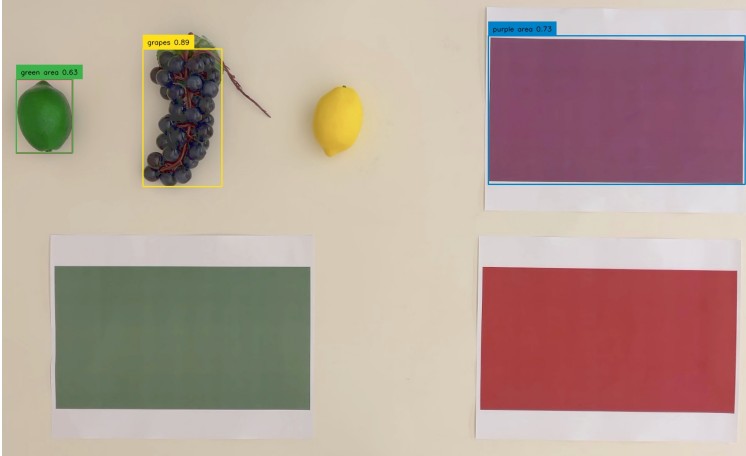

Figure 46: Failure Example: Misidentification of Lime as Green Area.

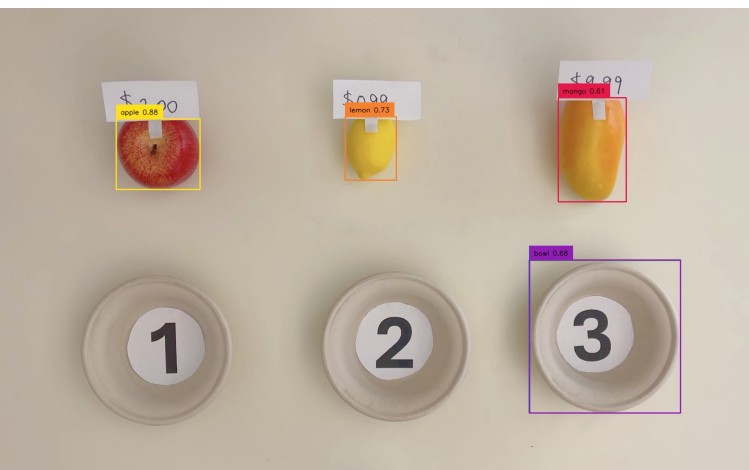

Figure 47: Failure Example: GroundingDINO Identifies Only a Single Bowl.

- **Vision Model Limitations:** GroundingDINO, like other vision models such as OWL-ViT and LangSAM, struggles in complex environments or with less common objects. These limitations, extensively discussed in Appendix G.3, result in incomplete or failed detections that hinder the overall pipeline. For instance, intricate object arrangements, subtle textures, or unusual configurations exacerbate detection challenges, further limiting the model's reliability.

### G.5 Comparison with Fine-Tuned VLA Models

Our main experiments (Section 5) focus on methods that, like ours, employ **pretrained** large language models with vision capabilities (e.g., GPT-4o) without additional fine-tuning on robotics data. In contrast, another branch of work trains **Vision-Language-Action (VLA)** models on large-scale robot-manipulation datasets (see Section 4). Here, we evaluate two open-sourced state-of-the-art VLA systems:

- **RT-1** (Brohan et al., 2022): A variant of Google DeepMind's RT-1 family. RT-1 is a Transformer-based model that processes sequences of images and natural-language instructions to produce real-time robotic actions, fine-tuned on large-scale robotics data.

- **OpenVLA** (Kim et al., 2024): A 7B-parameter vision-language-action model combining LLaMA 2 with a fused visual encoder (DINOv2 + SigLIP features) and an action-detokenizer. Fine-tuned variants exist for WidowX and Google robots from Open X-Embodiment, plus a version trained on the simulated Libero environment (Liu et al., 2023a).

### G.5.1 Experimental Setup

**Environment and Robot Choice.** Directly evaluating fine-tuned VLA models like RT-1 and OpenVLA on the same tasks as our framework is challenging due to their fine-tuned nature and inherent dependencies:

1. **Robot-Specific Action Output:** RT-1 and OpenVLA are designed to generate actions tailored to specific robots (e.g., Google robots or WidowX). Transferring these models to a robot available in our lab would require additional fine-tuning, which itself demands significant data collection and training efforts. This makes a direct comparison infeasible without introducing additional biases.

2. **Sensitivity to Distribution Shifts.** Fine-tuned models like RT-1 and OpenVLA rely on datasets with specific assumptions about backgrounds, object layouts, and robot kinematics. Even small variations, such as changes in camera pose, can cause significant performance drops—up to 50% in both simulation and real-world settings for RT-1, as reported by Li et al. (2024).

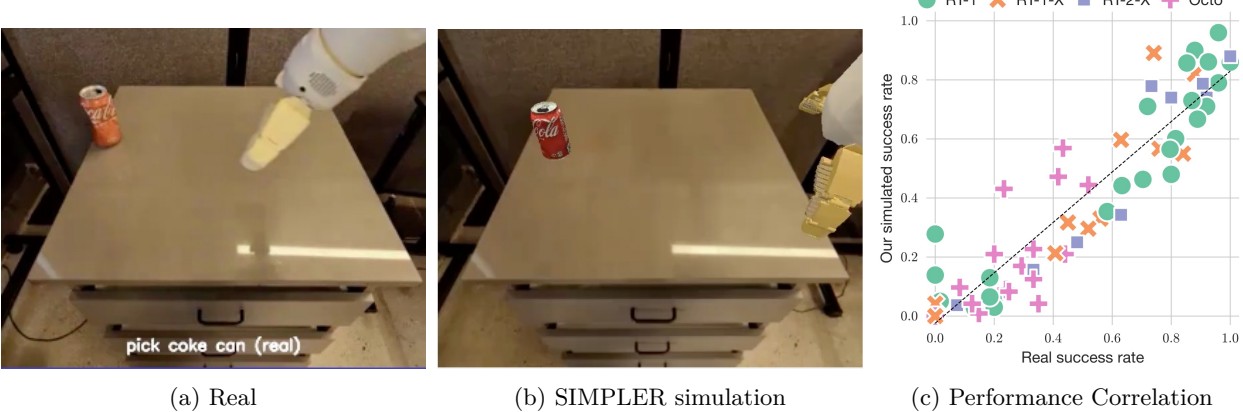

(a) Real  (b) SIMPLER simulation  (c) Performance Correlation

Figure 48: Figures adapted from Li et al. (2024). (a) Real-world setup mimicking Open X Embodiment videos, (b) the corresponding high-fidelity SIMPLER simulation, and (c) strong correlation between SIMPLER and real-world task performance.

To make the comparison as fair as possible, we leverage **SIMPLER** (Li et al., 2024), a real-to-sim framework shown by Li et al. (2024) to correlate strongly with real-world performance (Figure 48(c)). By reconstructing our tasks in SIMPLER Environment, we hope to reduce the potential domain gap for RT-1 and OpenVLA evaluation. Nevertheless, discussion threads (e.g., GitHub issues) indicate that OpenVLA may be more sensitive to real-to-sim transfer than RT-based models, so we also test an OpenVLA variant trained on the Libero environment (Liu et al., 2023a).

**Task Choice.** We do not test all tasks from our suite because:

- **Lack of multimodal prompt support:** Many tasks in VIMABench rely on multimodal inputs, such as image-based prompts that reference objects or actions directly. However, RT-1 and OpenVLA lack the ability to process integrated text-image prompts (e.g., as supported by GPT-4o), accepting only text instructions and environmental frames. This limitation makes certain tasks incompatible with these models.

- **Challenges reconstructing complex tasks in simulated environment:** Fine-grained tasks such as opening a bottle cap or drawing a star require precise simulation physics, which are not fully supported by current physics engines. These tasks would likely introduce additional noise unrelated to the core evaluation of the models.

- **Early failures on simpler tasks.** Preliminary evaluations showed that both RT-1 and OpenVLA consistently failed on basic pick-and-place scenarios, with near 0% success rates in many cases. Given these results, we prioritize simpler tasks to pinpoint *where* and *why* these models fail, without introducing unnecessary complexity.

Rather than performing an exhaustive comparison across all tasks, our goal is to explore three critical aspects of fine-tuned VLA model performance:

1. Can these models understand **implicit prompts**, such as those in our implicit-goal tasks?

2. Are they capable of handling **long-horizon** manipulations involving multiple sequential substeps?

3. How well do they generalize to **uncommon objects or patterns**, such as those encountered in VIMABench but potentially out-of-distribution for their training data?

To address these questions, we specifically test two tasks (and their variations):

1. **Fruit Placement:** From our implicit-goal suite, designed to test reasoning and implicit planning.

2. **Simple Object Manipulation:** A VIMABench-inspired pick-and-place scenario representing the simplest form of long-horizon tasks.

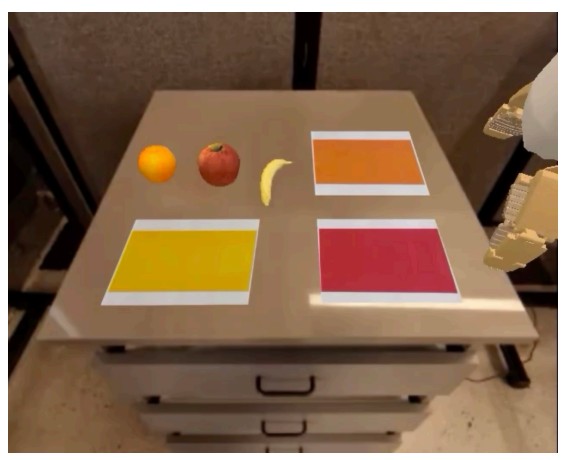

(a) Fruit Placement Setup in SIMPLER

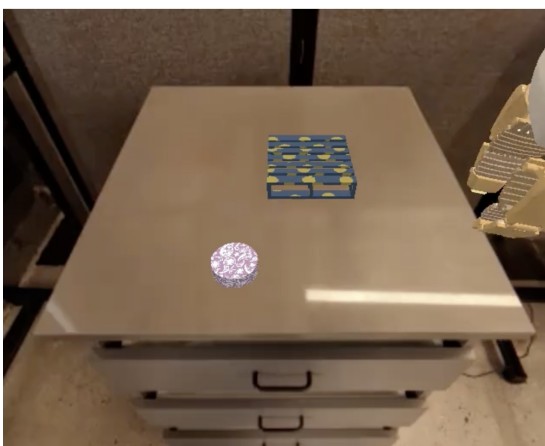

(b) Simpler Object Manipulation Setup in SIMPLER

Figure 49: Evaluation setups for tasks in the SIMPLER environment. (a) Fruit Placement task setup: objects are placed in color-matched regions based on implicit goal inference. (b) Simple Object Manipulation task setup: basic pick-and-place tasks adapted from VIMABench. Both setups replicate real-world configurations with high fidelity, ensuring consistency in object placement, camera angles, and environmental details for fair comparisons across methods.

We provide a Colab notebook[1] for reproducing these setups and evaluating both RT-1 and OpenVLA.

### G.5.2   Experimental Results and Ablations

**Overall Success on Two Tasks.**   Despite efforts to minimize the domain gap (Figures 49(a) and 49(b)), RT-1 and OpenVLA achieve limited success on the evaluated tasks in SIMPLER Environment. Specifically, on the Fruit Placement task and the simplest VIMABench pick-and-place scenario, RT-1 yields only a 10% success rate on the latter, while OpenVLA fails to complete either task, achieving 0% success in both cases. In contrast, our Wonderful Team framework achieves a perfect 100% success rate across both tasks (Table 20).

| Task | Wonderful Team | RT-1 | OpenVLA |
|---|---|---|---|
| Fruit Placement (Implicit) | 100% | 0% | 0% |
| Simple Object Manipulation (VIMABench) | 100% | 10% | 0% |

Table 20: Success rates (%) for Wonderful Team, RT-1, and OpenVLA on two simplified tasks in SIM-PLEREnv. Each experiment was conducted over 10 trials with a maximum of 300 steps per trial.

Below, we further break down **why** these failures occur using ablation experiments.

### G.5.3   Ablation Study: Fruit-Placement Task Variations

The original fruit-placement task, described in Appendix D.3, features a setup with three colored papers on a table, three to four fruits, and the potential for pre-misplaced fruits that require additional reasoning to identify and correctly re-place them. The task prompt is:

*"Place each fruit in the area that matches its color, if such an area exists."*

This setup tests implicit reasoning, such as inferring relationships between objects and spatial reasoning for proper alignment with the colored areas (Figure 21(a))

To analyze the performance of RT-1 and OpenVLA in handling progressively complex scenarios, we conducted an ablation study using the Fruit Placement Task. The setup is illustrated in Figure 50, where complexity is incrementally added by introducing more fruits, colors, and reasoning challenges.

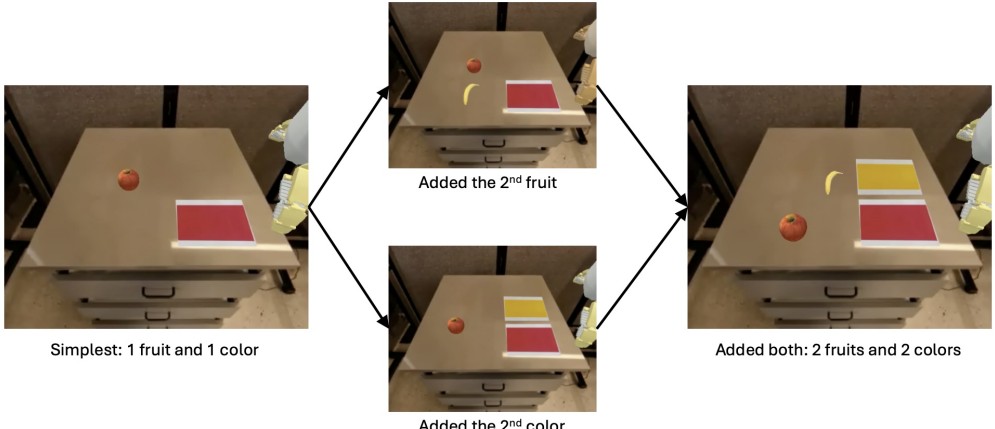

Figure 50: Ablation study setups for the Fruit Placement Task. Starting from the simplest configuration, complexity is progressively added.

---

[1]https://colab.research.google.com/drive/1C9JW4MXLNyNOLmqcrOqNEm7C6ghgfB9Z#scrollTo=H81NNh_jdIgP

Table 21 summarizes the grasping and task completion success rates for RT-1 across various levels of task complexity. OpenVLA consistently achieved a 0% success rate across all configurations, including the simplest setup. A common failure mode observed for OpenVLA was the robot remaining stationary throughout all 300 steps, unable to initiate any actions. This aligns with prior community observations, which highlight OpenVLA's significant sensitivity to distribution shifts.

RT-1 performs well in simpler configurations, successfully grasping the apple and placing it in the red area under minimal scene complexity. However, as the task becomes more challenging, such as introducing distractors or implicit reasoning prompts, RT-1's performance declines significantly.

| Colors | Fruits | Prompt | RT-1 Grasped (%) | RT-1 Success (%) |
|--------|--------|--------|------------------|------------------|
| 1 | 1 | Place the apple in the red area. | 100 | 70 |
| 1 | 2 | Place the apple in the red area. Ignore the banana. | 90 | 40 |
| 1 | 2 | Place the apple in the red area. | 70 | 20 |
| 1 | 2 | Place each fruit in the area that matches its color, if such an area exists. | 10 | 0 |
| 2 | 1 | Place the apple in the red area. | 100 | 30 |
| 2 | 1 | Place each fruit in the area that matches its color, if such an area exists. | 0 | 0 |
| 2 | 2 | Place the apple in the red area. | 70 | 30 |
| 2 | 2 | Place each fruit in the area that matches its color, if such an area exists. | 10 | 0 |

Table 21: Ablation results for Fruit Placement Task variations. RT-1 success rates (%) for grasping the correct object and completing the task are shown as complexity increases. OpenVLA results are omitted as all trials resulted in 0% success.

### G.5.4 Ablation Study: VIMABench Simple Manipulation Task Variations

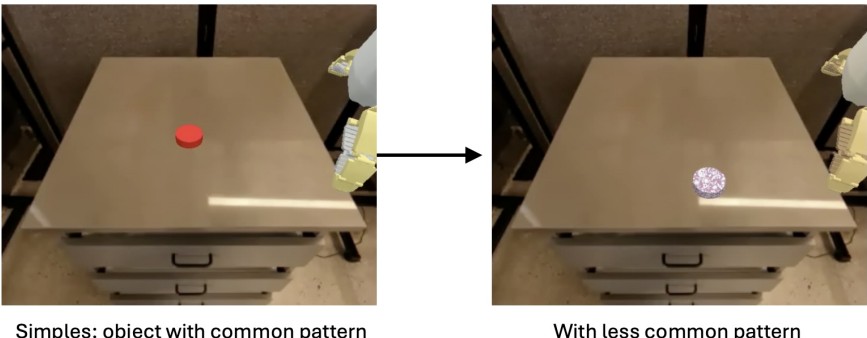

Simples: object with common pattern      With less common pattern

Figure 51: Ablation study on VIMABench focusing on pattern changes. Success rates for grasping vary significantly depending on the object pattern.

The VIMABench ablation study primarily examines the impact of pattern changes on task performance. As shown in Figure 51, RT-1's success rate for grasping dropped from **90%** for a red circle to **40%** for a purple paisley circle. This shows the model's sensitivity to less common object or pattern, a limitation that can hinder its robustness in scenarios with diverse or out-of-distribution objects and patterns.

**Additional Evaluation of OpenVLA in Libero.** Given OpenVLA's consistent 0% success rate in the SIMPLER Environment across tasks, we conducted additional evaluations using Libero (Liu et al., 2023a), a dataset specifically used to fine-tune OpenVLA-Libero. For in-distribution Libero objects, which primarily resemble sauce bottles and other common objects, OpenVLA achieved a 70% success rate in simple pick-and-place tasks. However, when the objects were replaced with pure, vibrant-colored cylinders (e.g., red, green, and blue), as shown in Figure 52, the success rate dropped to 0%. This result suggests that the failures observed in the SIMPLER Environment were not solely due to real-to-sim transfer issues but were instead indicative of OpenVLA's limited ability to generalize to out-of-distribution objects or layouts.

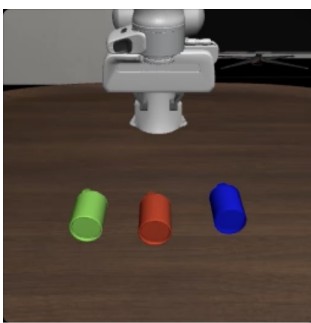

Figure 52: Evaluation of OpenVLA on Libero with a combination of in-distribution objects and out-of-distribution color-coded setups.

### G.5.5    Discussion

Our results highlight the trade-offs between specialized fine-tuned VLA models and general-purpose VLLM-based methods.

**Performance on novel and complex tasks.** While RT-1 and OpenVLA excel at tasks aligned with their training distributions, they struggle significantly under even mild domain shifts, such as novel objects, unexpected layouts, or implicit multi-step reasoning. For example, OpenVLA often remained idle when confronted with unseen conditions, and RT-1's performance deteriorated sharply as task complexity increased.

**Advantages of fine-tuned VLA models.** Fine-tuned VLA models remain valuable in scenarios closely aligned with their training data, where they can generate valid actions without additional post-processing. However, extending their capabilities to handle variations in object properties, camera viewpoints, and environmental contexts is a critical challenge.

**The trade-off between reasoning and action generation.** A shift toward action generation through fine-tuning raises another question: does this focus come at the expense of higher-level reasoning and planning abilities? Investigating this trade-off is essential for understanding the limitations and opportunities of fine-tuned models.

**Generalist robotics systems and the roles of VLLMs and VLAs.** This brings us to a key question for future research: should a single fine-tuned VLA model manage the entire robotics pipeline—from perception to planning to action—or is a modular, hierarchical approach more effective? The trade-off between reasoning and action generation lies at the core of this discussion, as a greater focus on action outputs may come at the expense of reasoning depth and planning flexibility.

From our experiments with Wonderful Team, even with GPT-4o—a state-of-the-art general-purpose reasoning model—a hierarchical structure with self-correction mechanisms proved essential for achieving robust performance across diverse robotics tasks. This suggests that robotics systems benefit significantly from a division of responsibilities: VLA models can excel as specialized modules for precise, subgoal-level action generation, while VLLMs, with their superior reasoning and world-modeling capabilities, handle high-level planning, abstract decision-making, and failure correction through language. Importantly, our insights suggest that VLLMs and VLAs need not be substitutes for one another but rather complementary components.

