# OpenReview forum: "Wonderful Team: Zero-Shot Physical Task Planning with Visual LLMs"
_TMLR — Accepted by TMLR_

### Review · Reviewer_245D · 2024-12-21

**Summary Of Contributions:**

The paper introduces "Wonderful Team", a zero-shot high-level robotic planning framework that uses a multi-agent VLLM to integrate perception, planning, and control seamlessly. The agentic system addresses key challenges in robotic manipulation and demonstrates robust self-correction capabilities. Without requiring training or fine-tuning, Wonderful Team achieves significant performance gains over prior methods in both simulated and real-world tasks, with improvements ranging from 30-70% in success rates for tasks involving semantic reasoning and spatial planning. Through extensive experiments and ablation studies, the framework highlights the growing potential of VLLMs for adaptable, context-aware robotic systems.

**Audience:**

Yes

**Broader Impact Concerns:**

The work might involve malicious applications, such as surveillance, weaponized robotics, or unauthorized data gathering.

**Claims And Evidence:**

Yes

**Requested Changes:**

**Q1**: How does the system handle scenarios where iterative self-correction fails or is computationally expensive?

**Q2**: The proposed baselines all rely on predefined modules or training. It is unclear if the comparison is entirely fair, given that Wonderful Team is designed as a zero-shot approach. How does Wonderful Team perform against the systems employing fine-tuning or hybrid approaches?

**Q3**: The framework's focus is on 3D manipulation and high-level task planning. Could the multi-agent VLLM framework be well adapted for broader robotic applications like navigation or low-level control?

Overall, I think this work deserves acceptance because of its technical solidness and extensive experimental analysis. I would be more convinced if the above questions could be addressed.

**Strengths And Weaknesses:**

**Strengths**

-	Performing perception, reasoning, and planning within a unified architecture is interesting compared to traditional modular pipelines.
-	The work validates its framework through extensive and diverse experimental settings. The ablation studies of analyzing individual components' contributions underscores the approach's robustness.
-	The paper is well-structured and effectively communicates complex ideas, making it accessible to a broad audience in robotics and AI.

**Weaknesses**

-	The reliance on VLLM self-correction for addressing initial errors might introduce inefficiencies, especially for tasks where real-time performance is critical.
-	Most of the real-world tasks seem constrained to controlled setups. Its performance in less controlled, noisy, or adversarial environments has not been involved.

---

> ### Author Response · Authors · 2025-01-12
> **Efficiency and Failure Handling in VLLM Self-Correction**
>
> We thank the reviewer for their thoughtful comments and valuable feedback!
>
> **Q: How does the system handle scenarios where iterative self-correction fails or is computationally expensive?**
>
> We acknowledge the trade-offs in using VLLM self-correction, particularly for tasks where real-time performance is critical. The iterative process inherently increases inference time, making it more suitable for static environments where accuracy takes precedence. In dynamic scenarios, even a single inference may become outdated, and additional iterations can further increase computational cost. To mitigate these challenges, our system is designed with the following measures:
>
> 1. **Task-Specific Limit:** Supervisor Agent determines the appropriate number of self-corrections based on the task scenario (e.g., time-sensitive vs. controlled tasks). This value is further capped by a predefined system-level limit.
> 2. **Failure Esclation:** The Grounding Mover and Checker collaborate through iterative checks to refine object localization. However, if localization fails after the allowed attempts—often due to the Grounding Manager providing an incorrect initial starting point—the system may encounter issues, such as the correct object being outside the zoomed view, which makes correction impossible. This can lead to prolonged correction loops. When this happens, the system reports the failure to the Grounding Manager, who selects a new, more feasible starting point. If failures persist and reach the predefined cap, the issue is escalated further. The Supervisor Agent then intervenes to reassess the situation and may propose a revised plan, terminate the task, or request additional input or resources.
> 3. **Efficient Verification:** From our experiments, we found that subgoal corrections are simpler than localization tasks and are typically resolved within a single iteration due to GPT-4o’s strong verbal reasoning skills. Thus, to maximize efficiency, we designed the system so that the Verification Agent provides the Supervisor with only one opportunity to revise each subgoal step. This ensures that corrections remain focused and computational costs are controlled.
>
> By integrating task-specific configurations, failure escalation, and single-round verification, our system addresses challenges like uncorrectable failures from imperfect starting points, balancing efficiency and accuracy. However, we acknowledge that this system design, together with the reliance on a Vision Large Language Model as its basis, have limitations in real-time, fast-changing dynamic environments due to outdated inferences. We see this as an exciting direction for future work.

---

> ### Author Response · Authors · 2025-01-13
> **Discussions on Broader Application**
>
> **Q: The framework's focus is on 3D manipulation and high-level task planning. Could the multi-agent VLLM framework be well adapted for broader robotic applications like navigation or low-level control?**
>
> We acknowledge that our real-world experiments primarily focus on structured, table-top settings, as this aligns with the main scope of our work. However, while the system is designed with such tasks in mind, the framework aims to maximize the capabilities of VLLMs for tasks requiring both complex reasoning and accurate grounding, making it adaptable to broader domains. We appreciate this insightful question and have worked diligently to explore possible applications beyond structured settings. To answer this question with detailed example results, we have included **Appendix B** in the revised manuscript (pages 17–21), which provides a detailed scope and summary of experiments beyond structured table-top environments. Notably, the Wonderful Team framework is not limited to such scenarios, as highlighted by the following key results:
>
> 1. **Unstructured & Noisy Environments (Appendix B.2**): We present an example of Wonderful Team successfully handling partially occluded objects in a cluttered cabinet, where it identified ingredients like soy sauce and miso paste among varied packaging. This demonstrates its capability for complex perception and robustness in unstructured and noisy environments.
>
> 2. **Navigation Tasks (Appendix B.3**): We explored the application of Wonderful Team in a navigation scenario within a simplified home environment. The system identified obstacles, planned collision-free paths, and refined target locations for pick-up and drop-off subgoals. These preliminary results indicate that the multi-agent architecture may be extended to navigation tasks with appropriate task-specific adjustments.
>
> We greatly appreciate the reviewers' insights and encourage them to refer to the revised manuscript for further details and discussions!

---

> ### Author Response · Authors · 2025-01-13
> **Additional Experiments on Comparison with Fine-Tuned VLA Models**
>
> **Q: The proposed baselines all rely on predefined modules or training. It is unclear if the comparison is entirely fair, given that Wonderful Team is designed as a zero-shot approach. How does Wonderful Team perform against the systems employing fine-tuning or hybrid approaches?**
>
> We greatly appreciate this insightful question and have taken considerable steps to address it comprehensively. In the updated manuscript, we have significantly expanded our comparison with fine-tuned Vision-Language-Action (VLA) models. Specifically, **Appendix Section G.5** (pages 65–69) provides a detailed side-by-side evaluation of RT-1 and OpenVLA against Wonderful Team (WT) in zero-shot settings.
>
> Despite efforts to minimize the domain gap, RT-1 and OpenVLA demonstrated limited success on the evaluated tasks, achieving near 0% success on even the simplest tasks compared to WT’s 100% success rate. To better understand their limitations, we conducted an ablation study starting with the simplest possible version of the task, progressively increasing task complexity (e.g., more fruits, colors, and reasoning challenges, as detailed in **Appendix Section D.3**). For instance, in simpler environments, such as a setup with only one apple and one color, and a concrete prompt like “Place the apple in the red area,” RT-1 performed well, achieving 100% success in grasping the correct object. However, as task complexity grew—requiring reasoning about novel objects, unexpected layouts, or implicit multi-step reasoning—RT-1 and OpenVLA struggled significantly. This highlights their key limitations in generalizing beyond their training distribution and adapting to domain shifts. We believe these results provide valuable insights into the trade-offs between fine-tuned VLA models and general-purpose systems like Wonderful Team. Thank you for raising this important point!

---

### Review · Reviewer_SRrs · 2024-12-29

**Summary Of Contributions:**

Wonderful Team is a multi-agent framework utilizing Visual Large Language Models (VLLMs) for high-level robotic planning. Given an image of the environment and a task description, the system generates the necessary sequence of actions for a robot to complete the task. Unlike methods relying on separate vision systems, Wonderful Team integrates perception, control, and planning within the VLLM, leading to more robust performance.

The authors provide a few motivating examples, which highlight the limitations of using separate LLMs and vision models for robotic planning.  LLMs struggle with ambiguous prompts requiring visual context (e.g., ranking fruits by price).  Separate vision models like LangSAM, while precise, lack contextual awareness, misinterpret spatial instructions, and struggle with uncommon objects. Replacing LangSAM with a VLLM alone doesn't solve the problem due to imprecision. However, the key observation is that VLLMs possess self-correction capabilities, motivating the development of Wonderful Team, which leverages these capabilities for more robust performance.

Experiments show that Wonderful Team outperforms baselines like Trajectory Generator and NLaP on VimaBench tasks and real-world scenarios involving implicit goal inference (e.g., placing objects based on color) and spatial planning (e.g., drawing a star).  The integrated approach allows for more nuanced reasoning and context understanding compared to methods using separate vision models.

**Audience:**

Yes

**Claims And Evidence:**

Yes

**Requested Changes:**

See weaknesses.

**Strengths And Weaknesses:**

## Strengths

- This paper presents compelling motivating examples that effectively highlight the limitations of previous methods and underscore the necessity for tighter integration between perception and planning. The question-answer structure enhances the clarity and readability of the presentation.

- The multi-agent VLLM framework is thoughtfully designed, incorporating distinct agents for task-level, target-level, and subgoal-level planning, which collaborate to address task planning challenges. This integrated design fosters stronger coordination between perception, planning, and action.

- The framework’s emphasis on self-correction is particularly noteworthy, enabling the system to detect and resolve errors in both perception and planning, thereby ensuring more robust and reliable performance.

- The paper presents compelling experimental results, demonstrating significant improvements over existing methods, including those using separate vision models and even some training-based approaches, across various simulated and real-world tasks.


## Weaknesses

- The proposed system is limited to generating high-level semantic actions. For tasks requiring fine-grained actions, such as making dumplings, the system would likely fail to perform effectively.

- The authors suggest that VLMs possess the ability to iteratively self-correct errors in perception. However, this claim necessitates more thorough validation. Is this capability universally applicable? If not, what are the limitations or scope of this assertion?

- If VLMs can verify detection results, why not employ a specialized detection model and then use VLMs solely to verify the outputs from the detection model?

- The proposed system appears to be specifically designed for 3rd-person view table-top pick-and-place tasks.

---

> ### Author Response · Authors · 2025-01-12
> **Clarifying System Limitations and Capabilities: Extending Beyond Semantic Actions and Tabletop Tasks**
>
> **Re: “The proposed system is limited to generating high-level semantic actions. For tasks requiring fine-grained actions, such as making dumplings, the system would likely fail to perform effectively.”**
>
> We acknowledge that our system is designed to output coordinate-level actions rather than directly controlling forces or torques, which limits its suitability for tasks requiring highly precise force application that cannot be effectively summarized through coordinate-level commands. Consequently, the applicability of our framework depends on whether a task can be decomposed into subgoals defined by coordinate-level control.
>
> Within these constraints, however, **our framework extends beyond simple semantic actions and basic pick-and-place tasks.** For example, in our Spatial Planning experiments discussed in **Section 5.3**, the task of drawing a five-pointed star requires the robot to plan and execute a complex trajectory with multiple intermediate waypoints. The precision necessary to form a perfect five-pointed star on the notebook highlights the system’s ability to perform fine-grained control beyond basic semantic actions. This capability is made possible by the Supervisor’s flexibility in generating and executing more complex functions when tasks demand fine-grained coordinate-level control.
>
> Thus, we agree that replicating very complex tasks, such as dumpling-making, in the most human-like manner—where decisions depend on precise force application, like how hard to press based on the dough’s shape—would not be possible within our current framework. However, if we view tasks like dumpling-making through a robotics lens, as demonstrated in RoboCook [1], such tasks can be decomposed into subgoals that our framework can effectively handle at the coordinate level. For example, following the RoboCook approach, the task can be divided into sequential steps such as initiating the dough, cutting it in half with a tool, pinching the dough, pressing, rolling, etc.
> Consider one of the more complex steps: rolling the dough. This step might involve:
> 1. *Picking up the rolling pin from a designated tools section.*
> 2. *Lowering it to the dough at a chosen midpoint.*
> 3. *Rolling outward to flatten the dough’s edge.*
> 4. *Lifting the gripper.*
> 5. *Returning to the midpoint for another pass.*
>
> Repeating these actions at different angles helps form a circular wrapper. Similarly, subsequent tasks like “cutting the dough,” “adding filling,” and “folding the wrapper” can also be broken into coordinate-based subtasks that align with our framework’s capabilities. While this process does not replicate the human dexterity seen in traditional dumpling-making, it demonstrates how our system can generate structured, sequential actions to complete fine-grained tasks. Once such tasks are broken into subgoals, they become transferable to our framework, showing its potential for handling complex tasks in a structured and systematic manner.
>
> [1] RoboCook: Long-Horizon Elasto-Plastic Object Manipulation with Diverse Tools, Shi et al. 2023
>
> **Re: “The proposed system appears to be specifically designed for 3rd-person view table-top pick-and-place tasks.”**
>
> As mentioned, we would like to clarify that while our paper focuses on 3rd-person table-top manipulation, our system is not limited to pick-and-place tasks. Within the scope of table-top manipulation, we evaluated tasks such as sweeping, drawing a star, and opening a bottle cap.
>
> The scope of this paper, both experimentally and in system design, is focused on table-top manipulation, allowing us to thoroughly evaluate and validate its capabilities in a well-defined and structured domain. However, the framework’s core innovation lies in its robust system design, which integrates VLLMs to address tasks that demand complex reasoning and precise grounding, giving it the potential to adapt to broader domains. To address this point with concrete examples, we have expanded **Appendix Section B** in the revised manuscript (pages 17–21) to provide a detailed scope and examples beyond table-top manipulation. Notably, the Wonderful Team framework is not constrained to these scenarios, as highlighted by the following key results:
>
> 1. **Unstructured Environments (Appendix B.2):** We present an example of Wonderful Team successfully handling partially occluded objects in a cluttered cabinet, where it identified ingredients like soy sauce and miso paste among varied packaging. This demonstrates its capability for complex perception and robustness in unstructured and noisy environments.
> 2. **Navigation Scenarios (Appendix B.3):** We explored adapting our framework to visual navigation tasks in simplified home environments. The system successfully identified obstacles, planned collision-free paths, and refined target locations for pick-up and drop-off subgoals. These results suggest that the multi-agent architecture can extend to navigation tasks with minimal task-specific adjustments..

---

> ### Author Response · Authors · 2025-01-12
> **Clarifying the Role of Vision LLMs in Detection and Grounding Tasks**
>
> **Re: “If VLMs can verify detection results, why not employ a specialized detection model and then use VLMs solely to verify the outputs from the detection model?”**
>
> We are thankful for the reviewer’s valuable feedback, as this touches on a core point of our motivation for using Vision LLMs to perform grounding rather than handing off all or part of the task to a specialized detection model. We greatly appreciate this observation and have **revised Section 2.1** on motivating examples to provide additional clarity on this point. Below, we further discuss our rationale:
>
> While Visual LLMs can verify detection results from specialized detection models, correcting detection errors remains a significant challenge. Detection models maintain full control over object identification, restricting Visual LLMs to iteratively refining prompts with alternative object descriptions. Most detection models we tested rely on short, phrase-based prompts to identify objects. However, our experiments showed that using similar prompts results in negligible changes to the output distribution, even when the initial detection is incorrect.
>
> Larger changes—such as shifting entirely from the object name (e.g., “soy sauce”) to a description of its shape or color (e.g., “tall, dark brown bottle”)—can sometimes yield different results. For instance, Figure 15 (Page 19) illustrates this soy sauce scenario. However, there is a finite set of descriptors available for any given object, which can lead to a dead-end. In some cases, this iterative process may result in an inefficient loop without successfully isolating the desired object, especially when the optimal summary phrase fails in the given context.
>
> Another potential solution involves having the detection model segment all visible objects or items and using Visual LLMs to select the correct one from the pool. However, in practice, we found that occlusion often excludes key objects from detection. This omission can make recovery impossible in such cases.
>
> These inefficiencies are further compounded by the inherent gap between detection models and Visual LLMs. Figures 1 (page 2) and 46 (**Appendix G.4.2**, page 63) illustrate cases where detection errors disrupt task execution despite the planning and grounding components functioning correctly. For example, the planner prompts the detection model to identify a “green area” for placing a lime and an “orange area” for placing an orange. However, the lime and orange are misidentified as the areas themselves. While seemingly minor, such errors disrupt task logic and underscore the detection model’s lack of contextual understanding—a gap our framework addresses through integrated feedback and coordination.
>
> Consequently, while VLLMs can identify detection issues, they lack the capacity to resolve them effectively without an integrated framework like Wonderful Team. Our framework incorporates a grounding team capable of:
>
> 1. **Receiving feedback** from higher-level agents.
> 2. **Iteratively correcting errors,** not just through self-correction but also based on external guidance.
>
> This design allows the system to overcome the limitations of standalone Visual LLMs and detection models, providing a robust mechanism for addressing detection errors.

---

> ### Author Response · Authors · 2025-01-12
> **Discussions on the Ability and Scope of Iterative Self-Correction in Vision LLMs**
>
> **Re: “The authors suggest that VLMs possess the ability to iteratively self-correct errors in perception. However, this claim necessitates more thorough validation. Is this capability universally applicable? If not, what are the limitations or scope of this assertion?”**
>
> **Validation:** To validate our claim that interactive self-correction enhances perception, we conducted a comprehensive ablation study analyzing the contribution of each component within the Wonderful Team framework. These results, detailed in **Appendix Section E** (Pages 32–34), highlight the critical role played by each part of the system. **Specifically, we examined the effect of removing the iterative correction process entirely from the grounding team** by disabling both the Box Checking Agent and the Box Moving Agent. In this setting, the initial grounding position determined by the Grounding Manager was used directly, without any subsequent verification or adjustment. This removal severely impacted the system’s ability to select precise action points, causing a dramatic drop in success rate from **93% to 30%** (Table 11, Page 34).
>
> Additionally, we conducted ablation studies on our toy experiment to explain the motivation behind the self-correction system, as detailed in **Appendix Section F** (Page 39). From these experiments, we observed the following:
>
> 1. *Initial Object Identification:* Even in the simplified toy environment, gpt-4o only achieved a **33%** success rate in correctly identifying the target object on the first attempt. However, 90% of the proposed points were not significantly off, falling within 3x the target radius. This demonstrates that these approximate points can serve as effective starting positions for further iterative refinement.
> 2. *Bounding Box Validation:* gpt-4o showed a **97%** success rate in determining whether a bounding box was sufficiently accurate, highlighting its reliability as a checker.
>
> These results highlight the potential of self-correction mechanisms to harness the strengths of VLLMs. By iteratively refining predictions, the system can significantly improve performance, even when initial attempts are imperfect.
>
> **Limitations and Scope:** Building on the key observations discussed above, our framework relies on two assumptions about the capabilities of VLLMs:
> 1. *Reasonable Initial Guesses:* They can make initial predictions that, while approximate, are not entirely off-base and fall within a reasonable margin of error. We use this capability by allowing the grounding manager to identify an initial point and start the iterative self-correction process with a zoomed-in image focused on the region of interest.
> 2. *Reliable Validation:* They are highly effective at distinguishing between good and bad bounding boxes. This capability forms the basis of our interactive correction process with the grounding checker being the judge, enabling systematic refinement of predictions through iterative feedback and validation.
>
> Together, these assumptions imply that the VLLM has a foundational understanding of the environment, even if it lacks perfect accuracy. The effectiveness of self-correction diminishes in scenarios where these assumptions break down. For instance, if the environment contains excessive noise or overwhelming visual information, the VLLM may fail to generate meaningful initial guesses or accurately validate bounding boxes. Although we allow the grounding checker and manager to zoom in on specific regions of interest, it cannot fully resolve cases where the input remains too cluttered. Additionally, When an object is mostly occluded, the model may lack sufficient information to make accurate predictions. In such cases, self-correction cannot compensate for the inherent limitations of the base model’s perception.

---

### Review · Reviewer_DGSH · 2025-01-02

**Summary Of Contributions:**

The Wonderful Team framework is a novel multi-agent VLLM approach for zero-shot high-level robotic planning. This framework allows for integrated perception, control, and planning within a single VLLM by utilizing a multi-agent structure with specialized agents and internal memory for context. This differs from other methods that use separate vision systems.  Wonderful Team also has the ability to self-correct at both the planning and perception levels. It has shown significant improvements over existing methods in both simulation (VIMABench) and real-world tasks,  with a 40% improvement over NLaP on VimaBench and a 30% improvement over Trajectory Generators on drawing/wiping tasks.

**Audience:**

Yes

**Broader Impact Concerns:**

● Positive Impacts: Potential to improve robot autonomy, impacting manufacturing, logistics, and healthcare.
● Potential Misuse: Potential for misuse, particularly in autonomous weapon systems or surveillance.
● Job Displacement: Increased automation could lead to job displacement.
● Accessibility: Uneven accessibility to resources and expertise for use of these technologies.

**Claims And Evidence:**

Yes

**Requested Changes:**

● Incorporate robust 3D reasoning and planning capabilities.
● Implement real-time dynamic error detection and more efficient replanning.
● Address issues with partial observability.
● Expand scope to other robotic domains.
● Provide more detailed comparison with Trajectory Generator, addressing the limitations of LangSAM.
● Discuss practical limitations in complex real-world environments.

**Strengths And Weaknesses:**

●Strengths:
○ Zero-shot capability allows for new environments without training.
○ Integrated planning and perception reduces reliance on brittle external systems.
○ Self-correction improves success rates.
○ Multi-agent design improves precision and efficiency.
○ Empirical validation in simulation and real-world settings.
○ Addresses limitations of methods using separate LLMs and vision models.
○ Outperforms fine-tuned models in zero-shot settings.

● Weaknesses:
○ Limited 3D reasoning, largely confined to 2D space.
○ Struggles with partial observability.
○ Lacks real-time dynamic error detection, with computationally expensive replanning.
○ Focuses on high-level planning, not addressing visual navigation or low-level kinematic control.

---

> ### Author Response · Authors · 2025-01-12
> **Detailed Comparison with Trajectory Generator**
>
> We sincerely thank the reviewer for their time and valuable input!
>
> **Re: provide more detailed comparison with Trajectory Generator, addressing the limitations of LangSAM**
>
> **Appendix Section G.3** (Pages 55–59) provides an in-depth evaluation of the Trajectory Generator and our framework in the Gymnasium Fetch environment for 3D pick-and-place tasks. This analysis highlights the fundamental differences between the two systems, underscores the limitations of LangSAM, and demonstrates the strengths of our multi-agent approach.
>
> Specifically, LangSAM faces significant challenges in correctly identifying objects, as shown in Figure 41 (Page 58). For example:
>
> - When asked to locate the wooden toy train, LangSAM incorrectly identifies the Fetch robot.
> - When tasked with finding the lid, it misidentifies the entire table.
> - When asked to identify the rightmost object, it again points to the Fetch robot.
> - Similarly, when tasked with locating the tomato soup can, it mistakenly points to the mustard bottle.
>
> These examples highlight a critical limitation of LangSAM: its static approach to object identification makes correcting detection errors inherently challenging. LangSAM's design prevents it from adapting dynamically to errors, leaving Visual LLMs (VLLMs) to rely on iterative prompt refinements without the ability to directly control object identification. This limitation often compounds errors, as illustrated in Figure 15 (Page 19), where even an exhaustive list of keywords fails to resolve detection issues. A similar segmentation model, GroundingDINO, demonstrates comparable shortcomings, making recovery nearly impossible even with VLLM corrections.
>
> In contrast, our multi-agent framework incorporates iterative self-correction and multi-attempt replanning, significantly enhancing performance in complex 3D environments with gripper torque control. As demonstrated in Section G.3:
>
> - Allowing multi-attempt replanning in our Wonderful Team framework improved the success rate from **50% to 80%**, primarily due to iterative refinement of action points for more stable picking.
> - By comparison, the Trajectory Generator showed a limited success rate improvement from **0% to 5%**, due to poor initial segmentation and an inability to recover from detection errors.
>
> These results highlight the critical advantage of our multi-agent approach with a grounding module based on gpt-4o that’s capable of both receiving feedback from higher-level agents and iteratively correcting errors, not just through self-correction but also based on external guidance. This design allows the system to overcome the limitations of standalone Visual LLMs and detection models, providing a robust mechanism for addressing detection errors. Extensive comparisons with PIVOT, MOKA, and fine-tuned VLA models are provided in **Section G**, starting on Page 43, and we invite the reviewers to refer to this section for further details.

---

> > ### Author Response · Authors · 2025-01-12
> > **Discussions on Limitations**
> >
> > **Re: robust 3D reasoning and planning capabilities and issues with partial observability**
> >
> > As outlined in our limitations, integrating depth cameras allows Vision LLMs to operate in 3D space, but their reasoning and planning remain largely confined to 2D. This is due to the 2D nature of most image inputs, with depth images providing limited 3D spatial information. VLLMs are pretrained on 2D RGB images, making the inclusion of depth data practical but insufficient for fully capturing 3D relationships. Although richer representations, such as 3D meshes or point clouds, could theoretically enhance detail, they introduce a significant distribution shift from the training data of pretrained visual large language models, making them less effective without extensive fine-tuning. Addressing these challenges falls beyond the scope of our current work. To address partial observability, our system uses top and front views of the environment at the start of each task, improving spatial understanding of object relationships. When objects remain hidden, the Supervisor Agent can request additional perspectives (e.g., from the left or right). The success of this approach depends on the hardware setup and the specific task’s camera configuration.
> >
> > **Re: real-time dynamic error detection and more efficient replanning**
> >
> > We acknowledge the limitation of lacking real-time dynamic error detection, as highlighted in our limitations section. While integrating such a mechanism using VLLMs is theoretically possible, it is very computationally intensive. Streaming environment images to the VLLM in real time would not be practical for rapidly changing environments, as the inference speed of VLLMs would lag significantly behind environmental changes, making the detection outdated by the time it is processed. Our framework already employs two modes of error correction tailored to our scope of static or slow-changing environments:
> >
> > 1. *Proactive Correction (Pre-Execution):*  Before executing an action sequence, our Verification Agent extensively analyzes the plan for potential failures or infeasible states based on task and environment constraints. This preventive approach ensures that many errors are addressed before they occur, making it highly effective in static settings where the goal remains stable.
> > 2. *Reactive Correction (Post-Execution):* After executing the action sequence, our Replanning Agent identifies failures, examines their causes, and provides improvement suggestions for subsequent attempts. This allows the system to adapt iteratively based on observed outcomes.
> >
> > **Re: Discuss practical limitations in complex real-world environments**
> >
> > Our self-correction system leverages Vision-Language Models (VLLMs) to iteratively refine predictions, relying on two key assumptions: reasonable initial guesses and reliable bounding box validation. While effective in structured environments, its practical limitations arise in scenarios where these assumptions break down. In extremely noisy or cluttered settings that are hard to read by humans, VLLMs may fail to generate meaningful initial guesses, even with zooming, and struggle with occluded or partially visible objects, where insufficient input data limits accuracy. Additionally, bounding box validation can fail in cases with ambiguous edges or overlapping objects, compounding errors that self-correction cannot resolve. These challenges highlight the framework's dependency on VLLMs' foundational understanding of the environment.

---

> ### Author Response · Authors · 2025-01-12
> **Discussions on Wonderful Team's Applications Beyond High-Level Pick-and Place Tasks**
>
> **Re: expand scope to other robotic domains**
>
> While our primary focus is on structured, table-top tasks, this scope was chosen to align with the main objectives of our work. However, our framework is designed to leverage the capabilities of VLLMs for tasks requiring both complex reasoning and precise grounding, making it adaptable to a broader range of applications, including navigation and kinematic control.
>
> To address this question, we have expanded our analysis in the revised manuscript. **Appendix B** (pages 17–21) outlines experiments and results beyond structured settings, highlighting the framework’s versatility:
>
> 1. **Tasks Beyond Pick-and-Place (Appendix B.1):** We evaluated tasks requiring complex spatial reasoning and fine-grained motion control, such as the VIMABench “Sweep Without Exceeding” task and drawing a five-pointed star. These tasks demonstrate the system’s ability to handle non-trivial motion planning and trajectory execution. For example, the star-drawing task showcases how our framework can break down tasks into coordinate-level subgoals, allowing for precise planning and execution of low-level trajectories.
> 2. **Potential Applications in Unstructured Environments (Appendix B.2):** Experiments in cluttered and noisy settings, such as identifying and manipulating partially occluded objects in a cabinet, demonstrate the robustness of the system in handling real-world variability and perception challenges.
> 3. **Navigation Scenarios (Appendix B.3):** We explored adapting our framework to visual navigation tasks in simplified home environments. The system successfully identified obstacles, planned collision-free paths, and refined target locations for pick-up/drop-off subgoals. These results suggest that our multi-agent architecture can extend to navigation tasks with minimal task-specific modifications.
>
> Additionally, while our system outputs coordinates rather than directly controlling forces or torques, it is capable of handling tasks requiring fine-grained manipulation if the tasks can be decomposed into coordinate-level subgoals. For example, in the star-drawing task, the Supervisor Agent generates a high-level plan, and the Grounding Team pinpoints the precise starting point. The trajectory, composed of sequential coordinate-level instructions, allows the system to execute structured and precise actions. While this approach may not replicate human dexterity, it demonstrates how our framework can handle fine-grained tasks once they are translated into coordinate-level instructions.
>
> In summary, while our system has limitations in tasks requiring precise force application, the experiments in Appendix B highlight its adaptability to tasks beyond structured environments, including navigation, unstructured perception, and fine-grained kinematic control. These results broaden the scope of our contributions and illustrate the system’s potential for diverse real-world robotics applications.

---

### Author Response · Authors · 2025-01-13
**Revisions in the Updated Manuscript**

Dear Reviewers,

We sincerely thank you for your valuable feedback. We have carefully revised the manuscript to address your concerns and incorporated significant changes to clarify and strengthen our work. Additionally, we have added 2 new appendices with experiments to tackle two key areas:

1. **Scope Clarification and Expanded Testing (Appendix Section B, Pages 17–21):** We have further clarified the scope of our tabletop manipulation problem and provided new examples showcasing the performance of our framework beyond its primary scope. Our results demonstrate that Wonderful Team continues to generate reliable grounding results in non-tabletop, cluttered environments without any modifications to the prompt or system. Additionally, we extended the framework to simple navigation tasks with minimal prompt adaptations, highlighting its potential applicability to broader scenarios.

2. **Comparison with Fine-Tuned Models (Appendix Section G.5, Pages 65–69):** We also worked extensively to conduct a comprehensive comparison with fine-tuned Vision-Language-Action (VLA) models, including RT-1 and OpenVLA, alongside detailed ablation studies to analyze their failure modes. This section explores the limitations of these models compared to our approach and highlights the strengths of our framework in handling complex tasks more effectively.

We encourage you to review these updates and hope they provide a deeper understanding of the robustness and generalizability of our framework. Thank you once again for your thoughtful feedback and for helping us refine this work.

---

### Decision · Action_Editor_b8sM · 2025-01-30

**Recommendation:** Accept as is

**Comment:**

All reviewers unanimously recommend acceptance and AE echos the supportive sentiment!

**Audience:**

Yes

**Claims And Evidence:**

This paper introduces Wonderful Team, a multi-agent Vision Large Language Model (VLLM) framework for executing high-level robotic planning in a zero-shot regime. The multi-agent VLLM framework is thoughtfully designed, incorporating distinct agents for task-level, target-level, and subgoal-level planning, which collaborate to address task-planning challenges. This integrated design fosters stronger coordination between perception, planning, and action. Extensive experiments show that Wonderful Team outperforms baselines like Trajectory Generator and NLaP on VimaBench tasks and real-world scenarios involving implicit goal inference (e.g., placing objects based on color) and spatial planning (e.g., drawing a star).

---

> ### Author Response · Authors · 2025-02-03
> **Camera Ready Revision**
>
> We sincerely thank the reviewers, Action Editor, and Editors-in-Chief for their efforts. We received valuable feedback from the review process, which has helped us refine the manuscript.
>
> We have uploaded the final camera-ready version, incorporating all additional experiments, discussions, and minor revisions for clarity. The official implementation codebase is now linked for transparency and reproducibility. We thank the TMLR community for their support.